# A Theoretical Understanding of shallow Vision Transformers: Learning, Generalization, and Sample Complexity

**Hongkang Li** [*]         **Meng Wang** [†]         **Sijia Liu** [‡]         **Pin-Yu Chen** [§]

## Abstract

Vision Transformers (ViTs) with self-attention modules have recently achieved great empirical success in many vision tasks. Due to non-convex interactions across layers, however, the theoretical learning and generalization analysis is mostly elusive. Based on a data model characterizing both label-relevant and label-irrelevant tokens, this paper provides the first theoretical analysis of training a shallow ViT, i.e., one self-attention layer followed by a two-layer perceptron, for a classification task. We characterize the sample complexity to achieve a zero generalization error. Our sample complexity bound is positively correlated with the inverse of the fraction of label-relevant tokens, the token noise level, and the initial model error. We also prove that a training process using stochastic gradient descent (SGD) leads to a sparse attention map, which is a formal verification of the general intuition about the success of attention. Moreover, this paper indicates that a proper token sparsification can improve the test performance by removing label-irrelevant and/or noisy tokens, including spurious correlations. Empirical experiments on synthetic data and CIFAR-10 dataset justify our theoretical results and generalize to deeper ViTs.

## 1 Introduction

As the backbone of Transformers (Vaswani et al., 2017), the self-attention mechanism (Bahdanau et al., 2014) computes the feature representation by globally modeling long-range interactions within the input. Transformers have demonstrated tremendous empirical success in numerous areas, including nature language processing (Kenton & Toutanova, 2019; Radford et al., 2019; 2018; Brown et al., 2020), recommendation system (Zhou et al., 2018; Chen et al., 2019; Sun et al., 2019), and reinforcement learning (Chen et al., 2021; Janner et al., 2021; Zheng et al., 2022). Starting from the advent of Vision Transformer (ViT) (Dosovitskiy et al., 2020), Transformer-based models (Touvron et al., 2021; Jiang et al., 2021; Wang et al., 2021; Liu et al., 2021a) gradually replace convolutional neural network (CNN) architectures and become prevalent in vision tasks. Various techniques have been developed to train ViT efficiently. Among them, token sparsification (Pan et al., 2021; Rao et al., 2021; Liang et al., 2022; Tang et al., 2022; Yin et al., 2022) removes redundant tokens (image patches) of data to improve the computational complexity while maintaining a comparable learning performance. For example, Liang et al. (2022); Tang et al. (2022) prune tokens following criteria designed based on the magnitude of the attention map. Despite the remarkable empirical success, one fundamental question about training Transformers is still vastly open, which is

*Under what conditions does a Transformer achieve satisfactory generalization?*

Some recent works analyze Transformers theoretically from the perspective of proved Lipschitz constant of self-attention (James Vuckovic, 2020; Kim et al., 2021), properties of the neural tangent kernel (Hron et al., 2020; Yang, 2020) and expressive power and Turing-completeness (Dehghani et al., 2018; Yun et al., 2019; Bhattamishra et al., 2020a;b; Edelman et al., 2022; Dong et al., 2021;

---

[*]Rensselaer Polytechnic Institute. Email: `lih35@rpi.edu`

[†]Rensselaer Polytechnic Institute. Email: `wangm7@rpi.edu`

[‡]Michigan State University & IBM Research. Email: `liusiji5@msu.edu`

[§]IBM Research. Email: `Pin-Yu.Chen@ibm.com`

Likhosherstov et al., 2021; Cordonnier et al., 2019; Levine et al., 2020) with statistical guarantees (Snell et al., 2021; Wei et al., 2021). Likhosherstov et al. (2021) showed a model complexity for the function approximation of the self-attention module. Cordonnier et al. (2019) provided sufficient and necessary conditions for multi-head self-attention structures to simulate convolution layers. None of these works, however, characterize the generalization performance of the learned model theoretically. Only Edelman et al. (2022) theoretically proved that a single self-attention head can represent a sparse function of the input with a sample complexity for a generalization gap between the training loss and the test loss, but no discussion is provided regarding what algorithm to train the Transformer to achieve a desirable loss.

**Contributions**: To the best of our knowledge, this paper provides the first learning and generalization analysis of training a basic shallow Vision Transformer using stochastic gradient descent (SGD). This paper focuses on a binary classification problem on structured data, where tokens with discriminative patterns determine the label from a majority vote, while tokens with non-discriminative patterns do not affect the labels. We train a ViT containing a self-attention layer followed by a two-layer perceptron using SGD from a proper initial model. This paper explicitly characterizes the required number of training samples to achieve a desirable generalization performance, referred to as the sample complexity. Our sample complexity bound is positively correlated with the inverse of the fraction of label-relevant tokens, the token noise level, and the error from the initial model, indicating a better generalization performance on data with fewer label-irrelevant patterns and less noise from a better initial model. The highlights of our technical contributions include:

**First, this paper proposes a new analytical framework to tackle the non-convex optimization and generalization for shallow ViTs.** Due to the more involved non-convex interactions of learning parameters and diverse activation functions across layers, the ViT model, i.e., a three-layer neural network with one self-attention layer, considered in this paper is more complicated to analyze than three-layer CNNs considered in Allen-Zhu et al. (2019a); Allen-Zhu & Li (2019), the most complicated neural network model that has been analyzed so far for across-layer nonconvex interactions. We consider a structured data model with relaxed assumptions from existing models and establish a new analytical framework to overcome the new technical challenges to handle ViTs.

**Second, this paper theoretically depicts the evolution of the attention map during the training and characterizes how "attention" is paid to different tokens during the training.** Specifically, we show that under the structured data model, the learning parameters of the self-attention module grow in the direction that projects the data to the label-relevant patterns, resulting in an increasingly sparse attention map. This insight provides a theoretical justification of the magnitude-based token pruning methods such as (Liang et al., 2022; Tang et al., 2022) for efficient learning.

**Third, we provide a theoretical explanation for the improved generalization using token sparsification.** We quantitatively show that if a token sparsification method can remove class-irrelevant and/or highly noisy tokens, then the sample complexity is reduced while achieving the same testing accuracy. Moreover, token sparsification can also remove spurious correlations to improve the testing accuracy (Likhomanenko et al., 2021; Zhu et al., 2021a). This insight provides a guideline in designing token sparsification and few-shot learning methods for Transformer (He et al., 2022; Guibas et al., 2022).

## 1.1 BACKGROUND AND RELATED WORK

**Efficient ViT learning.** To alleviate the memory and computation burden in training (Dosovitskiy et al., 2020; Touvron et al., 2021; Wang et al., 2022), various acceleration techniques have been developed other than token sparsification. Zhu et al. (2021b) identifies the importance of different dimensions in each layer of ViTs and then executes model pruning. Liu et al. (2021b); Lin et al. (2022); Li et al. (2022d) quantize weights and inputs to compress the learning model. Li et al. (2022a) studies automated progressive learning that automatically increases the model capacity on-the-fly. Moreover, modifications of attention modules, such as the network architecture based on local attention (Wang et al., 2021; Liu et al., 2021a; Chu et al., 2021), can simplify the computation of global attention for acceleration.

**Theoretical analysis of learning and generalization of neural networks.** One line of research (Zhong et al., 2017b; Fu et al., 2020; Zhong et al., 2017a; Zhang et al., 2020a;b; Li et al., 2022c) analyzes the generalization performance when the number of neurons is smaller than the number

of training samples. The neural-tangent-kernel (NTK) analysis (Jacot et al., 2018; Allen-Zhu et al., 2019a;b; Arora et al., 2019; Cao & Gu, 2019; Zou & Gu, 2019; Du et al., 2019; Chen et al., 2020; Li et al., 2022b) considers strongly overparameterized networks and eliminates the nonconvex interactions across layers by linearizing the neural network around the initialization. The generalization performance is independent of the feature distribution and cannot explain the advantages of self-attention modules.

**Neural network learning on structured data.** Li & Liang (2018) provide the generalization analysis of a fully-connected neural network when the data comes from separated distributions. Daniely & Malach (2020); Shi et al. (2021); Karp et al. (2021); Brutzkus & Globerson (2021); Zhang et al. (2023) study fully connected networks and convolutional neural networks assuming that data contains discriminative patterns and background patterns. Allen-Zhu & Li (2022) illustrates the robustness of adversarial training by introducing the feature purification mechanism, in which neural networks with non-linear activation functions can memorize the data-dependent features. Wen & Li (2021) extends this framework to the area of self-supervised contrastive learning. All these works consider one-hidden-layer neural networks without self-attention.

**Notations**: Vectors are in bold lowercase, and matrices and tensors are in bold uppercase. Scalars are in normal fonts. Sets are in calligraphy font. For instance, $\boldsymbol{Z}$ is a matrix, and $\boldsymbol{z}$ is a vector. $z_i$ denotes the $i$-th entry of $\boldsymbol{z}$, and $Z_{i,j}$ denotes the $(i, j)$-th entry of $\boldsymbol{Z}$. $[K]$ ($K > 0$) denotes the set including integers from 1 to $K$. We follow the convention that $f(x) = O(g(x))$ (or $\Omega(g(x))$, $\Theta(g(x))$) means that $f(x)$ increases at most, at least, or in the order of $g(x)$, respectively.

## 2 PROBLEM FORMULATION AND LEARNING ALGORITHM

We study a binary classification problem [1] following the common setup in (Dosovitskiy et al., 2020; Touvron et al., 2021; Jiang et al., 2021). Given $N$ training samples $\{(\boldsymbol{X}^n, y^n)\}_{n=1}^N$ generated from an unknown distribution $\mathcal{D}$ and a fair initial model, the goal is to find an improved model that maps $\boldsymbol{X}$ to $y$ for any $(\boldsymbol{X}, y) \sim \mathcal{D}$. Here each data point contains $L$ tokens $\boldsymbol{x}_1^n, \boldsymbol{x}_2^n, \cdots, \boldsymbol{x}_L^n$, i.e., $\boldsymbol{X}^n = [\boldsymbol{x}_1^n, \cdots, \boldsymbol{x}_L^n] \in \mathbb{R}^{d \times L}$, where each token is $d$-dimensional and unit-norm. $y^n \in \{+1, -1\}$ is a scalar. A token can be an image patch (Dosovitskiy et al., 2020). We consider a general setup that also applies to token sparsification, where some tokens are set to zero to reduce the computational time. Let $\mathcal{S}^n \subseteq [L]$ denote the set of indices of remaining tokens in $\boldsymbol{X}^n$ after sparsification. Then $|\mathcal{S}^n| \leq L$, and $\mathcal{S}^n = [L]$ without token sparsification.

Learning is performed over a basic three-layer Vision Transformer, a neural network with a single-head self-attention layer and a two-layer fully connected network, as shown in (1). This is a simplified model of practical Vision Transformers (Dosovitskiy et al., 2020) to avoid unnecessary complications in analyzing the most critical component of ViTs, the self-attention.

$$F(\boldsymbol{X}^n) = \frac{1}{|\mathcal{S}^n|} \sum_{l \in \mathcal{S}^n} \boldsymbol{a}_{(l)}^\top \text{Relu}(\boldsymbol{W}_O \boldsymbol{W}_V \boldsymbol{X}^n \text{softmax}(\boldsymbol{X}^{n\top} \boldsymbol{W}_K^\top \boldsymbol{W}_Q \boldsymbol{x}_l^n)), \tag{1}$$

where the queue weights $\boldsymbol{W}_Q$ in $\mathbb{R}^{m_b \times d}$, the key weights $\boldsymbol{W}_K$ in $\mathbb{R}^{m_b \times d}$, and the value weights $\boldsymbol{W}_V$ in $\mathbb{R}^{m_a \times d}$ in the attention unit are multiplied with $\boldsymbol{X}^n$ to obtain the queue vector $\boldsymbol{W}_Q \boldsymbol{X}^n$, the key vector $\boldsymbol{W}_K \boldsymbol{X}^n$, and the value vector $\boldsymbol{W}_V \boldsymbol{X}^n$, respectively (Vaswani et al., 2017). $\boldsymbol{W}_O$ is in $\mathbb{R}^{m \times m_a}$ and $\boldsymbol{A} = (\boldsymbol{a}_{(1)}, \boldsymbol{a}_{(2)}, \cdots, \boldsymbol{a}_L)$ where $\boldsymbol{a}_{(l)} \in \mathbb{R}^m$, $l \in [L]$ are the hidden-layer and output-layer weights of the two-layer perceptron, respectively. $m$ is the number of neurons in the hidden layer. Relu : $\mathbb{R}^m \to \mathbb{R}^m$ where $\text{Relu}(\boldsymbol{x}) = \max\{\boldsymbol{x}, 0\}$. softmax : $\mathbb{R}^L \to \mathbb{R}^L$ where $\text{softmax}(\boldsymbol{x}) = (e^{x_1}, e^{x_2}, \cdots, e^{x_L}) / \sum_{i=1}^L e^{x_i}$. Let $\psi = (\boldsymbol{A}, \boldsymbol{W}_O, \boldsymbol{W}_V, \boldsymbol{W}_K, \boldsymbol{W}_Q)$ denote the set of parameters to train. The training problem minimizes the empirical risk $f_N(\psi)$,

$$\min_{\psi} : f_N(\psi) = \frac{1}{N} \sum_{n=1}^N \ell(\boldsymbol{X}^n, y^n; \psi), \tag{2}$$

where $\ell(\boldsymbol{X}^n, y^n; \psi)$ is the Hinge loss function, i.e.,

$$\ell(\boldsymbol{X}^n, y^n; \psi) = \max\{1 - y^n \cdot F(\boldsymbol{X}^n), 0\}. \tag{3}$$

---

[1]Extension to multi-classification is briefly discussed in Section G.1.

The generalization performance of a learned model $\psi$ is evaluated by the population risk $f(\psi)$, where

$$f(\psi) = f(\boldsymbol{A}, \boldsymbol{W}_O, \boldsymbol{W}_V, \boldsymbol{W}_K, \boldsymbol{W}_Q) = \mathbb{E}_{(\boldsymbol{X}, y) \sim \mathcal{D}}[\max\{1 - y \cdot F(\boldsymbol{X}), 0\}]. \tag{4}$$

The training problem (2) is solved via a mini-batch stochastic gradient descent (SGD), as summarized in Algorithm 1. At iteration $t$, $t = 0, 1, 2, \cdots, T - 1$, the gradient is computed using a mini-batch $\mathcal{B}_t$ with $|\mathcal{B}_t| = B$. The step size is $\eta$.

Similar to (Dosovitskiy et al., 2020; Touvron et al., 2021; Jiang et al., 2021), $\boldsymbol{W}_V^{(0)}$, $\boldsymbol{W}_Q^{(0)}$, and $\boldsymbol{W}_K^{(0)}$ come from an initial model. Every entry of $\boldsymbol{W}_O$ is generated from $\mathcal{N}(0, \xi^2)$. Every entry of $\boldsymbol{a}_l^{(0)}$ is sampled from $\{+\frac{1}{\sqrt{m}}, -\frac{1}{\sqrt{m}}\}$ with equal probability. $\boldsymbol{A}$ does not update during the training[2].

# 3 THEORETICAL RESULTS

## 3.1 MAIN THEORETICAL INSIGHTS

Before formally introducing our data model and main theory, we first summarize the major insights. We consider a data model where tokens are noisy versions of *label-relevant* patterns that determine the data label and *label-irrelevant* patterns that do not affect the label. $\alpha_*$ is the fraction of label-relevant tokens. $\sigma$ represents the initial model error, and $\tau$ characterizes the token noise level.

**(P1). A Convergence and sample complexity analysis of SGD to achieve zero generalization error.** We prove SGD with a proper initialization converges to a model with zero generalization error. The required number of iterations is proportional to $1/\alpha_*$ and $1/(\Theta(1) - \sigma - \tau)$. Our sample complexity bound is linear in $\alpha_*^{-2}$ and $(\Theta(1) - \sigma - \tau)^{-2}$. Therefore, the learning performance is improved, in the sense of a faster convergence and fewer training samples to achieve a desirable generalization, with a larger fraction of label-relevant patterns, a better initial model, and less token noise.

**(P2). A theoretical characterization of increased sparsity of the self-attention module during training.** We prove that the attention weights, which are softmax values of each token in the self-attention module, become increasingly sparse during the training, with non-zero weights concentrated at label-relevant tokens. This formally justifies the general intuition that the attention layer makes the neural network focus on the most important part of data.

**(P3). A theoretical guideline of designing token sparsification methods to reduce sample complexity.** Our sample complexity bound indicates that the required number of samples to achieve zero generalization can be reduced if a token sparsification method removes some label-irrelevant tokens (reducing $\alpha_*$), or tokens with large noise (reducing $\sigma$), or both. This insight provides a guideline to design proper token sparsification methods.

**(P4). A new theoretical framework to analyze the nonconvex interactions in three-layer ViTs.** This paper develops a new framework to analyze ViTs based on a more general data model than existing works like (Brutzkus & Globerson, 2021; Karp et al., 2021; Wen & Li, 2021). Compared with the nonconvex interactions in three-layer feedforward neural networks, analyzing ViTs has technical challenges that the softmax activation is highly non-linear, and the gradient computation on token correlations is complicated. We develop new tools to handle this problem by exploiting structures in the data and proving that SGD iterations increase the magnitude of label-relevant tokens only rather than label-irrelevant tokens. This theoretical framework is of independent interest and can potentially applied to analyze different variants of Transformers and attention mechanisms.

## 3.2 DATA MODEL

There are $M$ ($2 < M < m_a, m_b$) distinct patterns $\{\boldsymbol{\mu}_1, \boldsymbol{\mu}_2, \cdots, \boldsymbol{\mu}_M\}$ in $\mathbb{R}^d$, where $\boldsymbol{\mu}_1, \boldsymbol{\mu}_2$ are *discriminative patterns* that determine the binary labels, and the remaining $M - 2$ patterns $\boldsymbol{\mu}_3, \boldsymbol{\mu}_4, \cdots, \boldsymbol{\mu}_M$ are *non-discriminative patterns* that do not affect the labels. Let $\kappa =$

---

[2]It is common to fix the output layer weights as the random initialization in the theoretical analysis of neural networks, including NTK (Allen-Zhu et al., 2019a; Arora et al., 2019), model recovery (Zhong et al., 2017b), and feature learning (Karp et al., 2021; Allen-Zhu & Li, 2022) type of approaches. The optimization problem here of $\boldsymbol{W}_Q$, $\boldsymbol{W}_K$, $\boldsymbol{W}_V$, and $\boldsymbol{W}_O$ with non-linear activations is still highly non-convex and challenging.

$\min_{1 \leq i \neq j \leq M} \|\boldsymbol{\mu}_i - \boldsymbol{\mu}_j\| > 0$ denote the minimum distance between patterns. Each token $\boldsymbol{x}_l^n$ of $\boldsymbol{X}^n$ is a noisy version of one of the patterns, i.e.,

$$\min_{j \in [M]} \|\boldsymbol{x}_l^n - \boldsymbol{\mu}_j\| \leq \tau, \tag{5}$$

and the noise level $\tau < \kappa/4$. We take $\kappa - 4\tau$ as $\Theta(1)$ for the simplicity of presentation.

The label $y^n$ is determined by the tokens that correspond to discriminative patterns through a majority vote. If the number of tokens that are noisy versions of $\boldsymbol{\mu}_1$ is larger than the number of tokens that correspond to $\boldsymbol{\mu}_2$ in $\boldsymbol{X}^n$, then $y^n = 1$. In this case that the label $y^n = 1$, the tokens that are noisy $\boldsymbol{\mu}_1$ are refereed to as *label-relevant* tokens, and the tokens that are noisy $\boldsymbol{\mu}_2$ are referred to as *confusion* tokens. Similarly, if there are more tokens that are noisy $\boldsymbol{\mu}_2$ than those that are noisy $\boldsymbol{\mu}_1$, the former are label-relevant tokens, the latter are confusion tokens, and $y^n = -1$. All other tokens that are not label-relevant are called label-irrelevant tokens.

Let $\alpha_*$ and $\alpha_\#$ as the average fraction of the label-relevant and the confusion tokens over the distribution $\mathcal{D}$, respectively. We consider a balanced dataset. Let $\mathcal{D}_+ = \{(\boldsymbol{X}^n, y^n)|y^n = +1, n \in [N]\}$ and $\mathcal{D}_- = \{(\boldsymbol{X}^n, y^n)|y^n = -1, n \in [N]\}$ denote the sets of positive and negative labels, respectively. Then $\left|\,|\mathcal{D}^+| - |\mathcal{D}^-|\,\right| = O(\sqrt{N})$.

Our model is motivated by and generalized from those used in the state-of-art analysis of neural networks on structured data (Li & Liang, 2018; Brutzkus & Globerson, 2021; Karp et al., 2021). All the existing models require that only one discriminative pattern exists in each sample, i.e., either $\boldsymbol{\mu}_1$ or $\boldsymbol{\mu}_2$, but not both, while our model allows both patterns to appear in the same sample.

### 3.3 FORMAL THEORETICAL RESULTS

Before presenting our main theory below, we first characterize the behavior of the initial model through Assumption 1. Some important notations are summarized in Table 1.

Table 1: Some important notations

| $\sigma$ | Initialization error for value vectors | $\delta$ | Initialization error for query and key vectors |
|---|---|---|---|
| $\kappa$ | Minimum of $\|\boldsymbol{\mu}_i - \boldsymbol{\mu}_j\|$ for any $i, j \in [M]$, $i \neq j$. | $\tau$ | Token noise level |
| $M$ | Total number of patterns | $m$ | The number of neurons in $\boldsymbol{W}_O$ |
| $\alpha_*$ | Average fraction of label-relevant tokens | $\alpha_\#$ | Average fraction of confusion tokens |

**Assumption 1.** *Assume* $\max(\|\boldsymbol{W}_V^{(0)}\|, \|\boldsymbol{W}_K^{(0)}\|, \|\boldsymbol{W}_Q^{(0)}\|) \leq 1$ *without loss of generality. There exist three (not necessarily different) sets of orthonormal bases* $\mathcal{P} = \{\boldsymbol{p}_1, \boldsymbol{p}_2, \cdots, \boldsymbol{p}_M\}$, $\mathcal{Q} = \{\boldsymbol{q}_1, \boldsymbol{q}_2, \cdots, \boldsymbol{q}_M\}$, *and* $\mathcal{R} = \{\boldsymbol{r}_1, \boldsymbol{r}_2, \cdots, \boldsymbol{r}_M\}$, *where* $\boldsymbol{p}_l \in \mathbb{R}^{m_a}$, $\boldsymbol{q}_l$, $\boldsymbol{r}_l \in \mathbb{R}^{m_b}$, $\forall l \in [M]$, $\boldsymbol{q}_1 = \boldsymbol{r}_1$, *and* $\boldsymbol{q}_2 = \boldsymbol{r}_2$[3] *such that*

$$\|\boldsymbol{W}_V^{(0)} \boldsymbol{\mu}_j - \boldsymbol{p}_j\| \leq \sigma, \quad \|\boldsymbol{W}_K^{(0)} \boldsymbol{\mu}_j - \boldsymbol{q}_j\| \leq \delta, \quad and \quad \|\boldsymbol{W}_Q^{(0)} \boldsymbol{\mu}_j - \boldsymbol{r}_j\| \leq \delta. \tag{6}$$

*hold for some* $\sigma = O(1/M)$ *and* $\delta < 1/2$.

Assumption 1 characterizes the distance of query, key, and value vectors of patterns $\{\boldsymbol{\mu}_j\}_{j=1}^M$ to orthonormal vectors. The requirement on $\delta$ is minor because $\delta$ can be in the same order as $\|\boldsymbol{\mu}_j\|$.

**Theorem 1** (Generalization of ViT). *Suppose Assumption 1 holds;* $\tau \leq \min(\sigma, \delta)$; *a sufficiently large model with*

$$m \gtrsim \epsilon^{-2} M^2 \log N \text{ for } \epsilon > 0, \tag{7}$$

*the average fraction of label-relevant patterns satisfies*

$$\alpha_* \geq \frac{\alpha_\#}{\epsilon_{\mathcal{S}} e^{-(\delta+\tau)}(1 - (\sigma + \tau))}, \tag{8}$$

---

[3]The condition $\boldsymbol{q}_1 = \boldsymbol{r}_1$ and $\boldsymbol{q}_2 = \boldsymbol{r}_2$ is to eliminate the trivial case that the initial attention value is very small. This condition can be relaxed but we keep this form to simplify the representation.

for some constant $\epsilon_{\mathcal{S}} \in (0, \frac{1}{2})$; and the mini-batch size and the number of sampled tokens of each data $\boldsymbol{X}^n$, $n \in [N]$ satisfy

$$B \geq \Omega((\alpha_* - e^{-(\delta+\tau)}(\tau + \sigma))^{-2}), \quad |\mathcal{S}^n| \geq \Omega(1) \tag{9}$$

Then as long as the number of training samples $N$ satisfies

$$N \geq \Omega\left(\frac{1}{(\alpha_* - c'(1 - \zeta) - c''(\sigma + \tau))^2}\right) \tag{10}$$

for some constant $c', c'' > 0$, and $\zeta \gtrsim 1 - \eta^{10}$, after $T$ number of iterations such that

$$T = \Theta\left(\frac{1}{(1 - \epsilon - \frac{(\sigma+\tau)M}{\pi})\eta\alpha_*}\right) \tag{11}$$

with a probability at least $0.99$, the returned model achieves zero generalization error as

$$f(\boldsymbol{A}^{(0)}, \boldsymbol{W}_O^{(T)}, \boldsymbol{W}_V^{(T)}, \boldsymbol{W}_K^{(T)}, \boldsymbol{W}_Q^{(T)}) = 0 \tag{12}$$

Theorem 1 characterizes under what condition of the data the neural network with self-attention in (1) trained with Algorithm 1 can achieve zero generalization error. To show that the self-attention layer can improve the generalization performance by reducing the required sample complexity to achieve zero generalization error, we also quantify the sample complexity when there is no self-attention layer in the following proposition.

**Proposition 1** (Generalization without self-attention). *Suppose assumptions in Theorem 1 hold. When there is no self-attention layer, i.e., $\boldsymbol{W}_K$ and $\boldsymbol{W}_Q$ are not updated during the training, if $N$ satisfies*

$$N \geq \Omega\left(\frac{1}{(\alpha_*(\alpha_* - \sigma - \tau))^2}\right) \tag{13}$$

*then after $T$ iterations with $T$ in (11), the returned model achieves zero generalization error as*

$$f(\boldsymbol{A}^{(0)}, \boldsymbol{W}_O^{(T)}, \boldsymbol{W}_V^{(T)}, \boldsymbol{W}_K^{(0)}, \boldsymbol{W}_Q^{(0)}) = 0 \tag{14}$$

**Remark 1.** *(Advantage of the self-attention layer) Because $m \gg m_a, m_b, d$, the number of trainable parameter remains almost the same with or without updating the attention layer. Combining Theorem 1 and Proposition 1, we can see that with the additional self-attention layer, the sample complexity[4] is reduced by a factor $1/\alpha_*^2$ with an approximately equal number of network parameters.*

**Remark 2.** *(Generalization improvement by token sparsification). (10) and (11) show that the sample complexity $N$ and the required number of iterations $T$ scale with $1/\alpha_*^2$ and $1/\alpha_*$, respectively. Then, increasing $\alpha_*$, the fraction of label-relevant tokens, can reduce the sample complexity and speed up the convergence. Similarly, $N$ and $T$ scale with $1/(\Theta(1) - \tau)^2$ and $1/(\Theta(1) - \tau)$. Then decreasing $\tau$, the noise in the tokens, can also improve the generalization. Note that a properly designed token sparsification method can both increase $\alpha_*$ by removing label-irrelevant tokens and decrease $\tau$ by removing noisy tokens, thus improving the generalization performance.*

**Remark 3.** *(Impact of the initial model) The initial model $\boldsymbol{W}_V^{(0)}, \boldsymbol{W}_K^{(0)}, \boldsymbol{W}_Q^{(0)}$ affects the learning performance through $\sigma$ and $\delta$, both of which decrease as the initial model is improved. Then from (10) and (11), the sample complexity reduces and the convergence speeds up for a better initial model.*

Proposition 2 shows that the attention weights are increasingly concentrated on label-relevant tokens during the training. Proposition 2 is a critical component in proving Theorem 1 and is of independent interest.

**Proposition 2.** *The attention weights for each token become increasingly concentrated on those correlated with tokens of the label-relevant pattern during the training, i.e.,*

$$\sum_{i \in \mathcal{S}_*^n} softmax(\boldsymbol{X}^{n\top} \boldsymbol{W}_K^{(t)\top} \boldsymbol{W}_Q^{(t)} \boldsymbol{x}_l^n)_i = \sum_{i \in \mathcal{S}_*^n} \frac{\exp(\boldsymbol{x}_i^{n\top} \boldsymbol{W}_K^{(t)\top} \boldsymbol{W}_Q^{(t)} \boldsymbol{x}_l^n)}{\sum_{r \in \mathcal{S}^n} \exp(\boldsymbol{x}_r^{n\top} \boldsymbol{W}_K^{(t)\top} \boldsymbol{W}_Q^{(t)} \boldsymbol{x}_l^n)} \to 1 - \eta^C \tag{15}$$

*at a sublinear rate of $O(1/t)$ when $t$ is large for a large $C > 0$ and all $l \in \mathcal{S}^n$ and $n \in [N]$.*

---

[4]The sample complexity bounds in (10) and (13) are sufficient but not necessary. Thus, rigorously speaking, one can not compare two cases based on sufficient conditions only. In our analysis, however, these two bounds are derived with exactly the same technique with the only difference in handling the self-attention layer. Therefore, we believe it is fair to compare these two bounds to show the advantage of ViT.

Proposition 2 indicates that only label-relevant tokens are highlighted by the learned attention of ViTs, while other tokens have less weight. This provides a theoretical justification of magnitude-based token sparsification methods. softmax$(\cdot)_i$ in (15) denotes the $i$-th entry of softmax$(\cdot)$.

**Proof idea sketch**: The main proof idea is to show that the SGD updates scale up value, query, and key vectors of discriminative patterns, while keeping the magnitude of the projections of non-discriminative patterns and the initial model error almost unchanged. To be more specific, by Lemma 3, 4, we can identify two groups of neurons in the hidden layer $W_O$, where one group only learns the positive pattern, and the other group only learns the negative pattern. Claim 1 of Lemma 2 states that during the SGD updates, the neuron weights in these two groups evolve in the direction of projected discriminative patterns, $p_1$ and $p_2$, respectively. Meanwhile, Claim 2 of Lemma 2 indicates that $W_K$ and $W_Q$ update in the direction of increasing the magnitude of the query and key vectors of label-relevant tokens from 1 to $\Theta(\log T)$, such that the attention weights correlated with label-relevant tokens gradually become dominant. Moreover, by Claim 3 of Lemma 2, the update of $W_V$ increases the magnitude of the value vectors of label-relevant tokens, by adding partial neuron weights of $W_O$ that are aligned with the value vectors to these vectors. Due to the above properties during the training, one can simplify the training process to show that the output of neural network (1) changes linearly in the iteration number $t$. From the above analysis, we can develop the sample complexity and the required number of iterations for the zero generalization guarantee.

**Technical novelty**: Our proof technique is inspired by the feature learning technique in analyzing fully connect networks and convolution neural networks (Shi et al., 2021; Brutzkus & Globerson, 2021). Our paper makes new technical contributions from the following aspects. First, we provide a new framework of studying the nonconvex interactions of multiple weight matrices in a three-layer ViT while other feature learning works (Shi et al., 2021; Brutzkus & Globerson, 2021; Karp et al., 2021; Allen-Zhu & Li, 2022; Wen & Li, 2021; Zhang et al., 2023) only study one trainable weight matrix in the hidden layer of a two-layer network. Second, we analyze the updates of the self-attention module with the softmax function during the training, while other papers either ignore this issue without exploring convergence analysis (Edelman et al., 2022) or oversimplify the analysis by applying the neural-tangent-kernel (NTK) method that considers impractical over-parameterization and updates the weights only around initialization. (Hron et al., 2020; Yang, 2020; Allen-Zhu et al., 2019a; Arora et al., 2019). Third, we consider a more general data model, where discriminative patterns of multiple classes can exist in the same data sample, but the data models in (Brutzkus & Globerson, 2021; Karp et al., 2021) require one discriminative pattern only in each sample.

## 4 NUMERICAL EXPERIMENTS

### 4.1 EXPERIMENTS ON SYNTHETIC DATASETS

We first verify the theoretical bounds in Theorem 1 on synthetic data. We set the dimension of data and attention embeddings to be $d = m_a = m_b = 10$. Let $c_0 = 0.01$. Let the total number of patterns $M = 5$, and $\{\boldsymbol{\mu}_1, \boldsymbol{\mu}_2, \cdots, \boldsymbol{\mu}_M\}$ be a set of orthonormal bases. To satisfy Assumption 1, we generate every token that is a noisy version of $\boldsymbol{\mu}_i$ from a Gaussian distribution $\mathcal{N}(\boldsymbol{\mu}_i, c_0^2 \cdot I)$ with the mean $\boldsymbol{\mu}_i$ and covariance $c_0^2 I$, where $I \in \mathbb{R}^d$ is the identity matrix. $W_Q^{(0)} = W_Q^{(0)} = \delta^2 I / c_0^2$, $W_V^{(0)} = \sigma^2 U / c_0^2$, and each entry of $W_O^{(0)}$ follows $\mathcal{N}(0, \xi^2)$, where $U$ is an $m_a \times m_a$ orthonormal matrix, and $\xi = 0.01$. The number of neurons $m$ of $W_O$ is 1000. We set the ratio of different patterns the same among all the data for simplicity.

**Sample complexity and convergence rate**: We first study the impact of the fraction of the label-relevant patterns $\alpha_*$ on the sample complexity. Let the number of tokens after sparsification be $|\mathcal{S}^n| = 100$, the initialization error $\sigma = 0.1$, and $\delta = 0.2$. The fraction of non-discriminative patterns is fixed to be 0.5. We implement 20 independent experiments with the same $\alpha_*$ and $N$ and record the Hinge loss values of the testing data. An experiment is successful if the testing loss is smaller than $10^{-3}$. Figure 1 (a) shows the success rate of these experiments. A black block means that all the trials fail. A white block means that they all succeed. The sample complexity is indeed almost linear in $\alpha_*^{-2}$, as predicted in 10. We next explore the impact on $\sigma$. Set $\alpha_* = 0.3$ and $\alpha_\# = 0.2$. The number of tokens after sparsification is fixed at 50 for all the data. Figure 1 (b) shows that $1/\sqrt{N}$ is linear in $\Theta(1) - \sigma$, matching our theoretical prediction in (10). The result on

the noise level $\tau$ is similar to Figure 1 (b), and we skip it here. In Figure 2, we verify the number of iterations $T$ against $\alpha_*^{-1}$ in (11) where we set $\sigma = 0.1$ and $\delta = 0.4$.

**Advantage of self-attention**: To verify Proposition 1, we compare the performance on ViT in 1 and on the same network with $\boldsymbol{W}_K$ and $\boldsymbol{W}_Q$ fixed during the training, i.e., a three-layer CNN. Compared with ViT, the number of trainable parameters in CNN is reduced by only $1\%$. Figure 3 shows the sample complexity of CNN is almost linear in $\alpha_*^{-4}$ as predicted in (13). Compared with Figure 2 (a), the sample complexity significantly increases for small $\alpha_*$, indicating a much worse generalization of CNN.

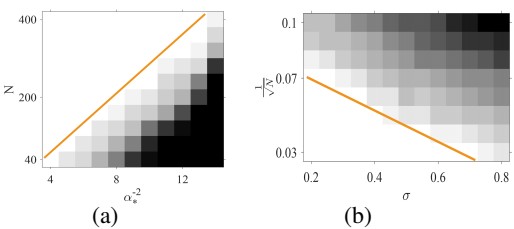
(a)          (b)

Figure 1: The impact of $\alpha_*$ and $\sigma$ on sample complexity.

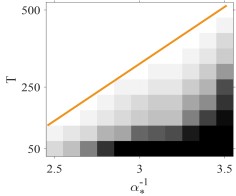

Figure 2: The number of iterations against $\alpha_*^{-1}$.

**Attention map**: We then evaluate the evolution of the attention map during the training. Let $|\mathcal{S}^n| = 50$ for all $n \in [N]$. The number of training samples is $N = 200$. $\sigma = 0.1$, $\delta = 0.2$, $\alpha_* = 0.5$, $\alpha_\# = 0.05$. In Figure 4, the red line with asterisks shows that the sum of attention weights on label-relevant tokens, i.e., the left side of (15) averaged over all $l$, indeed increases to be close to 1 when the number of iterations increases. Correspondingly, the sum of attention weights on other tokens decreases to be close to 0, as shown in the blue line with squares. This verifies Lemma 2 on a sparse attention map.

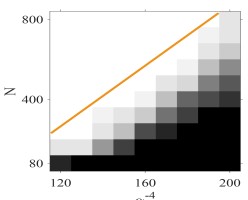
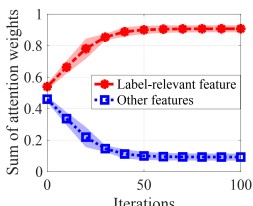
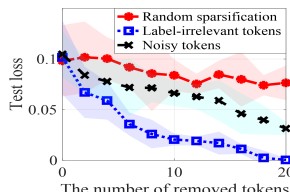

Figure 3: Comparison of ViT and CNN

Figure 4: Concentration of attention weights

Figure 5: Impact of token sparsification on testing loss

**Token sparsification**: We verify the improvement by token sparsification in Figure 5. The experiment is duplicated 20 times. The number of training samples $N = 80$. Let $|\mathcal{S}^n| = 50$ for all $n \in [N]$. Set $\sigma = 0.1$, $\delta = 0.5$, $\alpha_* = 0.6$, $\alpha_\# = 0.05$. If we apply random sampling over all tokens, the performance cannot be improved as shown in the red curve because $\alpha_*$ and $\sigma$ do not change. If we remove either label-irrelevant tokens or tokens with significant noise, the testing loss decreases, as indicated in the blue and black curves. This justifies our insight **P3** on token sparsification.

### 4.2 EXPERIMENTS ON IMAGE CLASSIFICATION DATASETS

**Dataset**: To characterize the effect of label-relevant and label-irrelevant tokens on generalization, following the setup of image integration in (Karp et al., 2021), we adopt an image from CIFAR-10 dataset (Krizhevsky et al., 2010) as the label-relevant image pattern and integrate it with a noisy background image from the IMAGENET Plants synset (Karp et al., 2021; Deng et al., 2009), which plays the role of label-irrelevant feature. Specifically, we randomly cut out a region with size $26 \times 26$ in the IMAGENET image and replace it with a resized CIFAR-10 image.

**Architecture**: Experiments are implemented on a deep ViT model. Following (Dosovitskiy et al., 2020), the network architecture contains 5 blocks, where we have a 4-head self-attention layer and a one-layer perceptron with skip connections and Layer-Normalization in each block.

We first evaluate the impact on generalization of token sparsification that removes label-irrelevant patterns to increase $\alpha_*$. We consider a ten-classification problem where in both the training and testing datasets, the images used for integration are randomly selected from CIFAR-10 and IMA-GENET. The number of samples for training and testing is $50K$ and $10K$, respectively. A pre-trained model from CIFAR-100 (Krizhevsky et al., 2010) is used as the initial model with the output layer randomly initialized. Without token sparsification, the fraction of class-relevant tokens is $\alpha_* \approx 0.66$. $\alpha_* = 1$ implies all background tokens are removed. Figure 6 (a) indicates that a larger $\alpha_*$ by removing more label-irrelevant tokens leads to higher test accuracy. Moreover, the test performance improves with more training samples. These are consistent with our sample complexity analysis in (10). Figure 6 (b) presents the required sample complexity to learn a model with desirable test accuracy. We run 10 independent experiments for each pair of $\alpha_*$ and $N$, and the experiment is considered a success if the learned model achieves a test accuracy of at least $77.5\%$.

We then evaluate the impact of token sparsification on removing spurious correlations (Sagawa et al., 2020), as well as the impact of the initial model. We consider a binary classification problem that differentiates "bird" and "airplane" images. To introduce spurious correlations in the training data, $90\%$ of bird images in the training data are integrated into the IMAGENET plant background, while only $10\%$ of airplane images have the plant background. The remaining training data are integrated into a clean background by zero padding. Therefore, the label "bird" is spuriously correlated with the class-irrelevant plant background. The testing data contain $50\%$ birds and $50\%$ airplanes, and each class has $50\%$ plant background and $50\%$ clean background. The numbers of training and testing samples are $10K$ and $2K$, respectively. We initialize the ViT using two pre-trained models. The first one is pre-trained with CIFAR-100, which contains images of 100 classes not including birds and airplanes. The other initial model is trained with a modified CIFAR-10 with $500$ images per class for a total of eight classes, excluding birds and airplanes. The pre-trained model on CIFAR-100 is a better initial model because it is trained on a more diverse dataset with more samples.

In Figure 6 (c), the token sparsification method removes the tokens of the added background, and the corresponding $\alpha_*$ increases. Note that removing background in the training dataset also reduces the spurious correlations between birds and plants. Figure 6 (c) shows that from both initial models, the testing accuracy increases when more background tokens are removed. Moreover, a better initial model leads to a better testing performance. This is consistent with Remarks 2 and 3.

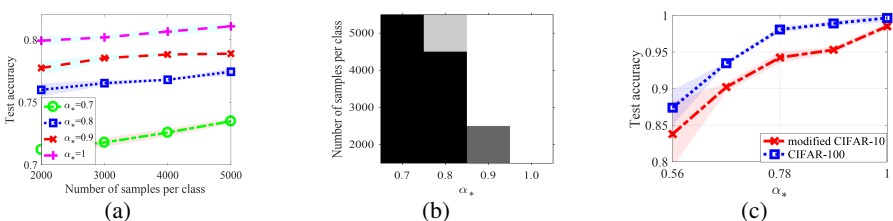

Figure 6: (a) Test accuracy when $N$ and $\alpha_*$ change. (b) Relationship of sample complexity against $\alpha^*$. (c) Test accuracy when token sparsification removes spurious correlations.

## 5 CONCLUSION

This paper provides a novel theoretical generalization analysis of three-layer ViTs. Focusing on a data model with label-relevant and label-irrelevant tokens, this paper explicitly quantifies the sample complexity as a function of the fraction of label-relevant tokens and the token noise projected by the initial model. It proves that the learned attention map becomes increasingly sparse during the training, where the attention weights are concentrated on those of label-relevant tokens. Our theoretical results also offer a guideline on designing proper token sparsification methods to improve the test performance.

This paper considers a simplified but representative Transformer architecture to theoretically examine the role of self-attention layer as the first step. One future direction is to analyze more practical architectures such as those with skip connection, local attention layers, and Transformers in other areas. We see no ethical or immediate negative societal consequence of our work.

ACKNOWLEDGMENTS

This work was supported by AFOSR FA9550-20-1-0122, ARO W911NF-21-1-0255, NSF 1932196 and the Rensselaer-IBM AI Research Collaboration (http://airc.rpi.edu), part of the IBM AI Horizons Network (http://ibm.biz/AIHorizons). We thank Dr. Shuai Zhang at Rensselaer Polytechnic Institute for useful discussions. We thank all anonymous reviewers for their constructive comments.

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

**The appendix contains 6 sections. We first provides a brief discussion about comparisons between our works and other two related works in Section A. In Section B, we add additional experiments for the verification of our theory. In Section C, we introduce some definitions and assumptions in accordance with the main paper for the ease of the proof in the following. Section D first states a core lemma for the proof, based on which we provide the proof of Theorem 1 and Proposition 1 and 2. Section E gives the proof of Lemma 2 with three subsections to prove its three main claims. Section F shows key lemmas and the proof of lemmas for this paper. We finally discuss the extension of our analysis in Section G, including extension to multi-classification cases, general data model cases, multi-head attention cases, and cases with skip connections in Section G.1, G.2, G.3, and G.4, respectively.**

## A    COMPARISON WITH TWO RELATED WORKS

### A.1    COMPARISON WITH (ALLEN-ZHU & LI, 2023)

Allen-Zhu & Li (2023) studies ensemble learning and knowledge distillation. Its main proof idea is that given large amounts of multi-view data, each single model learns one feature, and then ensemble learning integrates all learned features and, thus, improves over single models. Knowledge distillation applies softmax logits to make use of information learned from the ensemble model. It is analyzed in a similar approach to studying single models. The single models considered in (Allen-Zhu & Li, 2023) is a two-layer Relu network.

In contrast, in this paper, we consider a two-layer Relu network with an additional self-attention layer. The network architecture and training algorithm for the self-attention layer is completely different from those for the softmax logit in the knowledge distillation function. In our proof, we analyze the impact of the gradient of $\boldsymbol{W}_Q$, $\boldsymbol{W}_K$, and $\boldsymbol{W}_V$ on different patterns (Claims 2 and 3 of Lemma 2), showing that the training process helps to enlarge the magnitude of label-relevant features. We also show that neurons in $\boldsymbol{W}_O$ mainly learn from discriminative patterns (Claim 1 of Lemma 2). Such a learning process is affected by the error in the initial model and the noise in tokens. Please see details in "Proof idea sketch" and "Technical novelty" in Section 3.3 on Page 7 of the paper. This technique we develop plays a critical role in our analysis of self-attention layers. This technique is novel and did not appear in any existing works.

### A.2    COMPARISON WITH (JELASSI ET AL., 2022)

Jelassi et al. (2022) is a concurrent work which theoretically studies Vision Transformers. The major difference between (Jelassi et al., 2022) and our work is that we consider different data models and network architectures. In (Jelassi et al., 2022), the data model requires spatial association between tokens. The attention map is replaced with position encoding, and the training process of the attention map is simplified to train a linear layer. Our setup models the data mainly based on the category of patterns. We keep the classical structure and training process of self-attention, where $\boldsymbol{W}_Q$, $\boldsymbol{W}_K$, and $\boldsymbol{W}_V$ are trained separately. The required number of samples and iterations are derived as functions of the fraction of label-relevant patterns. In addition, the non-linear activation function they consider is polynomial activation, instead of Relu or Gelu as in practice. Based on these conditions, they are able to study a different and mroe general labelling function.

## B    MORE EXPERIMENTS

Following a similar setup in Section 4, we add more experiments.
For experiments on synthetic data, we set the dimension of data and attention embeddings to be $d = m_a = m_b = 20$. We vary the number of patterns $M$ to be 10, 15, and 20. Data generation and the network architecture follow the setup in Section 4. One can observe the same trend in Figure 7 and 8 as in Figure 4 and 5, respectively, indicating that our conclusion that the attention map becomes sparse during the training, and that pruning label-irrelevant tokens or noisy tokens improves the performance, both hold for different choices of $M$.

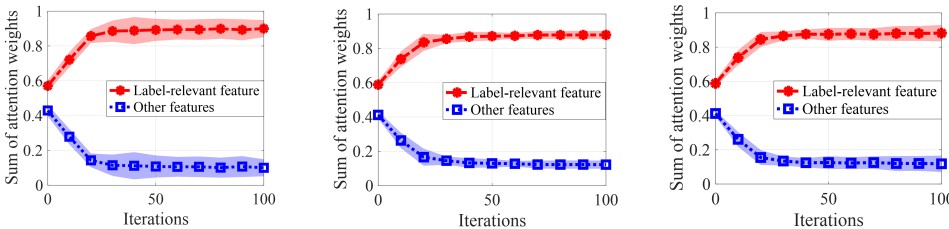

Figure 7: Concentration of attention weights when (a) $M = 10$ (b) $M = 15$ (c) $M = 20$.

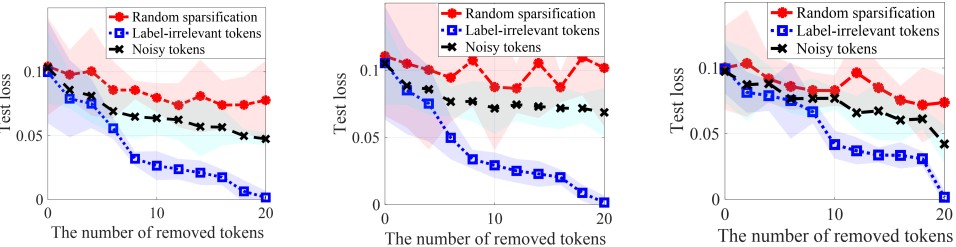

Figure 8: Impact of token sparsification on testing data when (a) $M = 10$ (b) $M = 15$ (c) $M = 20$.

## C  PRELIMINARIES

We first formally restate the neural network with different notations of loss functions, and the Algorithm 1 of the training steps after token sparsification. The notations used in the Appendix is summarized in Table 2.

Table 2: Summary of notations

| | |
|---|---|
| $F(\boldsymbol{X}^n), \text{Loss}(\boldsymbol{X}^n, y^n)$ | The network output for $\boldsymbol{X}^n$ and the loss function of a single data. |
| $\overline{\text{Loss}}_b, \overline{\text{Loss}}, \text{Loss}$ | The loss function of a mini-batch, the empirical loss, and the population loss, respectively. |
| $\boldsymbol{p}_j(t), \boldsymbol{q}_j(t), \boldsymbol{r}_j(t)$ | The features in value, key, and query vectors at the iteration $t$ for pattern $j$, respectively. We have $\boldsymbol{p}_j(0) = \boldsymbol{p}_j$, $\boldsymbol{q}_j(0) = \boldsymbol{q}_j$, and $\boldsymbol{r}_j(0) = \boldsymbol{r}_j$. |
| $\boldsymbol{z}_j^n(t), \boldsymbol{n}_j^n(t), \boldsymbol{o}_j^n(t)$ | The error terms in the value, key, and query vectors of the $j$-th token and $n$-th data compared to their features at iteration $t$. |
| $\mathcal{W}(t), \mathcal{U}(t)$ | The set of lucky neurons at $t$-th iterations. |
| $\phi_n(t), \nu_n(t), p_n(t), \lambda$ | The bounds of value of some attention weights at iteration $t$. $\lambda$ is the threshold between inner products of tokens from the same pattern and different patterns. |
| $\mathcal{S}_j^n, \mathcal{S}_*^n, \mathcal{S}_\#^n$ | $\mathcal{S}_j^n$ is the set of sampled tokens of pattern $j$ for the $n$-th data. $\mathcal{S}_*^n, \mathcal{S}_\#^n$ are sets of sampled tokens of the label-relevant pattern and the confusion pattern for the $n$-th data, respectively. |
| $\alpha_*, \alpha_\#, \alpha_{nd}$ | The mean of fraction of label-relevant tokens, confusion tokens, and non-discriminative tokens, respectively. |

For the network[5]

$$F(\boldsymbol{X}^n) = \frac{1}{|\mathcal{S}^n|} \sum_{l \in \mathcal{S}^n} \boldsymbol{a}_{(l)}^\top \text{Relu}(\boldsymbol{W}_O \boldsymbol{W}_V \boldsymbol{X}^n \text{softmax}(\boldsymbol{X}^{n\top} \boldsymbol{W}_K^\top \boldsymbol{W}_Q \boldsymbol{x}_l^n)) \tag{16}$$

The loss function of a single data, a mini-batch, the empirical loss, and the population loss is defined in the following.

$$\text{Loss}(\boldsymbol{X}^n, y^n) = \max\{1 - y^n \cdot F(\boldsymbol{X}^n), 0\} \tag{17}$$

---

[5]Note that in our proof in the Appendix, we often use the notation $\text{softmax}(\boldsymbol{x}_i^{n\top} \boldsymbol{W}_K^{(t)\top} \boldsymbol{W}_Q^{(t)} \boldsymbol{x}_l^n)$, which is the same meaning as $\text{softmax}(\boldsymbol{X}^{n\top} \boldsymbol{W}_K^{(t)\top} \boldsymbol{W}_Q^{(t)} \boldsymbol{x}_l^n)_i$.

$$\overline{\text{Loss}}_b = \frac{1}{B} \sum_{n \in \mathcal{B}_b} \text{Loss}(\boldsymbol{X}^n, y^n) \tag{18}$$

$$\overline{\text{Loss}} = \frac{1}{N} \sum_{n=1}^{N} \text{Loss}(\boldsymbol{X}^n, y^n) \tag{19}$$

$$\text{Loss} = \mathbb{E}_{(\boldsymbol{X}, y) \sim \mathcal{D}} [\overline{\text{Loss}}] \tag{20}$$

The formal algorithm is as follows. We assume that each entry of $\boldsymbol{W}_O^{(0)}$ is randomly initialized from $\mathcal{N}(0, \xi^2)$ where $\xi = \frac{1}{\sqrt{M}}$. Define that $a_{(l)_i}^{(0)}$, $i \in [m]$, $l \in [L]$ is uniformly initialized from $+\{\frac{1}{\sqrt{m}}, -\frac{1}{\sqrt{m}}\}$ and fixed during the training. $\boldsymbol{W}_V$, $\boldsymbol{W}_K$, and $\boldsymbol{W}_Q$ are initialized from a good pretrained model.

---

**Algorithm 1** Training with SGD

---

1: **Input:** Training data $\{(\boldsymbol{X}^n, y^n)\}_{n=1}^N$, the step size $\eta$, the total number of iterations $T$, batch size $B$.

2: **Initialization:** Every entry of $\boldsymbol{W}_O^{(0)}$ from $\mathcal{N}(0, \xi^2)$, and every entry of $\boldsymbol{a}_{(l)}^{(0)}$ from Uniform($\{+\frac{1}{\sqrt{m}}, -\frac{1}{\sqrt{m}}\}$). $\boldsymbol{W}_V^{(0)}$, $\boldsymbol{W}_K^{(0)}$ and $\boldsymbol{W}_Q^{(0)}$ from a pre-trained model.

3: **Stochastic Gradient Descent:** for $t = 0, 1, \cdots, T-1$ and $\boldsymbol{W}^{(t)} \in \{\boldsymbol{W}_O^{(t)}, \boldsymbol{W}_V^{(t)}, \boldsymbol{W}_K^{(t)}, \boldsymbol{W}_Q^{(t)}\}$

$$\boldsymbol{W}^{(t+1)} = \boldsymbol{W}^{(t)} - \eta \cdot \frac{1}{B} \sum_{n \in \mathcal{B}_t} \nabla_{\boldsymbol{W}^{(t)}} \ell(\boldsymbol{X}^n, y^n; \boldsymbol{W}_O^{(t)}, \boldsymbol{W}_V^{(t)}, \boldsymbol{W}_K^{(t)}, \boldsymbol{W}_Q^{(t)}) \tag{21}$$

4: **Output:** $\boldsymbol{W}_O^{(T)}$, $\boldsymbol{W}_V^{(T)}$, $\boldsymbol{W}_K^{(T)}$, $\boldsymbol{W}_Q^{(T)}$.

---

Assumption 1 can be interpreted as that we initialize $\boldsymbol{W}_V$, $\boldsymbol{W}_K$, and $\boldsymbol{W}_Q$ to be the matrices that can map tokens to orthogonal features with added error terms.

**Assumption 2.** *Define* $\boldsymbol{P} = (\boldsymbol{p}_1, \boldsymbol{p}_2, \cdots, \boldsymbol{p}_M) \in \mathbb{R}^{m_a \times M}$, $\boldsymbol{Q} = (\boldsymbol{q}_1, \boldsymbol{q}_2, \cdots, \boldsymbol{q}_M) \in \mathbb{R}^{m_b \times M}$ *and* $\boldsymbol{R} = (\boldsymbol{r}_1, \boldsymbol{r}_2, \cdots, \boldsymbol{r}_M) \in \mathbb{R}^{m_b \times M}$ *as three feature matrices, where* $\mathcal{P} = \{\boldsymbol{p}_1, \boldsymbol{p}_2, \cdots, \boldsymbol{p}_M\}$, $\mathcal{Q} = \{\boldsymbol{q}_1, \boldsymbol{q}_2, \cdots, \boldsymbol{q}_M\}$ *and* $\mathcal{R} = \{\boldsymbol{r}_1, \boldsymbol{r}_2, \cdots, \boldsymbol{r}_M\}$ *are three sets of orthonormal bases. Define the noise terms* $\boldsymbol{z}_j^n(t)$, $\boldsymbol{n}_j^n(t)$ *and* $\boldsymbol{o}_j^n(t)$ *with* $\|\boldsymbol{z}_j^n(0)\| \leq \sigma + \tau$ *and* $\|\boldsymbol{n}_j^n(0)\|, \|\boldsymbol{o}_j^n(0)\| \leq \delta + \tau$ *for* $j \in [L]$. $\boldsymbol{q}_1 = \boldsymbol{r}_1$, $\boldsymbol{q}_2 = \boldsymbol{r}_2$. *Suppose* $\|\boldsymbol{W}_V^{(0)}\|, \|\boldsymbol{W}_K^{(0)}\|, \|\boldsymbol{W}_Q^{(0)}\| \leq 1$, $\sigma, \tau < O(1/M)$ *and* $\delta < 1/2$. *Then, for* $\boldsymbol{x}_l^n \in \mathcal{S}_j^n$

1. $\boldsymbol{W}_V^{(0)} \boldsymbol{x}_l^n = \boldsymbol{p}_j + \boldsymbol{z}_j^n(0)$.

2. $\boldsymbol{W}_K^{(0)} \boldsymbol{x}_l^n = \boldsymbol{q}_j + \boldsymbol{n}_j^n(0)$.

3. $\boldsymbol{W}_Q^{(0)} \boldsymbol{x}_l^n = \boldsymbol{r}_j + \boldsymbol{o}_j^n(0)$.

Assumption 2 is a straightforward combination of Assumption 1 and (5) by applying the triangle inequality to bound the error terms for tokens.

**Definition 1.**      1. $\phi_n(t) = \frac{1}{|\mathcal{S}_1^n| e^{\|\boldsymbol{q}_1(t)\|^2 + (\delta+\tau)\|\boldsymbol{q}_1(t)\|} + |\mathcal{S}^n| - |\mathcal{S}_1^n|}$.

2. $\nu_n(t) = \frac{1}{|\mathcal{S}_1^n| e^{\|\boldsymbol{q}_1(t)\|^2 - (\delta+\tau)\|\boldsymbol{q}_1(t)\|} + |\mathcal{S}^n| - |\mathcal{S}_1^n|}$.

3. $p_n(t) = |\mathcal{S}_1^n| e^{\|\boldsymbol{q}_1(t)\|^2 - (\delta+\tau)\|\boldsymbol{q}_1(t)\|} \nu_n(t)$.

4. $\mathcal{S}_*^n = \begin{cases} \mathcal{S}_1^n, & \text{if } y^n = 1 \\ \mathcal{S}_2^n, & \text{if } y^n = -1 \end{cases}$, $\mathcal{S}_\#^n = \begin{cases} \mathcal{S}_2^n, & \text{if } y^n = 1 \\ \mathcal{S}_1^n, & \text{if } y^n = -1 \end{cases}$

5. $\alpha_* = \mathbb{E}\left[\frac{|\mathcal{S}_*^n|}{|\mathcal{S}^n|}\right]$, $\alpha_\# = \mathbb{E}\left[\frac{|\mathcal{S}_\#^n|}{|\mathcal{S}^n|}\right]$, $\alpha_{nd} = \sum_{l=3}^{M} \mathbb{E}\left[\frac{|\mathcal{S}_l^n|}{|\mathcal{S}^n|}\right]$.

**Definition 2.** *Let $\theta_1^i$ be the angle between $\boldsymbol{p}_1$ and $\boldsymbol{W}_{O_{(i,\cdot)}}$. Let $\theta_2^i$ be the angle between $\boldsymbol{p}_2$ and $\boldsymbol{W}_{O_{(i,\cdot)}}$. Define $\mathcal{W}(t), \mathcal{U}(t)$ as the sets of lucky neurons at the $t$-th iteration such that*

$$\mathcal{W}(t) = \{i : \theta_1^i \leq \sigma + \tau, i \in [m]\} \tag{22}$$

$$\mathcal{U}(t) = \{i : \theta_2^i \leq \sigma + \tau, i \in [m]\} \tag{23}$$

**Assumption 3.** *For one data $\boldsymbol{X}^n$, if the patch $i$ and $j$ correspond to the same feature $k \in [M]$, i.e., $i \in \mathcal{S}_k^n$ and $j \in \mathcal{S}_k^n$, we have*

$$\boldsymbol{x}_i^{n\top} \boldsymbol{x}_j^n \geq 1 \tag{24}$$

*If the patch $i$ and $j$ correspond to the different feature $k, l \in [M]$, $k \neq l$ i.e., $i \in \mathcal{S}_k^n$ and $j \in \mathcal{S}_l^n$, $k \neq l$, we have*

$$\boldsymbol{x}_i^{n\top} \boldsymbol{x}_j^n \leq \lambda < 1 \tag{25}$$

This assumption is equivalent to the data model by (5) since $\tau < O(1/M)$. For the simplicity of presentation, we scale up all tokens a little bit to make the threshold of linear separability be 1. We also take $1 - \lambda$ and $\lambda$ as $\Theta(1)$ for the simplicity.

**Definition 3.** *(Vershynin, 2010) We say $X$ is a sub-Gaussian random variable with sub-Gaussian norm $K > 0$, if $(\mathbb{E}|X|^p)^{\frac{1}{p}} \leq K\sqrt{p}$ for all $p \geq 1$. In addition, the sub-Gaussian norm of X, denoted $\|X\|_{\psi_2}$, is defined as $\|X\|_{\psi_2} = \sup_{p \geq 1} p^{-\frac{1}{2}}(\mathbb{E}|X|^p)^{\frac{1}{p}}$.*

**Lemma 1.** *(Vershynin (2010) Proposition 5.1, Hoeffding's inequality) Let $X_1, X_2, \cdots, X_N$ be independent centered sub-gaussian random variables, and let $K = \max_i \|\boldsymbol{X}_i\|_{\psi_2}$. Then for every $\boldsymbol{a} = (a_1, \cdots, a_N) \in \mathbb{R}^N$ and every $t \geq 0$, we have*

$$\mathbb{P}\left\{\left|\sum_{i=1}^N a_i X_i\right| \geq t\right\} \leq e \cdot \exp(-\frac{ct^2}{K^2\|\boldsymbol{a}\|^2}) \tag{26}$$

*where $c > 0$ is an absolute constant.*

# D    PROOF OF THE MAIN THEOREM AND PROPOSITIONS

We state Lemma 2 first before we introduce the proof of main theorems. Lemma 2 is the key lemma in our paper to show the training process of our ViT model using SGD. It has three major claims. Claim 1 involves the growth of $\boldsymbol{W}_O^{(t)}$ in terms of different directions of $\boldsymbol{p}_l$, $i \in [M]$. Claim 2 describes the training dynamics of $\boldsymbol{W}_Q^{(t)}$ and $\boldsymbol{W}_K^{(t)}$ separately to show the tendency to a sparse attention map. Claim 3 studies the gradient update process of $\boldsymbol{W}_V^{(t)}$.

**Lemma 2.** *For $l \in \mathcal{S}_1^n$ for the data with $y^n = 1$, define*

$$
\begin{aligned}
\boldsymbol{V}_l^n(t) =& \boldsymbol{W}_V^{(t)} \boldsymbol{X}^n \mathrm{softmax}(\boldsymbol{X}^{n\top} \boldsymbol{W}_K^{(t)\top} \boldsymbol{W}_Q^{(t)} \boldsymbol{x}_l^n) \\
=& \sum_{s \in \mathcal{S}_1} \mathrm{softmax}(\boldsymbol{x}_s^{n\top} \boldsymbol{W}_K^{(t)\top} \boldsymbol{W}_Q^{(t)} \boldsymbol{x}_l^n)\boldsymbol{p}_1 + \boldsymbol{z}(t) + \sum_{j \neq 1} W_j^n(t)\boldsymbol{p}_j \\
& - \eta \sum_{b=1}^t \left(\sum_{i \in \mathcal{W}(b)} V_i(b)\boldsymbol{W}_{O_{(i,\cdot)}}^{(b)\top} + \sum_{i \notin \mathcal{W}(b)} V_i(b)\lambda \boldsymbol{W}_{O_{(i,\cdot)}}^{(b)\top}\right)
\end{aligned}
\tag{27}
$$

*with*

$$W_l^n(t) \leq \nu_n(t)|\mathcal{S}_j^n| \tag{28}$$

$$V_i(t) \lesssim \frac{1}{2B} \sum_{n \in \mathcal{B}_{b+}} -\frac{|\mathcal{S}_1^n|}{mL}p_n(t), \quad i \in \mathcal{W}(t)] \tag{29}$$

$$V_i(t) \gtrsim \frac{1}{2B} \sum_{n \in \mathcal{B}_{b-}} \frac{|\mathcal{S}_2^n|}{mL}p_n(t), \quad i \in \mathcal{U}(t) \tag{30}$$

$$V_i(t) \geq -\frac{1}{\sqrt{B}m}, \quad \textit{if } i \textit{ is an unlucky neuron.} \tag{31}$$

*We also have the following claims:*

**Claim 1.** *For the lucky neuron $i \in \mathcal{W}(t)$ and $b \in [T]$, we have*

$$\boldsymbol{W}_{O_{(i,\cdot)}}^{(t)}\boldsymbol{p}_1 \gtrsim \frac{1}{Bt}\sum_{b=1}^{t}\sum_{n\in\mathcal{B}_b}\frac{\eta^2 tbm}{|\mathcal{S}^n|a^2}|\mathcal{S}_1^n|\|\boldsymbol{p}_1\|^2 p_n(b) + \xi(1-(\sigma+\tau)) \tag{32}$$

$$\boldsymbol{W}_{O_{(i,\cdot)}}^{(t)}\boldsymbol{p} \leq \xi\|\boldsymbol{p}\|, \quad for\ \boldsymbol{p}\in\mathcal{P}/\boldsymbol{p}_1, \tag{33}$$

$$(\frac{1}{Bt}\sum_{b=1}^{t}\sum_{n\in\mathcal{B}_b}\frac{\eta^2 tbm|\mathcal{S}_1^n|}{a^2|\mathcal{S}^n|}\|\boldsymbol{p}\|^2 p_n(b))^2 \leq \|\boldsymbol{W}_{O_{(i,\cdot)}}^{(t)}\|^2 \leq M\xi^2\|\boldsymbol{p}\|^2 + (\frac{\eta^2 t^2 m}{a^2})^2\|\boldsymbol{p}\|^2 \tag{34}$$

*and for the noise $\boldsymbol{z}_l(t)$,*

$$\|\boldsymbol{W}_{O_{(i,\cdot)}}^{(t)}\boldsymbol{z}_l(t)\| \leq ((\sigma+\tau))(\sqrt{M}\xi + \frac{\eta^2 t^2 m}{a^2})\|\boldsymbol{p}\| \tag{35}$$

*For $i \in \mathcal{U}(t)$, we also have equations as in (32) to (35), including*

$$\boldsymbol{W}_{O_{(i,\cdot)}}^{(t)}\boldsymbol{p}_2 \gtrsim \frac{1}{Bt}\sum_{b=1}^{t}\sum_{n\in\mathcal{B}_b}\frac{\eta^2 tbm|\mathcal{S}_2^n|}{|\mathcal{S}^n|a^2}\|\boldsymbol{p}_2\|^2 p_n(b) + \xi(1-(\sigma+\tau)) \tag{36}$$

$$\boldsymbol{W}_{O_{(i,\cdot)}}^{(t)}\boldsymbol{p} \leq \xi\|\boldsymbol{p}\|, \quad for\ \boldsymbol{p}\in\mathcal{P}/\boldsymbol{p}_2, \tag{37}$$

$$(\frac{1}{Bt}\sum_{b=1}^{t}\sum_{n\in\mathcal{B}_b}\frac{\eta^2 tb|\mathcal{S}_2^n|m}{a^2|\mathcal{S}^n|}\|\boldsymbol{p}\|^2 p_n(b))^2 \leq \|\boldsymbol{W}_{O_{(i,\cdot)}}^{(t)}\|^2 \leq M\xi^2\|\boldsymbol{p}\|^2 + (\frac{\eta^2 t^2 m}{a^2})^2\|\boldsymbol{p}\|^2 \tag{38}$$

*and for the noise $\boldsymbol{z}_l(t)$,*

$$\|\boldsymbol{W}_{O_{(i,\cdot)}}^{(t)}\boldsymbol{z}_l(t)\| \leq ((\sigma+\tau))(\sqrt{M}\xi + \frac{\eta^2 t^2 m}{a^2})\|\boldsymbol{p}\| \tag{39}$$

*For unlucky neurons, we have*

$$\boldsymbol{W}_{O_{(i,\cdot)}}^{(t)}\boldsymbol{p} \leq \xi\|\boldsymbol{p}\|, \quad for\ \boldsymbol{p}\in\mathcal{P}/\{\boldsymbol{p}_1,\boldsymbol{p}_2\} \tag{40}$$

$$\|\boldsymbol{W}_{O_{(i,\cdot)}}^{(t)}\boldsymbol{z}_l(t)\| \leq ((\sigma+\tau))\sqrt{M}\xi\|\boldsymbol{p}\| \tag{41}$$

$$\|\boldsymbol{W}_{O_{(i,\cdot)}}^{(t)}\|^2 \leq M\xi^2\|\boldsymbol{p}\|^2 \tag{42}$$

**Claim 2.** *Given conditions in (8), there exists $K(t), Q(t) > 0$, where $t$ is large enough before the end of training, such that for $j \in \mathcal{S}_*^n$,*

$$softmax(\boldsymbol{x}_j^{n\top}\boldsymbol{W}_K^{(t+1)}\boldsymbol{W}_Q^{(t+1)}\boldsymbol{x}_l^n) \gtrsim \frac{e^{(1+K(t))\|\boldsymbol{q}_1(t)\|^2-(\delta+\tau)\|\boldsymbol{q}_1(t)\|}}{|\mathcal{S}_1^n|e^{(1+K(t))\|\boldsymbol{q}_1(t)\|^2-(\delta+\tau)\|\boldsymbol{q}_1(t)\|} + (|\mathcal{S}^n|-|\mathcal{S}_1^n|)} \tag{43}$$

$$softmax(\boldsymbol{x}_j^{n\top}\boldsymbol{W}_K^{(t+1)\top}\boldsymbol{W}_Q^{(t+1)}\boldsymbol{x}_j^n) - softmax(\boldsymbol{x}_j^{n\top}\boldsymbol{W}_K^{(t)\top}\boldsymbol{W}_Q^{(t)}\boldsymbol{x}_l^n)$$
$$\gtrsim \frac{|\mathcal{S}^n|-|\mathcal{S}_1^n|}{(|\mathcal{S}_1^n|e^{(1+K(t))\|\boldsymbol{q}_1(t)\|^2-(\delta+\tau)\|\boldsymbol{q}_1(t)\|} + (|\mathcal{S}^n|-|\mathcal{S}_1^n|))^2}e^{\|\boldsymbol{q}_1(t)\|^2-(\delta+\tau)\|\boldsymbol{q}_1(t)\|}\cdot K(t), \tag{44}$$

*and for $j \notin \mathcal{S}_*^n$, we have*

$$softmax(\boldsymbol{x}_j^{n\top}\boldsymbol{W}_K^{(t+1)\top}\boldsymbol{W}_Q^{(t+1)}\boldsymbol{x}_l^n) \lesssim \frac{1}{|\mathcal{S}_1^n|e^{(1+K(t))\|\boldsymbol{q}_1(t)\|^2-\delta\|\boldsymbol{q}_1(t)\|} + (|\mathcal{S}^n|-|\mathcal{S}_1^n|)} \tag{45}$$

$$softmax(\boldsymbol{x}_j^{n\top}\boldsymbol{W}_K^{(t+1)}\boldsymbol{W}_Q^{(t+1)}\boldsymbol{x}_l^n) - softmax(\boldsymbol{x}_j^{n\top}\boldsymbol{W}_K^{(t)\top}\boldsymbol{W}_Q^{(t)}\boldsymbol{x}_l^n)$$
$$\lesssim -\frac{|\mathcal{S}_1^n|}{(|\mathcal{S}_1^n|e^{(1+K(t))\|\boldsymbol{q}_1(t)\|^2-\delta\|\boldsymbol{q}_1(t)\|} + (|\mathcal{S}^n|-|\mathcal{S}_1|))^2}e^{\|\boldsymbol{q}_1(t)\|^2-\delta\|\boldsymbol{q}_1(t)\|}\cdot K(t) \tag{46}$$

*For $i = 1, 2$,*

$$\boldsymbol{q}_i(t) = \sqrt{\prod_{l=0}^{t-1}(1+K(l))}\boldsymbol{q}_i \tag{47}$$

$$\boldsymbol{r}_i(t) = \sqrt{\prod_{l=0}^{t-1}(1+Q(l))}\boldsymbol{r}_i \tag{48}$$

**Claim 3.** *For the update of $\boldsymbol{W}_V^{(t)}$, there exists $\lambda \leq \Theta(1)$ such that*

$$\boldsymbol{W}_V^{(t)}\boldsymbol{x}_j^n = \boldsymbol{p}_1 - \eta \sum_{b=1}^{t}\Big(\sum_{i\in\mathcal{W}(b)} V_i(b)\boldsymbol{W}_{O_{(i,\cdot)}}^{(b)}{}^\top + \sum_{i\notin\mathcal{W}(b)}\lambda V_i(b)\boldsymbol{W}_{O_{(i,\cdot)}}^{(b)}{}^\top\Big) + \boldsymbol{z}_j(t), \quad j\in\mathcal{S}_1^n \quad (49)$$

$$\boldsymbol{W}_V^{(t)}\boldsymbol{x}_j^n = \boldsymbol{p}_1 - \eta \sum_{b=1}^{t}\Big(\sum_{i\in\mathcal{U}(b)} V_i(b)\boldsymbol{W}_{O_{(i,\cdot)}}^{(b)}{}^\top + \sum_{i\notin\mathcal{U}(b)}\lambda V_i(b)\boldsymbol{W}_{O_{(i,\cdot)}}^{(b)}{}^\top\Big) + \boldsymbol{z}_j(t), \quad j\in\mathcal{S}_2^n \quad (50)$$

$$\boldsymbol{W}_V^{(t+1)}\boldsymbol{x}_j^n = \boldsymbol{p}_1 - \eta \sum_{b=1}^{t}\sum_{i=1}^{m}\lambda V_i(b)\boldsymbol{W}_{O_{(i,\cdot)}}^{(b)}{}^\top + \boldsymbol{z}_j(t), \quad j\in[|\mathcal{S}^n|]/(\mathcal{S}_1^n\cup\mathcal{S}_2^n) \quad (51)$$

$$\|\boldsymbol{z}_j(t)\| \leq (\sigma+\tau) \quad (52)$$

To prove Theorem 1, we either show $F(\boldsymbol{X}^n) > 1$ for $y^n = 1$ or show $F(\boldsymbol{X}^n) < -1$ for $y^n = -1$. Take $y^n = 1$ as an example, the basic idea of the proof is to make use of Lemma 2 to find a lower bound as a function of $\alpha_*$, $\sigma$, $\tau$, etc.. The remaining step is to derive conditions on the sample complexity and the required number of iterations in terms of $\alpha_*$, $\sigma$, and $\tau$ such that the lower bound is greater than 1. Given a balanced dataset, these conditions also ensure that $F(\boldsymbol{X}^n) < -1$ for $y^n = -1$. During the proof, we may need to use some of equations as intermediate steps in the proof of Lemma 2. Since that these equations are not concise for presentation, we prefer not to state them formally in Lemma 2, but still refer to them as useful conclusions. The following is the details of the proof.

**Proof of Theorem 1:**
For $y^n = 1$, define $\mathcal{K}_+^l = \{i \in [m] : \boldsymbol{W}_{O_{(i,\cdot)}}^{(t)}\boldsymbol{V}_l^n(t) \geq 0\}$ and $\mathcal{K}_-^l = \{i \in [m] : \boldsymbol{W}_{O_{(i,\cdot)}}^{(t)}\boldsymbol{V}_l^n(t) < 0\}$. We have

$$F(\boldsymbol{X}^n) = \frac{1}{|\mathcal{S}^n|}\sum_{l\in\mathcal{S}^n}\sum_{i\in\mathcal{W}(t)}\frac{1}{m}\text{Relu}(\boldsymbol{W}_{O_{(i,\cdot)}}^{(t)}\boldsymbol{V}_l^n(t)) + \frac{1}{|\mathcal{S}^n|}\sum_{l\in\mathcal{S}^n}\sum_{i\in\mathcal{K}_+^l/\mathcal{W}(t)}\frac{1}{m}\text{Relu}(\boldsymbol{W}_{O_{(i,\cdot)}}^{(t)}\boldsymbol{V}_l^n(t))$$

$$- \frac{1}{|\mathcal{S}^n|}\sum_{l\in\mathcal{S}^n}\sum_{i\in\mathcal{K}_-^l}\frac{1}{m}\text{Relu}(\boldsymbol{W}_{O_{(i,\cdot)}}^{(t)}\boldsymbol{V}_l^n(t))$$

$$(53)$$

By Lemma 2, we have

$$\frac{1}{|\mathcal{S}^n|}\sum_{l\in\mathcal{S}^n}\sum_{i\in\mathcal{W}(t)}\frac{1}{m}\text{Relu}(\boldsymbol{W}_{O_{(i,\cdot)}}^{(t)}\boldsymbol{V}_l^n(t))$$

$$= \frac{1}{|\mathcal{S}^n|}\sum_{l\in\mathcal{S}_1^n}\sum_{i\in\mathcal{W}(t)}\frac{1}{m}\text{Relu}(\boldsymbol{W}_{O_{(i,\cdot)}}^{(t)}\boldsymbol{V}_l^n(t)) + \sum_{l\notin\mathcal{S}_1^n}\sum_{i\in\mathcal{W}(t)}\frac{1}{m}\text{Relu}(\boldsymbol{W}_{O_{(i,\cdot)}}^{(t)}\boldsymbol{V}_l^n(t))$$

$$\gtrsim |\mathcal{S}_1^n|\frac{1}{a|\mathcal{S}^n|}\cdot\boldsymbol{W}_{O_{(i,\cdot)}}^{(t)}\Big(\sum_{s\in\mathcal{S}_1^n}\boldsymbol{p}_s\text{softmax}(\boldsymbol{x}_s^n{}^\top\boldsymbol{W}_K^{(t)}{}^\top\boldsymbol{W}_Q^{(t)}\boldsymbol{x}_l^n) + \boldsymbol{z}(t) + \sum_{l\neq s}W_l(u)\boldsymbol{p}_l$$

$$- \eta t\Big(\sum_{j\in\mathcal{W}(t)}V_j(t)\boldsymbol{W}_{O_{(j,\cdot)}}^{(t)}{}^\top + \sum_{j\notin\mathcal{W}(t)}V_j(t)\lambda\boldsymbol{W}_{O_{(j,\cdot)}}^{(t)}{}^\top\Big)\Big)|\mathcal{W}(t)| + 0$$

$$\gtrsim \frac{|\mathcal{S}_1^n|m}{|\mathcal{S}^n|aM}\Big(1 - \epsilon_m - \frac{(\sigma+\tau)M}{\pi}\Big)\Big(\frac{1}{Bt}\sum_{b=1}^{t}\sum_{n\in\mathcal{B}_b}\frac{\eta^2 t^2 m}{a^2}\Big(\frac{b|\mathcal{S}_*^n|}{t|\mathcal{S}^n|}\|\boldsymbol{p}_1\|^2 p_n(b) - (\sigma+\tau)\Big)p_n(t)$$

$$+ \frac{1}{Bt}\sum_{b=1}^{t}\sum_{n\in\mathcal{B}_b}\frac{|\mathcal{S}_1^n|p_n(b)\eta t m}{|\mathcal{S}^n|aM}\Big(1 - \epsilon_m - \frac{(\sigma+\tau)M}{\pi}\Big)$$

$$\cdot\Big(\frac{1}{Bt}\sum_{b=1}^{t}\sum_{n\in\mathcal{B}_b}\frac{\eta^2 t b m|\mathcal{S}_1^n|}{a^2|\mathcal{S}^n|}p_n(b)\Big)^2\|\boldsymbol{p}_1\|^2 p_n(t)\Big)$$

$$(54)$$

where the second step comes from (27) and the last step is by (136). By the definition of $\mathcal{K}_+^l$, we have

$$\frac{1}{|\mathcal{S}^n|} \sum_{l \in \mathcal{S}^n} \sum_{i \in \mathcal{K}_+^l / \mathcal{W}(t)} \frac{1}{m} \text{Relu}(\boldsymbol{W}_{O_{(i,\cdot)}}^{(t)} \boldsymbol{V}_l^n(t)) \geq 0 \tag{55}$$

Combining (136) and (138), we can obtain

$$\frac{1}{|\mathcal{S}^n|} \sum_{l \in \mathcal{S}^n} \sum_{i \in \mathcal{K}_-^l} \frac{1}{m} \text{Relu}(\boldsymbol{W}_{O_{(i,\cdot)}}^{(t)} \boldsymbol{V}_l^n(t))$$

$$\leq \frac{|\mathcal{S}_2^n| m}{|\mathcal{S}^n| a M} \cdot (1 - \epsilon_m - \frac{(\sigma + \tau) M}{\pi})(\xi \|\boldsymbol{p}\| + \frac{1}{Bt} \sum_{b=1}^{t} \sum_{n \in \mathcal{B}_b} \frac{|\mathcal{S}_1^n| p_n(b) m}{|\mathcal{S}^n| a M}$$

$$\cdot (1 - \epsilon_m - \frac{(\sigma + \tau) M}{\pi})(\frac{\eta^2 t^2 m}{a^2})^2 (\sigma + \tau) \|\boldsymbol{p}\|$$

$$+ \frac{\eta^2 \lambda \sqrt{M} \xi t^2 m}{\sqrt{B} a^2} \|\boldsymbol{p}_1\|^2 + ((\sigma + \tau))(\sqrt{M} \xi + \frac{\eta^2 t^2 m}{a^2}) + \frac{1}{Bt} \sum_{b=1}^{t} \sum_{n \in \mathcal{B}_b} \frac{|\mathcal{S}_2^n| p_n(b) \eta t m}{|\mathcal{S}^n| a M}$$

$$\cdot (1 - \epsilon_m - \frac{(\sigma + \tau) M}{\pi})(\frac{\eta^2 t^2 m}{a^2})^2 \|\boldsymbol{p}_1\|^2) \phi_n(t) |\mathcal{S}_2^n| + \sum_{l=3}^{M} \frac{|\mathcal{S}_l^n|}{|\mathcal{S}^n|} (\xi \|\boldsymbol{p}\| + \frac{\eta^2 \lambda \sqrt{M} \xi t^2 m}{\sqrt{B} a^2} \|\boldsymbol{p}\|^2$$

$$+ ((\sigma + \tau))(\sqrt{M} \xi + \frac{\eta^2 t^2 m}{a^2}) \|\boldsymbol{p}\| + \frac{1}{Bt} \sum_{b=1}^{t} \sum_{n \in \mathcal{B}_b} \frac{(|\mathcal{S}_2^n| + |\mathcal{S}_1^n|) p_n(t) \eta t m}{|\mathcal{S}^n| a M} (1 - \epsilon_m - \frac{(\sigma + \tau) M}{\pi})$$

$$\cdot \frac{\eta^2 t^2 m}{a^2} \xi \|\boldsymbol{p}_1\|^2) \phi_n(t)(|\mathcal{S}^n| - |\mathcal{S}_1^n|) + \frac{|\mathcal{S}_1^n|}{|\mathcal{S}^n|} c_1^M \left( \frac{1}{Bt} \sum_{b=1}^{t} \sum_{n \in \mathcal{B}_b} \frac{\eta^2 t b m |\mathcal{S}_*^n|}{a^2 |\mathcal{S}^n|} \|\boldsymbol{p}_1\|^2 p_n(t) \right.$$

$$\cdot |(\sigma + \tau) - p_n(t)| + \frac{1}{Bt} \sum_{b=1}^{t} \sum_{n \in \mathcal{B}_b} \frac{|\mathcal{S}_1^n| p_n(t) \eta t m}{|\mathcal{S}^n| a M} c_2^M (\frac{1}{Bt} \sum_{b=1}^{t} \sum_{n \in \mathcal{B}_b} \frac{\eta^2 t b m |\mathcal{S}_1^n|}{a^2 |\mathcal{S}^n|}$$

$$\left. \cdot p_n(b))^2 \|\boldsymbol{p}_1\|^2 p_n(t) \right) \tag{56}$$

for some $c_1, c_2 \in (0, 1)$.

Note that at the $T$-th iteration,

$$K(t)$$

$$\gtrsim \eta \Big( \frac{1}{Bt} \sum_{b=1}^{t} \sum_{n \in \mathcal{B}_b} \frac{|\mathcal{S}_1^n| p_n(t) \eta t m}{|\mathcal{S}^n| a M} (1 - \epsilon_m - \frac{(\sigma + \tau) M}{\pi})(\frac{1}{Bt} \sum_{b=1}^{t} \sum_{n \in \mathcal{B}_b} \frac{\eta^2 t b m |\mathcal{S}_1^n|}{a^2 |\mathcal{S}^n|} p_n(b))^2 \|\boldsymbol{p}_1\|^2 p_n(t)$$

$$+ \frac{1}{Bt} \sum_{b=1}^{t} \sum_{n \in \mathcal{B}_b} \frac{\eta^2 t b m}{a^2} (\frac{|\mathcal{S}_1^n|}{|\mathcal{S}^n|} p_n(b) - (\sigma + \tau)) \|\boldsymbol{p}_1\|^2 p_n(t) \Big) \phi_n(t)(|\mathcal{S}^n| - |\mathcal{S}_1^n|) \|\boldsymbol{q}_1(t)\|^2$$

$$\gtrsim \frac{\eta}{e^{\|\boldsymbol{q}_1(t)\|^2 - (\delta + \tau) \|\boldsymbol{q}_1(t)\|}} \tag{57}$$

Since that

$$\boldsymbol{q}_1(T) \gtrsim (1 + \min_{l=0,1,\cdots,T-1} \{K(l)\})^T$$

$$\gtrsim (1 + \frac{\eta}{e^{\|\boldsymbol{q}_1(T)\|^2 - (\delta + \tau) \|\boldsymbol{q}_1(T)\|}})^T \tag{58}$$

To find the order-wise lower bound of $\boldsymbol{q}_1(T)$, we need to check the equation

$$\boldsymbol{q}_1(T) \lesssim (1 + \frac{1}{e^{\|\boldsymbol{q}_1(T)\|^2 - (\delta + \tau) \|\boldsymbol{q}_1(T)\|}})^T \tag{59}$$

One can obtain

$$\Theta(\log T) = \|\boldsymbol{q}_1(T)\|^2 = o(T) \tag{60}$$

Therefore,

$$p_n(T) \gtrsim \frac{T^C}{T^C + \frac{1-\alpha}{\alpha}} \geq 1 - \frac{1}{\frac{\alpha}{1-\alpha}(\eta^{-1})^C} \geq 1 - \Theta(\eta^C) \tag{61}$$

$$\phi_n(T)(|\mathcal{S}^n| - |\mathcal{S}_1^n|) \leq \eta^C \tag{62}$$

for some large $C > 0$. We require that

$$\frac{|\mathcal{S}_1^n|m}{|\mathcal{S}^n|aM}(1 - \epsilon_m - \frac{(\sigma+\tau)M}{\pi})\Big(\frac{1}{BT}\sum_{b=1}^{T}\sum_{n\in\mathcal{B}_b}\frac{|\mathcal{S}_1^n|p_n(b)\eta tm}{|\mathcal{S}^n|aM}(1 - \epsilon_m - \frac{(\sigma+\tau)M}{\pi})$$

$$\cdot(\frac{1}{BT}\sum_{b=1}^{T}\sum_{n\in\mathcal{B}_b}\frac{\eta^2 tbm|\mathcal{S}_1^n|}{a^2|\mathcal{S}^n|}p_n(b))^2\|\boldsymbol{p}_1\|^2 p_n(b)$$

$$+\frac{1}{BT}\sum_{b=1}^{T}\sum_{n\in\mathcal{B}_b}\frac{\eta^2 t^2 m}{a^2}(\frac{b|\mathcal{S}_*^n|}{t|\mathcal{S}^n|}\|\boldsymbol{p}_1\|^2 p_n(b) - (\sigma+\tau))p_n(b)\Big)$$

$$\gtrsim \frac{|\mathcal{S}_1^n|m}{|\mathcal{S}^n|aM}(1 - \epsilon_m - \frac{(\sigma+\tau)M}{\pi})\Big(\frac{1}{N}\sum_{n=1}^{N}\frac{|\mathcal{S}_1^n|p_n(T)\eta Tm}{|\mathcal{S}^n|aM}(1 - \epsilon_m - \frac{(\sigma+\tau)M}{\pi})\cdot(\frac{1}{N}\sum_{n=1}^{N}\frac{\eta^2 T^2 m|\mathcal{S}_1^n|}{a^2|\mathcal{S}^n|}$$

$$\cdot p_n(T))^2\|\boldsymbol{p}_1\|^2 p_n(T) + \frac{1}{N}\sum_{n=1}^{N}\frac{\eta^2 T^2 m}{a^2}(\frac{|\mathcal{S}_*^n|}{|\mathcal{S}^n|}\|\boldsymbol{p}_1\|^2 p_n(T) - (\sigma+\tau))p_n(T)\Big)$$

$$:=a_0(\eta T)^5 + a_1(\eta T)^2$$

$$>1,$$

$$\tag{63}$$

where the first step is by letting $a = \sqrt{m}$ and $m \gtrsim M^2$. We replace $p_n(b)$ with $p_n(T)$ because when $b$ achieves the level of $T$, $b^{o_1}p_n(b)^{o_2}$ is close to $b^{o_1}$ for $o_1, o_2 \geq 0$ by (61). Thus,

$$\sum_{b=1}^{T}b^{o_1}p_n(b)^{o_2} \gtrsim T^{o_1+1}p_n(\Theta(1)\cdot T)^{o_2} \gtrsim T^{o_1+1}p_n(T)^{o_2} \tag{64}$$

We also require

$$\frac{\eta^2\lambda\sqrt{M}\xi t^2 m}{\sqrt{B}a^2} \leq \epsilon_0, \tag{65}$$

for some $\epsilon_0 > 0$.

We know that

$$\Big|\frac{1}{N}\sum_{n=1}^{N}\frac{|\mathcal{S}_*^n|}{|\mathcal{S}^n|}p_n(T)(p_n(T) - (\sigma+\tau)) - \mathbb{E}\Big[\frac{|\mathcal{S}_*^n|}{|\mathcal{S}^n|}\Big]\Big|$$

$$\leq\Big|\frac{1}{N}\sum_{n=1}^{N}\frac{|\mathcal{S}_*^n|}{|\mathcal{S}^n|}p_n(T)(p_n(T) - (\sigma+\tau)) - \mathbb{E}\Big[\frac{|\mathcal{S}_*^n|}{|\mathcal{S}^n|}p_n(T)(p_n(T) - (\sigma+\tau))\Big]\Big|$$

$$+\Big|\mathbb{E}\Big[\frac{|\mathcal{S}_*^n|}{|\mathcal{S}^n|}\Big(p_n(T)(p_n(T) - (\sigma+\tau)) - 1\Big)\Big]\Big|$$

$$\lesssim\sqrt{\frac{\log N}{N}} + c'(1 - \zeta) + c''((\sigma+\tau)) \tag{66}$$

for $c' > 0$ and $c'' > 0$. We can then have

$$t \geq T = \frac{\eta^{-1}}{a_1} = \frac{\eta^{-1}}{\frac{|\mathcal{S}_1^n|}{|\mathcal{S}^n|}(1 - \epsilon_m - \frac{(\sigma+\tau)M}{\pi})\frac{1}{N}\sum_{n=1}^{N}(\frac{|\mathcal{S}_*^n|}{|\mathcal{S}^n|}\|\boldsymbol{p}_1\|^2 p_n(t) - (\sigma+\tau))p_n(t)}$$

$$=\Theta(\frac{\eta^{-1}}{(1 - \epsilon_m - \frac{(\sigma+\tau)M}{\pi})(\mathbb{E}\Big[\frac{|\mathcal{S}_*^n|}{|\mathcal{S}^n|}\Big] - \sqrt{\frac{\log N}{N}} - c'(1 - \zeta) - c''(\sigma+\tau))}) \tag{67}$$

$$=\Theta(\frac{\eta^{-1}}{(1 - \epsilon_m - \frac{(\sigma+\tau)M}{\pi})\mathbb{E}\Big[\frac{|\mathcal{S}_*^n|}{|\mathcal{S}^n|}\Big]})$$

where

$$\alpha \geq \frac{1 - \alpha_{nd}}{1 + \epsilon_{\mathcal{S}} e^{-(\delta+\tau)}(1 - (\sigma + \tau))} \tag{68}$$

by (157), as long as

$$N \geq \Omega\left(\frac{1}{(\alpha - c'(1 - \zeta) - c''((\sigma + \tau)))^2}\right) \tag{69}$$

and

$$B \gtrsim \frac{1}{((1 - (\tau + \sigma))e^{-(\delta+\tau)}\frac{1}{2}\alpha_* - (\tau + \sigma)e^{-(\delta+\tau)})^2} = \Theta\left(\frac{1}{(\alpha_* - e^{-(\delta+\tau)}(\tau + \sigma))^2}\right) = \frac{N}{\Theta(1)} \tag{70}$$

where $\zeta \geq 1 - \eta^{10}$. If there is no mechanism like the self-attention to compute the weight using the correlations between tokens, we have

$$c'(1 - \zeta) = O(\alpha_*(1 - \alpha_*)), \tag{71}$$

which can scale up the sample complexity in (69) by $\alpha_*^{-2}$.

Therefore, we can obtain

$$F(\boldsymbol{X}^n) > 1 \tag{72}$$

Similarly, we can derive that for $y = -1$,

$$F(\boldsymbol{X}) < -1 \tag{73}$$

Hence, for all $n \in [N]$,

$$\text{Loss}(\boldsymbol{X}^n, y^n) = 0 \tag{74}$$

We also have

$$\text{Loss} = \mathbb{E}_{(\boldsymbol{X}^n, y^n) \sim \mathcal{D}}[\text{Loss}(\boldsymbol{X}^n, y^n)] = 0 \tag{75}$$

with the conditions of sample complexity and the number of iterations.

**Proof of Proposition 1:**

The main proof is the same as the proof of Theorem 1. The only difference is that we need to modify (66) as follows

$$\begin{aligned}
&\left| \frac{1}{N} \sum_{n=1}^{N} \frac{|\mathcal{S}_*^n|}{|\mathcal{S}^n|} p_n(T)(p_n(T) - (\sigma + \tau)) - \mathbb{E}\left[\frac{|\mathcal{S}_*^n|}{|\mathcal{S}^n|}\right] \right| \\
&\leq \left| \frac{1}{N} \sum_{n=1}^{N} \frac{|\mathcal{S}_*^n|}{|\mathcal{S}^n|} p_n(0)(p_n(0) - (\sigma + \tau)) - \mathbb{E}\left[\frac{|\mathcal{S}_*^n|}{|\mathcal{S}^n|} p_n(0)(p_n(T) - (\sigma + \tau))\right] \right| \\
&\quad + \left| \mathbb{E}\left[\frac{|\mathcal{S}_*^n|}{|\mathcal{S}^n|}\left(p_n(0)(p_n(0) - (\sigma + \tau)) - 1\right)\right] \right| \\
&\lesssim \sqrt{\frac{\log N}{N}} + |1 - \Theta(\alpha_*^2) + \Theta(\alpha_*)(\sigma + \tau)|
\end{aligned} \tag{76}$$

where the first step is because $p_n(T)$ does not update since $\boldsymbol{W}_K^{(t)}$ and $\boldsymbol{W}_Q^{(t)}$ are fixed at initialization $\boldsymbol{W}_K^{(0)}$ and $\boldsymbol{W}_Q^{(0)}$, and the second step is by $p_n(0) = \Theta(\alpha_*)$. Since that

$$\sqrt{\frac{\log N}{N}} + |1 - \Theta(\alpha_*^2) + \Theta(\alpha_*)(\sigma + \tau)| \leq \Theta(1) \cdot \alpha_*, \tag{77}$$

we have

$$\begin{aligned}
N &\geq \frac{1}{(\Theta(\alpha_*) - 1 + \Theta(\alpha_*^2) - \Theta(\alpha_*)(\sigma + \tau))^2} \\
&= \Omega\left(\frac{1}{(\alpha_*(\alpha_* - \sigma - \tau))^2}\right)
\end{aligned} \tag{78}$$

**Proof of Proposition 2:**

It can be easily derived from Claim 2 of Lemma 2, (60), and (61).

# E  PROOF OF LEMMA 2

We prove the whole lemma by a long induction, which is the reason why we prefer to wrap three claims into one lemma. To make it easier to follow, however, we break this Section into three parts to introduce the proof of three claims of Lemma 2 separately.

## E.1  PROOF OF CLAIM 1 OF LEMMA 2

Although it looks cumbersome, the key idea of Claim 1 is to characterize the growth of $\boldsymbol{W}_O^{(t)}$ in terms of $\boldsymbol{p}_l$, $l \in [M]$. We compare $\boldsymbol{W}_{O_{(i,\cdot)}}^{(t+1)}\boldsymbol{p}_l$ and $\boldsymbol{W}_{O_{(i,\cdot)}}^{(t)}\boldsymbol{p}_l$ to see the direction of growth by computing the gradient. One can eventually find that lucky neurons grow the most in directions of $\boldsymbol{p}_1$ and $\boldsymbol{p}_2$, i.e., the feature of label-relevant patterns, while unlucky neurons do not change much in magnitude. We start our poof. At the $t$-th iteration, if $l \in \mathcal{S}_1^n$, let

$$
\begin{aligned}
\boldsymbol{V}_l^n(t) &= \boldsymbol{W}_V^{(t)}\boldsymbol{X}^n \text{softmax}(\boldsymbol{X}^{n\top}\boldsymbol{W}_K^{(t)\top}\boldsymbol{W}_Q^{(t)}\boldsymbol{x}_l^n) \\
&= \sum_{s \in \mathcal{S}_1} \text{softmax}(\boldsymbol{x}_s^{n\top}\boldsymbol{W}_K^{(t)\top}\boldsymbol{W}_Q^{(t)}\boldsymbol{x}_l^n)\boldsymbol{p}_1 + \boldsymbol{z}(t) + \sum_{j \neq 1} W_j^n(t)\boldsymbol{p}_j \\
&\quad - \eta\Big(\sum_{i \in \mathcal{W}(t)} V_i(t)\boldsymbol{W}_{O_{(i,\cdot)}}^{(t)\top} + \sum_{i \notin \mathcal{W}(t)} V_i(t)\lambda\boldsymbol{W}_{O_{(i,\cdot)}}^{(t)\top}\Big)
\end{aligned} \tag{79}
$$

, $l \in [M]$, where the second step comes from (27). Then we have

$$
W_l^n(t) \le \frac{|\mathcal{S}_j^n|e^{\delta\|\boldsymbol{q}_1(t)\|}}{(|\mathcal{S}^n| - |\mathcal{S}_1^n|)e^{\delta\|\boldsymbol{q}_1(t)\|} + |\mathcal{S}_1^n|e^{\|\boldsymbol{q}_1(t)\|^2 - \delta\|\boldsymbol{q}_1(t)\|}} = \nu_n(t)|\mathcal{S}_j^n| \tag{80}
$$

Hence, by (18),

$$
\frac{\partial\overline{\textbf{Loss}}_b}{\partial\boldsymbol{W}_{O_{(i,\cdot)}}^\top} = -\frac{1}{B}\sum_{n \in \mathcal{B}_b} y^n \frac{1}{|\mathcal{S}^n|}\sum_{l \in \mathcal{S}^n} a_{(l)_i}\mathbb{1}[\boldsymbol{W}_{O_{(i,\cdot)}}\boldsymbol{V}_l^n(t) \ge 0]\boldsymbol{V}_l^n(t)^\top \tag{81}
$$

Define that for $j \in [M]$,

$$
I_4 = \frac{1}{B}\sum_{n \in \mathcal{B}_b} \eta y^n \frac{1}{|\mathcal{S}^n|}\sum_{l \in \mathcal{S}^n} a_{(l)_i}\mathbb{1}[\boldsymbol{W}_{O_{(i,\cdot)}}^{(t)}\boldsymbol{V}_l^n(t) \ge 0]\sum_{k \in \mathcal{W}(t)} V_k(t)\boldsymbol{W}_{O_{(k,\cdot)}}^{(t)}\boldsymbol{p}_j \tag{82}
$$

$$
I_5 = \frac{1}{B}\sum_{n \in \mathcal{B}_b} \eta y^n \frac{1}{|\mathcal{S}^n|}\sum_{l \in \mathcal{S}^n} a_{(l)_i}\mathbb{1}[\boldsymbol{W}_{O_{(i,\cdot)}}^{(t)}\boldsymbol{V}_l^n(t) \ge 0]\sum_{k \notin \mathcal{W}(t)} V_k(t)\boldsymbol{W}_{O_{(k,\cdot)}}^{(t)}\boldsymbol{p}_j, \tag{83}
$$

and we can then obtain

$$
\begin{aligned}
&\left\langle \boldsymbol{W}_{O_{(i,\cdot)}}^{(t+1)\top}, \boldsymbol{p}_j \right\rangle - \left\langle \boldsymbol{W}_{O_{(i,\cdot)}}^{(t)\top}, \boldsymbol{p}_j \right\rangle \\
&= \frac{1}{B}\sum_{n \in \mathcal{B}_b} \eta y^n \frac{1}{|\mathcal{S}^n|}\sum_{l \in \mathcal{S}^n} a_{(l)_i}\mathbb{1}[\boldsymbol{W}_{O_{(i,\cdot)}}^{(t)}\boldsymbol{V}_l^n(t) \ge 0]\boldsymbol{V}_l^n(t)^\top\boldsymbol{p}_j \\
&= \frac{1}{B}\sum_{n \in \mathcal{B}_b} \eta y^n \frac{1}{|\mathcal{S}^n|}\sum_{l \in \mathcal{S}^n} a_{(l)_i}\mathbb{1}[\boldsymbol{W}_{O_{(i,\cdot)}}^{(t)}\boldsymbol{V}_l^n(t) \ge 0]\boldsymbol{z}_l(t)^\top\boldsymbol{p}_j \\
&\quad + \frac{1}{B}\sum_{n \in \mathcal{B}_b} \eta y^n \frac{1}{|\mathcal{S}^n|}\sum_{l \in \mathcal{S}^n} a_{(l)_i}\mathbb{1}[\boldsymbol{W}_{O_{(i,\cdot)}}^{(t)}\boldsymbol{V}_l^n(t) \ge 0]\sum_{s \in \mathcal{S}_l} \text{softmax}(\boldsymbol{x}_s^\top\boldsymbol{W}_K^{(t)\top}\boldsymbol{W}_Q^{(t)}\boldsymbol{x}_l)\boldsymbol{p}_l^\top\boldsymbol{p}_j \\
&\quad + \frac{1}{B}\sum_{n \in \mathcal{B}_b} \eta y^n \frac{1}{|\mathcal{S}^n|}\sum_{l \in \mathcal{S}^n} a_{(l)_i}\mathbb{1}[\boldsymbol{W}_{O_{(i,\cdot)}}^{(t)}\boldsymbol{V}_l^n(t) \ge 0]\sum_{k \neq l} W_l(t)\boldsymbol{p}_k^\top\boldsymbol{p}_j + I_4 + I_5 \\
&:= I_1 + I_2 + I_3 + I_4 + I_5,
\end{aligned} \tag{84}
$$

where

$$I_1 = \frac{1}{B} \sum_{n \in \mathcal{B}_b} \eta y^n \frac{1}{|\mathcal{S}^n|} \sum_{l \in \mathcal{S}^n} a_{(l)_i} \mathbb{1}[\boldsymbol{W}_{O_{(i,\cdot)}}^{(t)} \boldsymbol{V}_l^n(t) \geq 0] \boldsymbol{z}_l(t)^\top \boldsymbol{p}_j \tag{85}$$

$$I_2 = \frac{1}{B} \sum_{n \in \mathcal{B}_b} \eta y^n \frac{1}{|\mathcal{S}^n|} \sum_{l \in \mathcal{S}^n} a_{(l)_i} \mathbb{1}[\boldsymbol{W}_{O_{(i,\cdot)}}^{(t)} \boldsymbol{V}_l^n(t) \geq 0] \sum_{s \in \mathcal{S}_l} \mathrm{softmax}(\boldsymbol{x}_s^\top \boldsymbol{W}_K^{(t)\top} \boldsymbol{W}_Q^{(t)} \boldsymbol{x}_l) \boldsymbol{p}_l^\top \boldsymbol{p}_j \tag{86}$$

$$I_3 = \frac{1}{B} \sum_{n \in \mathcal{B}_b} \eta y^n \frac{1}{|\mathcal{S}^n|} \sum_{l \in \mathcal{S}^n} a_{(l)_i} \mathbb{1}[\boldsymbol{W}_{O_{(i,\cdot)}}^{(t)} \boldsymbol{V}_l^n(t) \geq 0] \sum_{k \neq l} W_l(t) \boldsymbol{p}_k^\top \boldsymbol{p}_j \tag{87}$$

We then show the statements in different cases.
(1) When $j = 1$, since that $\Pr(y^n = 1) = \Pr(y^n = -1) = 1/2$, by Hoeffding's inequality in (26), we can derive

$$\Pr\left(\left|\frac{1}{B} \sum_{n \in \mathcal{B}_b} y^n\right| \geq \sqrt{\frac{\log B}{B}}\right) \leq B^{-c} \tag{88}$$

$$\Pr\left(\left|\boldsymbol{z}_l(t)^\top \boldsymbol{p}_1\right| \geq \sqrt{((\sigma + \tau))^2 \log m}\right) \leq m^{-c} \tag{89}$$

Hence, with probability at least $1 - (mB)^{-c}$, we have

$$|I_1| \leq \frac{\eta((\sigma + \tau))}{a} \sqrt{\frac{\log m \log B}{B}} \tag{90}$$

For $i \in \mathcal{W}(t)$, from the derivation in (133) later, we have

$$\boldsymbol{W}_{O_{(i,\cdot)}}^{(t)} \sum_{s=1}^{L} \boldsymbol{W}_V^{(t)} \boldsymbol{x}_s^n \mathrm{softmax}(\boldsymbol{x}_s^{n\top} \boldsymbol{W}_K^{(t)\top} \boldsymbol{W}_Q^{(t)} \boldsymbol{x}_l^n) > 0 \tag{91}$$

Denote $p_n(t) = |\mathcal{S}_1^n| \nu_n(t) e^{\|\boldsymbol{q}_1(t)\|^2 - 2\delta\|\boldsymbol{q}_1(t)\|}$. Hence,

$$I_2 \gtrsim \eta \cdot \frac{1}{B} \sum_{n \in \mathcal{B}_b} \frac{|\mathcal{S}_1^n| - |\mathcal{S}_2^n|}{|\mathcal{S}^n|} \cdot \frac{1}{a} \|\boldsymbol{p}_1\|^2 \cdot p_n(t) \gtrsim \eta \frac{1}{B} \sum_{n \in \mathcal{B}_b} \frac{|\mathcal{S}_1^n|}{|\mathcal{S}^n|} \cdot \frac{1}{a} \|\boldsymbol{p}_1\|^2 \cdot p_n(t) \tag{92}$$

$$I_3 = 0 \tag{93}$$

$$I_4 \gtrsim \frac{1}{B} \sum_{b=1}^{t} \sum_{n \in \mathcal{B}_b} \frac{\eta^2 b |\mathcal{S}_1^n|}{|\mathcal{S}^n| a} \frac{1}{2B} \sum_{n \in \mathcal{B}_b} \frac{|\mathcal{S}_1^n| m}{|\mathcal{S}^n| aM} p_n(t) \|\boldsymbol{p}_1\|^2 \left(1 - \epsilon_m - \frac{(\sigma + \tau)M}{\pi}\right) \boldsymbol{W}_{O_{(i,\cdot)}} \boldsymbol{p}_1 \tag{94}$$

$$|I_5| \lesssim \frac{1}{B} \sum_{b=1}^{T} \sum_{n \in \mathcal{B}_b} \frac{\eta^2 b |\mathcal{S}_1^n|}{|\mathcal{S}^n| a} \left(1 - \epsilon_m - \frac{(\sigma + \tau)M}{\pi}\right) \frac{1}{2B} \sum_{n \in \mathcal{B}_b} \frac{|\mathcal{S}_2^n| m}{|\mathcal{S}^n| aM} p_n(t) \|\boldsymbol{p}_1\|^2 \boldsymbol{W}_{O_{(i,\cdot)}} \boldsymbol{p}_2$$
$$+ \frac{\eta^2 t m}{\sqrt{B} a^2} \boldsymbol{W}_{O_{(i,\cdot)}} \boldsymbol{p}_M (1 + (\sigma + \tau)) \tag{95}$$

Hence, combining (90), (92), (93), (94), and (95), we can obtain

$$\left\langle \boldsymbol{W}_{O_{(i,\cdot)}}^{(t+1)\top}, \boldsymbol{p}_1 \right\rangle - \left\langle \boldsymbol{W}_{O_{(i,\cdot)}}^{(t)\top}, \boldsymbol{p}_1 \right\rangle$$

$$\gtrsim \frac{\eta}{a} \cdot \frac{1}{B} \sum_{n \in \mathcal{B}_b} \left(\frac{|\mathcal{S}_1^n|}{|\mathcal{S}^n|} p_n(t) - ((\sigma + \tau)) + \frac{\eta t |\mathcal{S}_1^n|}{|\mathcal{S}^n|} \frac{1}{2B} \sum_{n \in \mathcal{B}_b} \frac{|\mathcal{S}_1^n| m}{|\mathcal{S}^n| aM} p_n(t) (1 - \epsilon_m - \frac{(\sigma + \tau)M}{\pi})\right)$$

$$\cdot \boldsymbol{W}_{O_{(i,\cdot)}} \boldsymbol{p}_1 (1 - (\sigma + \tau)) - \frac{\eta t |\mathcal{S}_1^n|}{|\mathcal{S}^n|} \frac{1}{2B} \sum_{n \in \mathcal{B}_b} \frac{|\mathcal{S}_2^n| m}{|\mathcal{S}^n| aM} p_n(t) (1 - \epsilon_m - \frac{(\sigma + \tau)M}{\pi})$$

$$\cdot \boldsymbol{W}_{O_{(i,\cdot)}} \boldsymbol{p}_2 (1 + (\sigma + \tau)) - \frac{\eta t m \boldsymbol{W}_{O_{(i,\cdot)}} \boldsymbol{p}_M (1 + (\sigma + \tau))}{\sqrt{B} a}) \|\boldsymbol{p}_1\|^2$$

$$\gtrsim \frac{\eta}{aB} \sum_{n \in \mathcal{B}_b} \left(\frac{|\mathcal{S}_1^n|}{|\mathcal{S}^n|} p_n(t) - ((\sigma + \tau)) + \frac{\eta t |\mathcal{S}_1^n|}{|\mathcal{S}^n|} \frac{1}{2B} \sum_{n \in \mathcal{B}_b} \frac{|\mathcal{S}_1^n| m}{|\mathcal{S}^n| aM} p_n(t)\right.$$

$$\left. \cdot (1 - \epsilon_m - \frac{(\sigma + \tau)M}{\pi}) \boldsymbol{W}_{O_{(i,\cdot)}} \boldsymbol{p}_1\right) \|\boldsymbol{p}_1\|^2 \tag{96}$$

Since that $\boldsymbol{W}_{O_{(i,\cdot)}}^{(0)} \sim \mathcal{N}(0, \frac{\xi^2 \boldsymbol{I}}{m_a})$, by the standard property of Gaussian distribution, we have

$$\Pr(\|\boldsymbol{W}_{O_{(i,\cdot)}}^{(0)}\| \leq \xi) \leq \xi \tag{97}$$

Therefore, with high probability for all $i \in [m]$, we have

$$\|\boldsymbol{W}_{O_{(i,\cdot)}}^{(0)}\| \gtrsim \xi \tag{98}$$

Therefore, we can derive

$$\boldsymbol{W}_{O_{(i,\cdot)}}^{(t+1)}\boldsymbol{p}_1 \gtrsim \exp(\frac{1}{B(t+1)} \sum_{b=1}^{t+1} \sum_{n \in \mathcal{B}_b} \frac{\eta^2 b(t+1)m}{|\mathcal{S}^n|a^2}|\mathcal{S}_1^n|\|\boldsymbol{p}_1\|^2 p_n(b)) + \xi(1-(\sigma+\tau))$$

$$\gtrsim \exp(\frac{1}{B} \sum_{n \in \mathcal{B}_b} \frac{\eta^2(t+1)^2m}{|\mathcal{S}^n|a^2}|\mathcal{S}_1^n|\|\boldsymbol{p}_1\|^2 p_n(t)) + \xi(1-(\sigma+\tau)) \tag{99}$$

by verifying that

$$\frac{\eta}{a} + \frac{\eta^2 tm}{a^2}\exp((\frac{1}{\Theta(1)} \cdot \frac{\eta^2 t^2 m}{a^2}) - 1 + \xi) \geq \exp(\frac{1}{\Theta(1)} \cdot \frac{\eta^2 t^2 m}{a^2})(\exp(\frac{\eta^2(2t+1)m}{\Theta(1) \cdot a^2}) - 1)$$

$$\gtrsim \exp(\frac{1}{\Theta(1)} \cdot \frac{\eta^2 t^2 m}{a^2})\frac{\eta^2 tm}{2a^2} \tag{100}$$

When $\eta t < \frac{a}{m}$, we have

$$\frac{\eta}{a} + \frac{\eta^2 tm}{a^2}(-1 + \xi) \geq 0 \tag{101}$$

When $\eta t \geq \frac{a}{m}$, we have that

$$g(t) := \frac{\eta^2 tm}{a^2}(\frac{1}{2}\exp(\frac{\eta^2 t^2 m}{a^2 \Theta(1)}) - 1 + \xi) + \frac{\eta}{a} \geq g(\frac{a}{\eta m}) > 0 \tag{102}$$

since that $g(t)$ is monotonically increasing. Hence, (99) is verified.
Since that

$$\eta t \leq O(1), \tag{103}$$

to simplify the further analysis, we will use the bound

$$\boldsymbol{W}_{O_{(i,\cdot)}}^{(t+1)}\boldsymbol{p}_1 \gtrsim \frac{1}{Bt}\sum_{b=1}^{t+1}\sum_{n \in \mathcal{B}_b} \frac{\eta^2(t+1)bm}{|\mathcal{S}^n|a^2}|\mathcal{S}_1^n|\|\boldsymbol{p}_1\|^2 p_n(b) + \xi(1-(\sigma+\tau)) \tag{104}$$

Note that this bound does not order-wise affect the final result of the required number of iterations.
(2) When $\boldsymbol{p}_j \in \mathcal{P}/p^+$, we have

$$I_2 = 0 \tag{105}$$

$$|I_3| \leq \frac{1}{B}\sum_{n \in \mathcal{B}_b}\nu_n(t)\frac{\eta|\mathcal{S}_l^n|}{a}\sqrt{\frac{\log m \log B}{B}}\|\boldsymbol{p}\|^2 \tag{106}$$

$$|I_4| \leq \frac{\eta^2}{a}\sum_{b=1}^{t}\sqrt{\frac{\log m \log B}{B}}\frac{1}{2B}\sum_{b=1}^{t}\sum_{n \in \mathcal{B}_b}\frac{|\mathcal{S}_1^n|\eta bm}{|\mathcal{S}^n|aM}p_n(b)(\frac{(\eta t)^2 m}{a^2} + \xi)\|\boldsymbol{p}\| \tag{107}$$

$$|I_5| \lesssim \frac{\eta^2 tm}{\sqrt{B}a^2}\xi\|\boldsymbol{p}\|^2 + \frac{\eta^2}{a}\sum_{b=1}^{t}\sqrt{\frac{\log m \log B}{B}}\frac{1}{2B}\sum_{n \in \mathcal{B}_b}\frac{|\mathcal{S}_2^n|m}{|\mathcal{S}^n|aM}p_n(t)\xi\|\boldsymbol{p}\| \tag{108}$$

with probability at least $1 - (mB)^{-c}$. (107) comes from (34). Then, combining (90), (105), (106), (107) and (108), we can obtain

$$\left|\left\langle \boldsymbol{W}_{O_{(i,\cdot)}}^{(t+1)\top}, \boldsymbol{p}_j \right\rangle - \left\langle \boldsymbol{W}_{O_{(i,\cdot)}}^{(t)\top}, \boldsymbol{p}_j \right\rangle\right|$$

$$\lesssim \frac{\eta}{a} \cdot \frac{1}{B}\sum_{n \in \mathcal{B}_b}(\frac{|\mathcal{S}_l^n|}{|\mathcal{S}^n|}|\mathcal{S}_l^n|\nu_n(t) + ((\sigma+\tau)))$$

$$+ \sum_{b=1}^{t}\frac{|\mathcal{S}_1^n|p_n(b)\eta m}{|\mathcal{S}^n|aM}(\frac{\eta^2 t^2 m}{a^2} + \xi))\sqrt{\frac{\log m \log B}{B}}\|\boldsymbol{p}\|^2 + \frac{\eta^2 tm}{\sqrt{B}a^2}\xi\|\boldsymbol{p}\| \tag{109}$$

Furthermore, we have

$$
\begin{aligned}
\boldsymbol{W}_{O_{(i,\cdot)}}^{(t+1)}\boldsymbol{p}_j \lesssim & \frac{\eta}{a}\sum_{b=1}^{(t+1)}\cdot\frac{1}{B}\sum_{n\in\mathcal{B}_b}(\frac{|\mathcal{S}_l^n|}{|\mathcal{S}^n|}|\mathcal{S}_l^n|\nu_n(b)+((\sigma+\tau)) \\
& +\sum_{b=1}^{t}\frac{|\mathcal{S}_1^n|p_n(b)\eta m}{|\mathcal{S}^n|aM}(\frac{\eta^2 t^2}{m}+\xi))\sqrt{\frac{\log m\log B}{B}}\|\boldsymbol{p}\|+\frac{\eta^2 t^2 m}{\sqrt{B}a^2}\xi\|\boldsymbol{p}\|+\xi\|\boldsymbol{p}\| \\
\leq & \xi\|\boldsymbol{p}\|
\end{aligned}
\tag{110}
$$

where the last step is by

$$
\eta t \leq O(1)
\tag{111}
$$

to ensure a non-zero gradient.

(3) If $i\in\mathcal{U}(t)$, following the derivation of (104) and (110), we can conclude that

$$
\boldsymbol{W}_{O_{(i,\cdot)}}^{(t+1)}\boldsymbol{p}_2 \gtrsim \frac{1}{B(t+1)}\sum_{b=1}^{t+1}\sum_{n\in\mathcal{B}_b}\frac{\eta^2(t+1)bm|\mathcal{S}_2^n|}{|\mathcal{S}^n|a^2}\|\boldsymbol{p}_2\|^2 p_n(b)+\xi(1-(\sigma+\tau))
\tag{112}
$$

$$
\boldsymbol{W}_{O_{(i,\cdot)}}^{(t)}\boldsymbol{p} \leq \xi\|\boldsymbol{p}\|, \quad \text{for } \boldsymbol{p}\in\mathcal{P}/\boldsymbol{p}_2,
\tag{113}
$$

(4) If $i\notin(\mathcal{W}(t)\cup\mathcal{U}(t))$,

$$
|I_2+I_3| \leq \frac{\eta}{a}\sqrt{\frac{\log m\log B}{B}}\|\boldsymbol{p}\|^2
\tag{114}
$$

Following (107) and (108), we have

$$
|I_4| \leq \sum_{b=1}^{t}\frac{\eta^2}{a}\sqrt{\frac{\log m\log B}{B}}\frac{1}{2B}\sum_{n\in\mathcal{B}_b}\frac{|\mathcal{S}_1^n|m}{|\mathcal{S}^n|aM}p_n(b)(\frac{\eta^2 t^2 m}{a^2}+\xi)\|\boldsymbol{p}\|
\tag{115}
$$

$$
|I_5| \lesssim \frac{\eta^2 tm}{\sqrt{B}a^2}\xi\|\boldsymbol{p}\|^2+\sum_{b=1}^{t}\frac{\eta^2}{a}\sqrt{\frac{\log m\log B}{B}}\frac{1}{2B}\sum_{n\in\mathcal{B}_b}\frac{|\mathcal{S}_2^n|m}{|\mathcal{S}^n|aM}p_n(b)\xi\|\boldsymbol{p}\|
\tag{116}
$$

Hence, combining (114), (115), and (116), we can obtain

$$
\begin{aligned}
& \left|\left\langle\boldsymbol{W}_{O_{(i,\cdot)}}^{(t+1)\top},\boldsymbol{p}\right\rangle-\left\langle\boldsymbol{W}_{O_{(i,\cdot)}}^{(t)\top},\boldsymbol{p}\right\rangle\right| \\
\lesssim & \frac{\eta}{a}\cdot(\|\boldsymbol{p}\|+((\sigma+\tau))+\sum_{b=1}^{t}\frac{|\mathcal{S}_1^n|p_n(b)\eta m}{|\mathcal{S}^n|Ma}(\frac{\eta^2 t^2 m}{a^2}+\xi))\sqrt{\frac{\log m\log B}{B}}\|\boldsymbol{p}\|+\frac{\eta^2 tm}{\sqrt{B}a^2}\xi\|\boldsymbol{p}\|^2,
\end{aligned}
\tag{117}
$$

and

$$
\begin{aligned}
\boldsymbol{W}_{O_{(i,\cdot)}}^{(t+1)}\boldsymbol{p} \lesssim & \sum_{b=1}^{t+1}\frac{\eta}{a}\cdot(\|\boldsymbol{p}\|+((\sigma+\tau))+\sum_{b=1}^{t}\frac{|\mathcal{S}_1^n|p_n(b)\eta m}{|\mathcal{S}^n|aM}(\frac{\eta^2 t^2 m}{a^2}+\xi)) \\
& \cdot\sqrt{\frac{\log m\log B}{B}}\|\boldsymbol{p}\|+\frac{\eta^2 t^2 m}{\sqrt{B}a^2}\xi\|\boldsymbol{p}\|^2+\xi\|\boldsymbol{p}\| \\
\leq & \xi\|\boldsymbol{p}\|
\end{aligned}
\tag{118}
$$

where the last step is by

$$
\eta t \leq O(1)
\tag{119}
$$

(5) We finally study the bound of $\boldsymbol{W}_{O_{(i,\cdot)}}^{(t)}$ and the product with the noise term according to the analysis above.

By (52), for the lucky neuron $i$, since that the update of $\boldsymbol{W}_{O_{(i,\cdot)}}^{(t)}$ lies in the subspace spanned by $\mathcal{P}$ and $\boldsymbol{p}_1,\boldsymbol{p}_2,\cdots,\boldsymbol{p}_M$ all have a unit norm, we can derive

$$
\begin{aligned}
\|\boldsymbol{W}_{O_{(i,\cdot)}}^{(t+1)}\|^2 = & \sum_{l=1}^{M}(\boldsymbol{W}_{O_{(i,\cdot)}}^{(t+1)}\boldsymbol{p}_l)^2 \geq (\boldsymbol{W}_{O_{(i,\cdot)}}^{(t+1)}\boldsymbol{p}_1)^2 \\
\gtrsim & (\frac{1}{B(t+1)}\sum_{b=1}^{t+1}\sum_{n\in\mathcal{B}_b}\frac{\eta^2(t+1)b|\mathcal{S}_1^n|m}{a^2|\mathcal{S}^n|}\|\boldsymbol{p}\|^2 p_n(b))^2
\end{aligned}
\tag{120}
$$

$$\|\boldsymbol{W}_{O_{(i,\cdot)}}^{(t+1)}\|^2 \leq M\xi^2\|\boldsymbol{p}\|^2 + (\frac{\eta^2(t+1)^2 m}{a^2})^2\|\boldsymbol{p}\|^2 \tag{121}$$

$$\|\boldsymbol{W}_{O_{(i,\cdot)}}^{(t+1)}\boldsymbol{z}_l(t)\| \leq \left|((\sigma+\tau))\sqrt{\sum_{\boldsymbol{p}\in\mathcal{P}}\left\langle \boldsymbol{W}_{O_{(i,\cdot)}}^{(t)}{}^\top, \boldsymbol{p}\right\rangle^2}\right|$$
$$\leq ((\sigma+\tau))(\sqrt{M}\xi + \frac{\eta^2(t+1)^2 m}{a^2})\|\boldsymbol{p}\| \tag{122}$$

For the unlucky neuron $i$, we can similarly obtain

$$|\boldsymbol{W}_{O_{(i,\cdot)}}^{(t+1)}\boldsymbol{z}_l(t)| \leq \left|((\sigma+\tau))\sum_{\boldsymbol{p}\in\mathcal{P}}\left\langle \boldsymbol{W}_{O_{(i,\cdot)}}^{(t+1)}{}^\top, \boldsymbol{p}\right\rangle\right| \leq ((\sigma+\tau))\sqrt{M}\xi\|\boldsymbol{p}\| \tag{123}$$

$$\|\boldsymbol{W}_{O_{(i,\cdot)}}^{(t+1)}\|^2 \leq M\xi^2\|\boldsymbol{p}\|^2 \tag{124}$$

We can also verify that this claim holds when $t = 1$. The proof of Claim 1 finishes here.

### E.2 PROOF OF CLAIM 2 OF LEMMA 2

The proof of Claim 2 is one of the most challenging parts in our paper, since that we need to deal with the complicated softmax function. The core idea of proof is that we pay more attention on the changes of label-relevant features in the gradient update, which should be the most crucial factor based on our data model. We then show the attention map converges to be sparse as long as the data model satisfies (8).

We first study the gradient of $\boldsymbol{W}_Q^{(t+1)}$ in part (a) and the gradient of $\boldsymbol{W}_K^{(t+1)}$ in part (b).
(a) By (232), we have

$$\eta\frac{1}{B}\sum_{n\in\mathcal{B}_b}\frac{\partial\mathbf{Loss}(\boldsymbol{X}^n, y^n)}{\partial\boldsymbol{W}_Q}$$
$$=\eta\frac{1}{B}\sum_{n\in\mathcal{B}_b}\frac{\partial\mathbf{Loss}(\boldsymbol{X}^n, y^n)}{\partial F(\boldsymbol{X}^n)}\frac{F(\boldsymbol{X}^n)}{\partial\boldsymbol{W}_Q}$$
$$=\eta\frac{1}{B}\sum_{n\in\mathcal{B}_b}(-y^n)\frac{1}{|\mathcal{S}^n|}\sum_{l\in\mathcal{S}^n}\sum_{i=1}^m a_{(l)_i}\mathbb{1}[\boldsymbol{W}_{O_{(i,\cdot)}}\boldsymbol{W}_V\boldsymbol{X}\mathrm{softmax}(\boldsymbol{X}^{n\top}\boldsymbol{W}_K^\top\boldsymbol{W}_Q\boldsymbol{x}_l^n) \geq 0]$$
$$\cdot\left(\boldsymbol{W}_{O_{(i,\cdot)}}\sum_{s\in\mathcal{S}^n}\boldsymbol{W}_V\boldsymbol{x}_s^n\mathrm{softmax}(\boldsymbol{x}_s^{n\top}\boldsymbol{W}_K^\top\boldsymbol{W}_Q\boldsymbol{x}_l^n)\right.$$
$$\left.\cdot\sum_{r\in\mathcal{S}^n}\mathrm{softmax}(\boldsymbol{x}_r^{n\top}\boldsymbol{W}_K^\top\boldsymbol{W}_Q\boldsymbol{x}_l^n)\boldsymbol{W}_K(\boldsymbol{x}_s^n - \boldsymbol{x}_r^n)\boldsymbol{x}_l^{n\top}\right) \tag{125}$$
$$=\eta\frac{1}{B}\sum_{n\in\mathcal{B}_b}(-y^n)\frac{1}{|\mathcal{S}^n|}\sum_{l\in\mathcal{S}^n}\sum_{i=1}^m a_{(l)_i}\mathbb{1}[\boldsymbol{W}_{O_{(i,\cdot)}}\boldsymbol{W}_V\boldsymbol{X}^n\mathrm{softmax}(\boldsymbol{X}^{n\top}\boldsymbol{W}_K^\top\boldsymbol{W}_Q\boldsymbol{x}_l^n) \geq 0]$$
$$\cdot\left(\boldsymbol{W}_{O_{(i,\cdot)}}\sum_{s\in\mathcal{S}^n}\boldsymbol{W}_V\boldsymbol{x}_s^n\mathrm{softmax}(\boldsymbol{x}_s^{n\top}\boldsymbol{W}_K^\top\boldsymbol{W}_Q\boldsymbol{x}_l^n)\right.$$
$$\left.\cdot(\boldsymbol{W}_K\boldsymbol{x}_s^n - \sum_{r\in\mathcal{S}^n}\mathrm{softmax}(\boldsymbol{x}_r^{n\top}\boldsymbol{W}_K^\top\boldsymbol{W}_Q\boldsymbol{x}_l^n)\boldsymbol{W}_K\boldsymbol{x}_r^n)\boldsymbol{x}_l^{n\top}\right)$$

For $r, l \in \mathcal{S}_1^n$, by (43) we have

$$\mathrm{softmax}(\boldsymbol{x}_j^{n\top}\boldsymbol{W}_K^{(t)}\boldsymbol{W}_Q^{(t)}\boldsymbol{x}_l^n) \gtrsim \frac{e^{\|\boldsymbol{q}_1(t)\|^2 - (\delta+\tau)\|\boldsymbol{q}_1(t)\|}}{|\mathcal{S}_1^n|e^{\|\boldsymbol{q}_1(t)\|^2 - (\delta+\tau)\|\boldsymbol{q}_1(t)\|} + (|\mathcal{S}^n| - |\mathcal{S}_1^n|)} \tag{126}$$

For $r \notin \mathcal{S}_1^n$ and $l \in \mathcal{S}_1^n$, we have

$$\mathrm{softmax}(\boldsymbol{x}_j^{n\top}\boldsymbol{W}_K^{(t+1)}{}^\top\boldsymbol{W}_Q^{(t+1)}\boldsymbol{x}_l^n) \lesssim \frac{1}{|\mathcal{S}_1^n|e^{(1+K(t))\|\boldsymbol{q}_1(t)\|^2 - (\delta+\tau)\|\boldsymbol{q}_1(t)\|} + (|\mathcal{S}^n| - |\mathcal{S}_1^n|)} \tag{127}$$

Therefore, for $s, r, l \in \mathcal{S}_1^n$, let

$$\boldsymbol{W}_K^{(t)} \boldsymbol{x}_s^n - \sum_{r \in \mathcal{S}^n} \text{softmax}(\boldsymbol{x}_r^{n\top} \boldsymbol{W}_K^{(t)\top} \boldsymbol{W}_Q^{(t)} \boldsymbol{x}_l^n) \boldsymbol{W}_K^{(t)} \boldsymbol{x}_r^n := \beta_1^n(t) \boldsymbol{q}_1(t) + \boldsymbol{\beta}_2^n(t), \tag{128}$$

where

$$\beta_1^n(t) \gtrsim \frac{|\mathcal{S}^n| - |\mathcal{S}_1^n|}{|\mathcal{S}_1^n| e^{\|\boldsymbol{q}_1(t)\|^2 + (\delta+\tau)\|\boldsymbol{q}_1(t)\|} + |\mathcal{S}^n| - |\mathcal{S}_1^n|}$$
$$:= \phi_n(t)(|\mathcal{S}^n| - |\mathcal{S}_1^n|). \tag{129}$$

$$\beta_1^n(t) \lesssim \nu_n(t)(|\mathcal{S}^n| - |\mathcal{S}_1^n|) \lesssim e^{2(\tau+\delta)\|\boldsymbol{q}_1(t)\|} \phi_n(t)(|\mathcal{S}^n| - |\mathcal{S}_1^n|) \le \phi_n(t)(|\mathcal{S}^n| - |\mathcal{S}_1^n|) \tag{130}$$

where the last inequality holds when the final iteration $\log T \le \Theta(1)$.

$$\boldsymbol{\beta}_2^n(t) \approx \Theta(1) \cdot \boldsymbol{o}_j^n(t) + Q_e(t) \boldsymbol{r}_2(t) + \sum_{l=3}^{M} \gamma_l' \boldsymbol{r}_l(t) - \sum_{a=1}^{M} \sum_{r \in \mathcal{S}_l^n} \text{softmax}(\boldsymbol{x}_r^\top \boldsymbol{W}_K^{(t)\top} \boldsymbol{W}_Q^{(t)} \boldsymbol{x}_l) \boldsymbol{r}_a(t)$$

$$= \Theta(1) \cdot \boldsymbol{o}_j^n(t) + \sum_{l=1}^{M} \zeta_l' \boldsymbol{r}_l(t) \tag{131}$$

for some $Q_e(t) > 0$ and $\gamma_l' > 0$. Here

$$|\zeta_l'| \le \beta_1^n(t) \frac{|\mathcal{S}_l^n|}{|\mathcal{S}^n| - |\mathcal{S}_1^n|} \tag{132}$$

for $l \ge 2$. Note that $|\zeta_l'| = 0$ if $|\mathcal{S}^n| = |\mathcal{S}_1^n|, l \ge 2$.
Therefore, for $i \in \mathcal{W}(t)$,

$$\boldsymbol{W}_{O_{(i,\cdot)}}^{(t)} \sum_{s \in \mathcal{S}^n} \boldsymbol{W}_V^{(t)} \boldsymbol{x}_s^n \text{softmax}(\boldsymbol{x}_s^{n\top} \boldsymbol{W}_K^{(t)\top} \boldsymbol{W}_Q^{(t)} \boldsymbol{x}_l^n)$$

$$= \boldsymbol{W}_{O_{(i,\cdot)}}^{(t)} \Big( \sum_{s \in \mathcal{S}_1} \boldsymbol{p}_s \text{softmax}(\boldsymbol{x}_s^{n\top} \boldsymbol{W}_K^{(t)\top} \boldsymbol{W}_Q^{(t)} \boldsymbol{x}_l^n) + \boldsymbol{z}(t) + \sum_{l \ne s} W_l(u) \boldsymbol{p}_l$$

$$- \eta \sum_{b=1}^{t} \Big( \sum_{j \in \mathcal{W}(b)} V_j(b) \boldsymbol{W}_{O_{(j,\cdot)}}^{(b)\top} + \sum_{j \notin \mathcal{W}(b)} V_j(b) \lambda \boldsymbol{W}_{O_{(j,\cdot)}}^{(b)\top} \Big) \Big)$$

$$\gtrsim \frac{1}{Bt} \sum_{b=1}^{t} \sum_{n \in \mathcal{B}_b} \frac{\eta^2 btm |\mathcal{S}_1^n|}{|\mathcal{S}^n| a^2} (p_n(b))^2 \|\boldsymbol{p}_1\|^2 - ((\sigma+\tau)) \frac{\eta^2 t^2 m}{a^2} \|\boldsymbol{p}_1\|^2 + \xi(1 - (\sigma+\tau)) - ((\sigma+\tau))$$

$$\cdot \sqrt{M} \xi \|\boldsymbol{p}_1\| - \xi \|\boldsymbol{p}_1\| + \eta \frac{1}{B} \sum_{b=1}^{t} \sum_{n \in \mathcal{B}_b} \frac{|\mathcal{S}_1^n| p_n(b) m}{|\mathcal{S}^n| aM} (1 - \epsilon_m - \frac{(\sigma+\tau)M}{\pi}) (\frac{\eta^2 t^2 m}{a^2})^2 \|\boldsymbol{p}_1\|^2$$

$$- \eta \frac{1}{B} \sum_{b=1}^{t} \sum_{n \in \mathcal{B}_b} \frac{|\mathcal{S}_2^n| p_n(b) m}{|\mathcal{S}^n| aM} (1 - \epsilon_m - \frac{(\sigma+\tau)M}{\pi}) (\frac{\eta^2 t^2 m}{a^2}) \xi \|\boldsymbol{p}_1\|^2 - \frac{\eta^2 \lambda \xi \sqrt{M} t^2 m}{\sqrt{B} a^2} \|\boldsymbol{p}_1\|^2$$

$$> 0 \tag{133}$$

where the first step is by (27) and the second step is a combination of (32) to (35). The final step holds as long as

$$\sigma + \tau \lesssim O(1), \tag{134}$$

and

$$B \ge \Big( \frac{\lambda \xi \sqrt{M}}{\epsilon \frac{1}{B} \sum_{n \in \mathcal{B}_b} \frac{|\mathcal{S}_1^n|}{|\mathcal{S}^n|} (p_n(t))^2} \Big)^2 \tag{135}$$

Then we study how large the coefficient of $\boldsymbol{q}_1(t)$ in (125).
If $s \in \mathcal{S}_1^n$, by basic computation given (32) to (35),

$$\boldsymbol{W}_{O_{(i,\cdot)}}^{(t)} \boldsymbol{W}_V^{(t)} \boldsymbol{x}_s^n \mathrm{softmax}(\boldsymbol{x}_s^{n\top} \boldsymbol{W}_K^{(t)\top} \boldsymbol{W}_Q^{(t)} \boldsymbol{x}_l^n)$$

$$\gtrsim \Big(\frac{\eta^2 t^2 m}{a^2}(\frac{1}{Bt}\sum_{b=1}^{t}\sum_{n\in\mathcal{B}_b}\frac{|\mathcal{S}_1^n|b}{|\mathcal{S}^n|t}p_n(t)-(\sigma+\tau))\|\boldsymbol{p}_1\|^2-((\sigma+\tau))\sqrt{M}\xi\|\boldsymbol{p}_1\|-\xi\|\boldsymbol{p}_1\|+\eta\sum_{b=1}^{t}\frac{1}{B}$$

$$\cdot \sum_{n\in\mathcal{B}_b}\frac{|\mathcal{S}_1^n|p_n(t)m}{|\mathcal{S}^n|aM}(1-\epsilon_m-\frac{(\sigma+\tau)M}{\pi})(\frac{1}{Bt}\sum_{b=1}^{t}\sum_{n\in\mathcal{B}_b}\frac{\eta^2 tb|\mathcal{S}_1^n|}{m|\mathcal{S}^n|}p_n(b))^2\|\boldsymbol{p}_1\|^2-\frac{\eta^2\lambda\xi\sqrt{M}t^2m}{\sqrt{B}a^2}$$

$$\cdot \|\boldsymbol{p}_1\|^2-\eta\sum_{b=1}^{t}\frac{1}{B}\sum_{n\in\mathcal{B}_b}\frac{|\mathcal{S}_2^n|p_n(t)m}{|\mathcal{S}^n|aM}(1-\epsilon_m-\frac{(\sigma+\tau)M}{\pi})(\frac{\eta^2 t^2 m}{a^2})^2(\sigma+\tau)\|\boldsymbol{p}_1\|^2\Big)\frac{p_n(t)}{|\mathcal{S}_1^n|}$$

$$\gtrsim \frac{1}{Bt}\sum_{b=1}^{t}\sum_{n\in\mathcal{B}_b}\frac{\eta^2 t^2 m}{a^2}(\frac{|\mathcal{S}_1^n|b}{|\mathcal{S}^n|t}p_n(t)-(\sigma+\tau))\|\boldsymbol{p}_1\|^2\frac{p_n(t)}{|\mathcal{S}_1^n|}$$

$$+\eta\sum_{b=1}^{t}\frac{1}{B}\sum_{n\in\mathcal{B}_b}\frac{|\mathcal{S}_1^n|p_n(b)m}{|\mathcal{S}^n|aM}(1-\epsilon_m-\frac{(\sigma+\tau)M}{\pi})(\frac{1}{Bt}\sum_{b=1}^{t}\sum_{n\in\mathcal{B}_b}\frac{\eta^2 tbm|\mathcal{S}_1^n|}{a^2|\mathcal{S}^n|}p_n(b))^2\|\boldsymbol{p}_1\|^2\frac{p_n(t)}{|\mathcal{S}_1^n|}$$

(136)

where the last step is by (134) and (135).
If $s \in \mathcal{S}_2^n$, from (36) to (39), we have

$$\boldsymbol{W}_{O_{(i,\cdot)}}^{(t)} \boldsymbol{W}_V^{(t)} \boldsymbol{x}_s^n \mathrm{softmax}(\boldsymbol{x}_s^{n\top} \boldsymbol{W}_K^{(t)\top} \boldsymbol{W}_Q^{(t)} \boldsymbol{x}_l^n)$$

$$\lesssim (\xi\|\boldsymbol{p}\|+\frac{1}{Bt}\sum_{b=1}^{t}\sum_{n\in\mathcal{B}_b}\frac{|\mathcal{S}_1^n|p_n(b)\eta bm}{|\mathcal{S}^n|aM}(1-\epsilon_m-\frac{(\sigma+\tau)M}{\pi})(\frac{\eta^2 t^2 m}{a^2})^2(\sigma+\tau)\|\boldsymbol{p}\|$$

$$+\frac{\eta^2\lambda\sqrt{M}\xi t^2 m}{\sqrt{B}a^2}\|\boldsymbol{p}_1\|^2+((\sigma+\tau))(\sqrt{M}\xi+\frac{\eta^2 t^2 m}{a^2})+\frac{1}{Bt}\sum_{b=1}^{t}\sum_{n\in\mathcal{B}_b}\frac{|\mathcal{S}_2^n|p_n(t)\eta bm}{|\mathcal{S}^n|aM}$$

$$\cdot(1-\epsilon_m-\frac{(\sigma+\tau)M}{\pi})(\frac{\eta^2 t^2 m}{a^2})^2\|\boldsymbol{p}_1\|^2)\phi_n(t)$$

(137)

If $i \in \mathcal{W}(t)$ and $s \notin (\mathcal{S}_1^n \cup \mathcal{S}_2^n)$,

$$\boldsymbol{W}_{O_{(i,\cdot)}}^{(t)} \boldsymbol{W}_V^{(t)} \boldsymbol{x}_s^n \mathrm{softmax}(\boldsymbol{x}_s^{n\top} \boldsymbol{W}_K^{(t)\top} \boldsymbol{W}_Q^{(t)} \boldsymbol{x}_l^n)$$

$$\lesssim (\xi\|\boldsymbol{p}\|+\frac{\eta^2\lambda\sqrt{M}\xi t^2 m}{\sqrt{B}a^2}\|\boldsymbol{p}\|^2+((\sigma+\tau))(\sqrt{M}\xi+\frac{\eta^2 t^2 m}{a^2})\|\boldsymbol{p}\|$$

$$+\eta\frac{1}{Bt}\sum_{b=1}^{t}\sum_{n\in\mathcal{B}_b}\frac{(|\mathcal{S}_2^n|+|\mathcal{S}_1^n|)p_n(b)\eta bm}{|\mathcal{S}^n|aM}(1-\epsilon_m-\frac{(\sigma+\tau)M}{\pi})\frac{\eta^2 t^2 m}{a^2}\xi\|\boldsymbol{p}_1\|^2)\phi_n(t)$$

(138)

by (40) to (42).
Hence, for $i \in \mathcal{W}(t)$, $j \in \mathcal{S}_1^g$, combining (129) and (136), we have

$$\boldsymbol{W}_{O_{(i,\cdot)}}^{(t)} \sum_{s\in\mathcal{S}^n} \boldsymbol{W}_V^{(t)} \boldsymbol{x}_s^n \mathrm{softmax}(\boldsymbol{x}_s^{n\top} \boldsymbol{W}_K^{(t)\top} \boldsymbol{W}_Q^{(t)} \boldsymbol{x}_l^n)\boldsymbol{q}_1(t)^\top$$

$$\cdot(\boldsymbol{W}_K^{(t)}\boldsymbol{x}_s^n-\sum_{r=1}^{L}\mathrm{softmax}(\boldsymbol{x}_r^{n\top}\boldsymbol{W}_K^{(t)\top}\boldsymbol{W}_Q^{(t)}\boldsymbol{x}_l^n)\boldsymbol{W}_K^{(t)}\boldsymbol{x}_r^n)\boldsymbol{x}_l^{n\top}\boldsymbol{x}_j^g$$

$$\gtrsim \Big(\frac{1}{Bt}\sum_{b=1}^{t}\sum_{n\in\mathcal{B}_b}\frac{|\mathcal{S}_1^n|p_n(b)\eta tm}{|\mathcal{S}^n|aM}(1-\epsilon_m-\frac{(\sigma+\tau)M}{\pi})(\frac{1}{Bt}\sum_{b=1}^{t}\sum_{n\in\mathcal{B}_b}\frac{\eta^2 tbm|\mathcal{S}_1^n|}{a^2|\mathcal{S}^n|}p_n(b))^2\|\boldsymbol{p}_1\|^2 p_n(t)$$

$$+\frac{1}{Bt}\sum_{b=1}^{t}\sum_{n\in\mathcal{B}_b}\frac{\eta^2 t^2 m}{a^2}(\frac{|\mathcal{S}_1^n|b}{|\mathcal{S}^n|t}p_n(b)-(\sigma+\tau))\|\boldsymbol{p}_1\|^2 p_n(t)\Big)\phi_n(t)(|\mathcal{S}^n|-|\mathcal{S}_1^n|)\|\boldsymbol{q}_1(t)\|^2$$

(139)

For $i \in \mathcal{U}(t)$ and $l \in \mathcal{S}_1^n, j \in \mathcal{S}_1^g$

$$
\boldsymbol{W}_{O_{(i,\cdot)}}^{(t)} \sum_{s \in \mathcal{S}^n} \boldsymbol{W}_V^{(t)} \boldsymbol{x}_s^n \mathrm{softmax}(\boldsymbol{x}_s^{n\top} \boldsymbol{W}_K^{(t)\top} \boldsymbol{W}_Q^{(t)} \boldsymbol{x}_l^n) \boldsymbol{q}_1(t)^\top
$$

$$
\cdot (\boldsymbol{W}_K^{(t)} \boldsymbol{x}_s^n - \sum_{r \in \mathcal{S}^n} \mathrm{softmax}(\boldsymbol{x}_r^{n\top} \boldsymbol{W}_K^{(t)\top} \boldsymbol{W}_Q^{(t)} \boldsymbol{x}_l^n) \boldsymbol{W}_K^{(t)} \boldsymbol{x}_r^n) \boldsymbol{x}_l^{n\top} \boldsymbol{x}_j^g
$$

$$
\lesssim \frac{1}{Bt} \sum_{b=1}^t \sum_{n \in \mathcal{B}_b} \frac{|\mathcal{S}_2^n| p_n(b) \eta t m}{|\mathcal{S}^n| a M} (1 - \epsilon_m - \frac{(\sigma + \tau)M}{\pi}) (\frac{\eta^2 t^2 m}{a^2})^2 \|\boldsymbol{p}_1\|^2 \phi_n(t) |\mathcal{S}_2^n| \beta_1(t) \|\boldsymbol{q}_1(t)\|^2
$$

$$
+ \frac{1}{Bt} \sum_{b=1}^t \sum_{n \in \mathcal{B}_b} \frac{|\mathcal{S}_1^n| p_n(b) \eta t m}{|\mathcal{S}^n| a M} (1 - \epsilon_m - \frac{(\sigma + \tau)M}{\pi}) (\frac{\eta^2 t^2 m}{a^2})^2 (\sigma + \tau) \|\boldsymbol{p}\|^2 \phi_n(t) |\mathcal{S}_2^n|
$$

$$
\cdot \beta_1(t) \|\boldsymbol{q}_1(t)\|^2
$$

(140)

For $i \notin (\mathcal{W}(t) \cup \mathcal{U}(t))$ and $l \in \mathcal{S}_1^n, j \in \mathcal{S}_1^g$,

$$
\boldsymbol{W}_{O_{(i,\cdot)}}^{(t)} \sum_{s \in \mathcal{S}^n} \boldsymbol{W}_V^{(t)} \boldsymbol{x}_s^n \mathrm{softmax}(\boldsymbol{x}_s^{n\top} \boldsymbol{W}_K^{(t)\top} \boldsymbol{W}_Q^{(t)} \boldsymbol{x}_l^n) \boldsymbol{q}_1(t)^\top
$$

$$
\cdot (\boldsymbol{W}_K^{(t)} \boldsymbol{x}_s^n - \sum_{r \in \mathcal{S}^n} \mathrm{softmax}(\boldsymbol{x}_r^{n\top} \boldsymbol{W}_K^{(t)\top} \boldsymbol{W}_Q^{(t)} \boldsymbol{x}_l^n) \boldsymbol{x}_r^n) \boldsymbol{x}_l^{n\top} \boldsymbol{x}_j^g
$$

$$
\lesssim (\xi \|\boldsymbol{p}\| + ((\sigma + \tau))(\frac{\eta^2 t^2 m}{a^2} + \sqrt{M}\xi) \|\boldsymbol{p}_1\|^2 + \frac{1}{Bt} \sum_{b=1}^t \sum_{n \in \mathcal{B}_b} \frac{(|\mathcal{S}_1^n| + |\mathcal{S}_2^n|) p_n(b) \eta t m}{a M |\mathcal{S}^n|}
$$

(141)

$$
\cdot (1 - \epsilon_m - \frac{(\sigma + \tau)M}{\pi}) \xi \frac{\eta^2 t^2 m}{a^2} \|\boldsymbol{p}_1\| + \frac{\eta^2 t^2 \lambda \xi \sqrt{M} m \|\boldsymbol{p}\|^2}{\sqrt{B} a^2}) \cdot \beta_1(t) \|\boldsymbol{q}_1(t)\|^2
$$

To study the case when $l \notin \mathcal{S}_1^n$ for all $n \in [N]$, we need to check all other $l$'s. Recall that we focus on the coefficient of $\boldsymbol{q}_1(t)$ in this part. Based on the computation in (137) and (138), we know that the contribution of coefficient from non-discriminative patches is no more than that from discriminative patches, i.e., for $l \notin (\mathcal{S}_1^n \cup \mathcal{S}_2^n)$, $n \in [N]$ and $k \in \mathcal{S}_1^n$,

$$
\left| \boldsymbol{W}_{O_{(i,\cdot)}}^{(t)} \sum_{s \in \mathcal{S}^n} \boldsymbol{W}_V^{(t)} \boldsymbol{x}_s^n \mathrm{softmax}(\boldsymbol{x}_s^{n\top} \boldsymbol{W}_K^{(t)\top} \boldsymbol{W}_Q^{(t)} \boldsymbol{x}_l^n) \boldsymbol{q}_1(t)^\top \right.
$$

$$
\cdot (\boldsymbol{W}_K^{(t)} \boldsymbol{x}_s^n - \sum_{r \in \mathcal{S}^n} \mathrm{softmax}(\boldsymbol{W}_K^{(t)} \boldsymbol{x}_r^{n\top} \boldsymbol{W}_K^{(t)\top} \boldsymbol{W}_Q^{(t)} \boldsymbol{x}_l^n) \boldsymbol{W}_K^{(t)} \boldsymbol{x}_r^n) \boldsymbol{x}_l^{n\top} \boldsymbol{x}_j^g \Big|
$$

(142)

$$
\leq \left| \boldsymbol{W}_{O_{(i,\cdot)}}^{(t)} \sum_{s \in \mathcal{S}^n} \boldsymbol{W}_V^{(t)} \boldsymbol{x}_s^n \mathrm{softmax}(\boldsymbol{x}_s^{n\top} \boldsymbol{W}_K^{(t)\top} \boldsymbol{W}_Q^{(t)} \boldsymbol{x}_k^n) \boldsymbol{q}_1(t)^\top \right.
$$

$$
\cdot (\boldsymbol{W}_K^{(t)} \boldsymbol{x}_s^n - \sum_{r \in \mathcal{S}^n} \mathrm{softmax}(\boldsymbol{W}_K^{(t)} \boldsymbol{x}_r^{n\top} \boldsymbol{W}_K^{(t)\top} \boldsymbol{W}_Q^{(t)} \boldsymbol{x}_l^n) \boldsymbol{W}_K^{(t)} \boldsymbol{x}_r^n) \boldsymbol{x}_k^{n\top} \boldsymbol{x}_j^g \Big|
$$

Similar to (139), we have that for $l \in \mathcal{S}_2^n$, $j \in \mathcal{S}_1^g$, and $i \in \mathcal{U}(t)$,

$$
\boldsymbol{W}_{O_{(i,\cdot)}}^{(t)} \sum_{s \in \mathcal{S}^n} \boldsymbol{W}_V^{(t)} \boldsymbol{x}_s^n \mathrm{softmax}(\boldsymbol{x}_s^{n\top} \boldsymbol{W}_K^{(t)\top} \boldsymbol{W}_Q^{(t)} \boldsymbol{x}_l^n) \boldsymbol{q}_1(t)^\top
$$

$$
\cdot (\boldsymbol{W}_K^{(t)} \boldsymbol{x}_s^n - \sum_{r \in \mathcal{S}^n} \mathrm{softmax}(\boldsymbol{W}_K^{(t)} \boldsymbol{x}_r^{n\top} \boldsymbol{W}_K^{(t)\top} \boldsymbol{W}_Q^{(t)} \boldsymbol{x}_l^n) \boldsymbol{W}_K^{(t)} \boldsymbol{x}_r^n) \boldsymbol{x}_l^{n\top} \boldsymbol{x}_j^g
$$

$$
\lesssim \frac{1}{Bt} \sum_{b=1}^t \sum_{n \in \mathcal{B}_b} \frac{|\mathcal{S}_2^n| p_n(t) \eta t m}{|\mathcal{S}^n| a M} (1 - \epsilon_m - \frac{(\sigma + \tau) M}{\pi})(\frac{\eta^2 t^2 m}{a^2})^2 \|\boldsymbol{p}_2\|^2 \cdot \beta_1(t) \lambda \frac{|\mathcal{S}_\#^n|}{|\mathcal{S}^n| - |\mathcal{S}_*^n|}
$$

$$
\cdot \|\boldsymbol{q}_1(t)\|^2 + \frac{1}{Bt} \sum_{b=1}^t \sum_{n \in \mathcal{B}_b} \frac{|\mathcal{S}_1^n| p_n(b) \eta t m}{|\mathcal{S}^n| a M} (1 - \epsilon_m - \frac{(\sigma + \tau) M}{\pi})(\frac{\eta^2 t^2 m}{a^2})^2 (\sigma + \tau) \|\boldsymbol{p}\|^2 \beta_1(t)
$$

$$
\cdot \lambda \frac{|\mathcal{S}_\#^n|}{|\mathcal{S}^n| - |\mathcal{S}_*^n|} \|\boldsymbol{q}_1(t)\|^2 \tag{143}
$$

Therefore, by the update rule,

$$
\begin{aligned}
\boldsymbol{W}_Q^{(t+1)} \boldsymbol{x}_j &= \boldsymbol{W}_Q^{(t)} \boldsymbol{x}_j - \eta \Big(\frac{\partial L}{\partial \boldsymbol{W}_Q} \Big| \boldsymbol{W}_Q^{(t)}\Big) \boldsymbol{x}_j \\
&= \boldsymbol{r}_1(t) + K(t) \boldsymbol{q}_1(t) + \Theta(1) \cdot \boldsymbol{n}_j(t) + \sum_{b=0}^{t-1} |K_e(b)| \boldsymbol{q}_2(b) + \sum_{l=3}^M \gamma_l' \boldsymbol{q}_l(t) \\
&= (1 + K(t)) \boldsymbol{q}_1(t) + \Theta(1) \cdot \boldsymbol{n}_j(t) + \sum_{b=0}^{t-1} |K_e(b)| \boldsymbol{q}_2(b) + \sum_{l=3}^M \gamma_l' \boldsymbol{q}_l(t)
\end{aligned} \tag{144}
$$

where the last step is by the condition that

$$
\boldsymbol{q}_1(t) = k_1(t) \cdot \boldsymbol{r}_1(t), \tag{145}
$$

and

$$
\boldsymbol{q}_2(t) = k_2(t) \cdot \boldsymbol{r}_2(t) \tag{146}
$$

for $k_1(t) > 0$ and $k_2(t) > 0$ from induction, i.e., $\boldsymbol{q}_1(t)$ and $\boldsymbol{r}_1(t)$, $\boldsymbol{q}_1(t)$ and $\boldsymbol{r}_1(t)$ are in the same direction, respectively. We also have

$$
\begin{aligned}
K(t)\\
\gtrsim \eta\Big(&\frac{1}{Bt}\sum_{b=1}^{t}\sum_{n\in\mathcal{B}_b}\frac{|\mathcal{S}_1^n|p_n(b)\eta t m}{|\mathcal{S}^n|aM}(1-\epsilon_m-\frac{(\sigma+\tau)M}{\pi})(\frac{1}{Bt}\sum_{b=1}^{t}\sum_{n\in\mathcal{B}_b}\frac{\eta^2 tbm|\mathcal{S}_1^n|}{a^2|\mathcal{S}^n|}p_n(b))^2\|\boldsymbol{p}_1\|^2\\
&\cdot p_n(t)+\frac{1}{Bt}\sum_{b=1}^{t}\sum_{n\in\mathcal{B}_b}\frac{\eta^2 t^2 m}{a^2}(\frac{b|\mathcal{S}_1^n|}{t|\mathcal{S}^n|}p_n(t)-(\sigma+\tau))\|\boldsymbol{p}_1\|^2 p_n(t)\Big)\phi_n(t)(|\mathcal{S}^n|-|\mathcal{S}_1^n|)\|\boldsymbol{q}_1(t)\|^2\\
&-\eta\frac{1}{Bt}\sum_{b=1}^{t}\sum_{n\in\mathcal{B}_b}\frac{|\mathcal{S}_2^n|p_n(b)\eta t m}{|\mathcal{S}^n|aM}(1-\epsilon_m-\frac{(\sigma+\tau)M}{\pi})(\frac{\eta^2 t^2 m}{a^2})^2\|\boldsymbol{p}_1\|^2\phi_n(t)|\mathcal{S}_2|\beta_1(t)\|\boldsymbol{q}_1(t)\|^2\\
&-\eta\frac{1}{Bt}\sum_{b=1}^{t}\sum_{n\in\mathcal{B}_b}\frac{|\mathcal{S}_1^n|p_n(b)\eta t m}{|\mathcal{S}^n|aM}(1-\epsilon_m-\frac{(\sigma+\tau)M}{\pi})(\frac{\eta^2 t^2 m}{a^2})^2(\sigma+\tau)\|\boldsymbol{p}\|^2\phi_n(t)|\mathcal{S}_2^n|\beta_1(t)\\
&\cdot\|\boldsymbol{q}_1(t)\|^2-\eta(\xi\|\boldsymbol{p}\|+((\sigma+\tau))(\frac{\eta^2 t^2 m}{a^2}+\sqrt{M}\xi)\|\boldsymbol{p}_1\|^2+\frac{1}{Bt}\sum_{b=1}^{t}\sum_{n\in\mathcal{B}_b}\frac{|\mathcal{S}_1^n|p_n(b)\eta t m}{aM|\mathcal{S}^n|}\\
&\cdot(1-\epsilon_m-\frac{(\sigma+\tau)M}{\pi})\xi\frac{\eta^2 t^2}{m}\|\boldsymbol{p}_1\|+\frac{\eta t\lambda\xi\sqrt{M}m\|\boldsymbol{p}\|^2}{\sqrt{B}a^2})\cdot\beta_1(t)\|\boldsymbol{q}_1(t)\|^2\\
&-\eta\frac{1}{Bt}\sum_{b=1}^{t}\sum_{n\in\mathcal{B}_b}\frac{|\mathcal{S}_2^n|p_n(b)\eta t m}{|\mathcal{S}^n|aM}(1-\epsilon_m-\frac{(\sigma+\tau)M}{\pi})(\frac{\eta^2 tbm}{a^2})^2\|\boldsymbol{p}_2\|^2\cdot\beta_1(t)\lambda\frac{|\mathcal{S}_\#^n|}{|\mathcal{S}^n|-|\mathcal{S}_*^n|}\\
&\cdot\|\boldsymbol{q}_1(t)\|^2-\eta\frac{1}{Bt}\sum_{b=1}^{t}\sum_{n\in\mathcal{B}_b}\frac{|\mathcal{S}_1^n|p_n(b)\eta t m}{|\mathcal{S}^n|aM}(1-\epsilon_m-\frac{(\sigma+\tau)M}{\pi})(\frac{\eta^2 t^2 m}{a^2})^2(\sigma+\tau)\|\boldsymbol{p}\|^2\beta_1(t)\\
&\cdot\lambda\frac{|\mathcal{S}_\#^n|}{|\mathcal{S}^n|-|\mathcal{S}_*^n|}\|\boldsymbol{q}_1(t)\|^2\\
\gtrsim \eta\Big(&\frac{1}{Bt}\sum_{b=1}^{t}\sum_{n\in\mathcal{B}_b}\frac{|\mathcal{S}_1^n|p_n(b)\eta t m}{|\mathcal{S}^n|aM}(1-\epsilon_m-\frac{(\sigma+\tau)M}{\pi})(\frac{1}{Bt}\sum_{b=1}^{t}\sum_{n\in\mathcal{B}_b}\frac{\eta^2 tb|\mathcal{S}_1^n|m}{a^2|\mathcal{S}^n|}p_n(b))^2\|\boldsymbol{p}_1\|^2\\
&\cdot p_n(t)+\frac{1}{Bt}\sum_{b=1}^{t}\sum_{n\in\mathcal{B}_b}\frac{\eta^2 t^2 m}{a^2}(\frac{b|\mathcal{S}_1^n|}{t|\mathcal{S}^n|}p_n(t)-(\sigma+\tau))\|\boldsymbol{p}_1\|^2 p_n(t)\Big)\phi_n(t)(|\mathcal{S}^n|-|\mathcal{S}_1^n|)\|\boldsymbol{q}_1(t)\|^2\\
> &0
\end{aligned}
$$
$$\tag{147}$$

$$
|\gamma_l'| \lesssim \frac{1}{B}\sum_{n\in\mathcal{B}_b}K(t)\cdot\frac{|\mathcal{S}_l^n|}{|\mathcal{S}^n|-|\mathcal{S}_1^n|}
\tag{148}
$$

$$
|K_e(t)| \lesssim \frac{1}{B}\sum_{n\in\mathcal{B}_b}K(t)\cdot\frac{|\mathcal{S}_2^n|}{|\mathcal{S}^n|-|\mathcal{S}_1^n|}
\tag{149}
$$

as long as

$$
\begin{aligned}
&\Big(\frac{1}{Bt}\sum_{b=1}^{t}\sum_{n\in\mathcal{B}_b}\epsilon_{\mathcal{S}}\frac{\eta^2 t^2 m}{a^2}p_n(t)(\frac{|\mathcal{S}_1^n|b}{|\mathcal{S}^n|t}p_n(b)-(\sigma+\tau))\|\boldsymbol{p}_1\|^2 + \frac{1}{Bt}\sum_{b=1}^{t}\sum_{n\in\mathcal{B}_b}\frac{|\mathcal{S}_1^n|p_n(t)\eta t m}{|\mathcal{S}^n|aM}\\
&\cdot(1-\epsilon_m-\frac{(\sigma+\tau)M}{\pi})(\frac{1}{Bt}\sum_{b=1}^{t}\sum_{n\in\mathcal{B}_b}\frac{\eta^2 t b m|\mathcal{S}_1^n|}{a^2|\mathcal{S}^n|}p_n(t))^2\|\boldsymbol{p}_1\|^2 p_n(t)\Big)\phi_n(t)\\
&\cdot(|\mathcal{S}^n|-|\mathcal{S}_1^n|)\|\boldsymbol{q}_1(t)\|^2\\
\gtrsim\ &\frac{1}{Bt}\sum_{b=1}^{t}\sum_{n\in\mathcal{B}_b}\frac{|\mathcal{S}_2^n|p_n(b)\eta t m}{|\mathcal{S}^n|aM}(1-\epsilon_m-\frac{(\sigma+\tau)M}{\pi})(\frac{\eta^2 t^2 m}{a^2})^2\|\boldsymbol{p}_2\|^2\cdot\beta_1(t)\lambda\frac{|\mathcal{S}_\#^n|}{|\mathcal{S}^n|-|\mathcal{S}_*^n|}\\
&\cdot\|\boldsymbol{q}_1(t)\|^2 + \frac{1}{Bt}\sum_{b=1}^{t}\sum_{n\in\mathcal{B}_b}\frac{|\mathcal{S}_1^n|p_n(b)\eta t m}{|\mathcal{S}^n|aM}(1-\epsilon_m-\frac{(\sigma+\tau)M}{\pi})(\frac{\eta^2 t^2 m}{a^2})^2(\sigma+\tau)\|\boldsymbol{p}\|^2\beta_1(t)\\
&\cdot\lambda\frac{|\mathcal{S}_\#^n|}{|\mathcal{S}^n|-|\mathcal{S}_*^n|}\|\boldsymbol{q}_1(t)\|^2
\end{aligned}
\tag{150}
$$

To find the sufficient condition for (150), we first compare the first terms of both sides in (150). Note that when

$$\eta t \le O(1),\tag{151}$$

we have

$$\eta^2 t^2 \gtrsim \eta^5 t^5\tag{152}$$

When $|\mathcal{S}^n|>|\mathcal{S}_1^n|$, by (130),

$$\phi_n(t)(|\mathcal{S}^n|-|\mathcal{S}_1^n|)\gtrsim\beta_1^n(t)\tag{153}$$

From Definition 1, we know

$$1\ge p_n(t)\ge p_n(0)=\Theta\big(\frac{|\mathcal{S}_1^n|}{|\mathcal{S}_1^n|e^{-(\delta+\tau)}+|\mathcal{S}^n|-|\mathcal{S}_1^n|}\big)\ge\Theta(e^{-(\delta+\tau)})\tag{154}$$

Moreover,

$$
\begin{aligned}
&\Big|\big(\frac{1}{Bt}\sum_{b=1}^{t}\sum_{n\in\mathcal{B}_b}\frac{|\mathcal{S}_1^n|b}{|\mathcal{S}^n|t}-(\tau+\sigma)\big)e^{-(\delta+\tau)}-(1-(\tau+\sigma))e^{-(\delta+\tau)}\frac{1}{2}\mathbb{E}_{\mathcal{D}}[\frac{|\mathcal{S}_1^n|}{|\mathcal{S}^n|}]\Big|\\
\le\ &\Big|\big(\frac{1}{Bt}\sum_{b=1}^{t}\sum_{n\in\mathcal{B}_b}\frac{|\mathcal{S}_1^n|b}{|\mathcal{S}^n|t}-(\tau+\sigma)\big)e^{-(\delta+\tau)}-e^{-(\delta+\tau)}\mathbb{E}_{\mathcal{D}}[(\frac{1}{Bt}\sum_{b=1}^{t}\sum_{n\in\mathcal{B}_b}\frac{|\mathcal{S}_1^n|b}{|\mathcal{S}^n|t}-(\tau+\sigma))]\Big|\\
&+\Big|e^{-(\delta+\tau)}\mathbb{E}_{\mathcal{D}}[\frac{1}{Bt}\sum_{b=1}^{t}\sum_{n\in\mathcal{B}_b}\frac{|\mathcal{S}_1^n|}{|\mathcal{S}^n|}(\frac{1}{2}-\frac{b}{t})]\Big|+e^{-(\delta+\tau)}(\tau+\sigma)(1-\frac{1}{2}\mathbb{E}_{\mathcal{D}}[\frac{|\mathcal{S}_1^n|}{|\mathcal{S}^n|}])\\
\le\ &e^{-(\delta+\tau)}\sqrt{\frac{\log Bt}{Bt}}+1\cdot\frac{1}{2t}e^{-(\delta+\tau)}+(\tau+\sigma)e^{-(\delta+\tau)}\\
\le\ &e^{-(\delta+\tau)}\sqrt{\frac{\log Bt}{Bt}}+(\tau+\sigma)e^{-(\delta+\tau)}
\end{aligned}
\tag{155}
$$

where the first inequality is by the triangle inequality and the second inequality comes from $|\mathcal{S}_1^n|b/(|\mathcal{S}^n|t)\le 1$. Therefore, a sufficient condition for (150) is

$$\epsilon_{\mathcal{S}}e^{-(\delta+\tau)}(1-(\tau+\sigma))\frac{1}{2}\mathbb{E}_{\mathcal{D}}[\frac{|\mathcal{S}_1^n|}{|\mathcal{S}^n|}]\gtrsim\frac{1}{Bt}\sum_{b=1}^{t}\sum_{n\in\mathcal{B}_b}\frac{|\mathcal{S}_2^n|}{|\mathcal{S}^n|}\ge\mathbb{E}_{\mathcal{D}}[\frac{|\mathcal{S}_2^n|}{|\mathcal{S}^n|}]-\sqrt{\frac{\log Bt}{Bt}}\tag{156}$$

by Hoeffding's inequality in (26). Thus,

$$\alpha_*=:\mathbb{E}_{\mathcal{D}}[\frac{|\mathcal{S}_1^n|}{|\mathcal{S}^n|}]\ge\frac{1-\alpha_{\mathrm{nd}}}{1+\epsilon_{\mathcal{S}}e^{-(\delta+\tau)}(1-(\tau+\sigma))}\tag{157}$$

if

$$Bt \geq \frac{1}{((1 - (\tau + \sigma))e^{-(\delta+\tau)\frac{1}{2}\alpha_*} - \tau)^2} \tag{158}$$

For the second terms on both sides in (150), since $(\tau + \sigma) \leq O(1/M)$, the inequality also holds with the same condition as in $\alpha_*$ and $Bt$.

and

$$\alpha_\# = \mathbb{E}\Big[\frac{|\mathcal{S}_\#^n|}{|\mathcal{S}^n|}\Big] \tag{159}$$

$$\alpha_{nd} = \mathbb{E}\Big[\sum_{l=3}^M \frac{|\mathcal{S}_l^n|}{|\mathcal{S}^n|}\Big] \tag{160}$$

Given that $|\mathcal{S}^n| = \Theta(1)$, (157) is equivalent to (8).

For the second terms on both sides in (150), since $(\sigma + \tau) \leq 1/M$, the inequality also holds with the same condition on $\alpha_*$ and $Bt$.

Note that if $|\mathcal{S}^n| = |\mathcal{S}_1^n|$, we let $|\mathcal{S}_l^n|/(|\mathcal{S}^n| - |\mathcal{S}_1^n|) = 0$ for $l \in [M]$. We use the presentation in (148, 149) above and (170, 171) below for simplicity.

Then we give a brief derivation of $\boldsymbol{W}_Q^{(t+1)}\boldsymbol{x}_j^n$ for $j \notin \mathcal{S}_1^n$ in the following.

To be specific, for $j \in \mathcal{S}_n/(\mathcal{S}_1^n \cup \mathcal{S}_2^n)$,

$$\left\langle \eta\frac{1}{B}\sum_{n\in\mathcal{B}_b}\frac{\partial\mathbf{Loss}(\boldsymbol{X}^n, y^n)}{\partial\boldsymbol{W}_Q^{(t)}}\boldsymbol{x}_j^n, \boldsymbol{q}_1(t)\right\rangle$$

$$\gtrsim \eta\Big(\eta\sum_{b=1}^t\frac{1}{B}\sum_{n\in\mathcal{B}_b}\frac{|\mathcal{S}_1^n|p_n(b)m}{|\mathcal{S}^n|aM}(1 - \epsilon_m - \frac{(\sigma+\tau)M}{\pi})(\frac{1}{Bt}\sum_{b=1}^t\sum_{n\in\mathcal{B}_b}\frac{\eta^2 tbm|\mathcal{S}_1^n|}{a^2|\mathcal{S}^n|}p_n(b))^2\|\boldsymbol{p}_1\|^2 p_n'(t)$$

$$+ \frac{1}{Bt}\sum_{b=1}^t\sum_{n\in\mathcal{B}_b}\frac{\eta^2 t^2 m}{a^2}(\frac{|\mathcal{S}_1^n|b}{|\mathcal{S}^n|t}p_n(t) - (\sigma+\tau))\|\boldsymbol{p}_1\|^2 p_n'(t)\Big)\phi_n(t)(|\mathcal{S}^n| - |\mathcal{S}_1^n|)\|\boldsymbol{q}_1(t)\|^2 \tag{161}$$

where

$$p_n'(t) = \frac{|\mathcal{S}_1^n|e^{\boldsymbol{q}_1(t)^\top\sum_{b=1}^t K(b)\boldsymbol{q}_1(0) - (\delta+\tau)]\|\boldsymbol{q}_1(t)\|}}{|\mathcal{S}_1^n|e^{\boldsymbol{q}_1(t)^\top\sum_{b=1}^t K(b)\boldsymbol{q}_1(b) - (\delta+\tau)]\|\boldsymbol{q}_1(t)\|} + |\mathcal{S}^n| - |\mathcal{S}_1^n|} \tag{162}$$

When $K(b)$ is close to $0^+$, we have

$$\prod_{b=1}^t\sqrt{1 + K(b)}\|\boldsymbol{q}(0)\|^2 \gtrsim e^{\sum_{b=1}^t K(b)\|\boldsymbol{q}_1(0)\|^2} \geq \sum_{b=1}^t K(b)\|\boldsymbol{q}_1(0)\|^2 \tag{163}$$

where the first step is by $\log(1 + x) \approx x$ when $x \to 0^+$. Therefore, one can derive that

$$\left\langle \eta\frac{1}{B}\sum_{n\in\mathcal{B}_b}\frac{\partial\mathbf{Loss}(\boldsymbol{X}^n, y^n)}{\partial\boldsymbol{W}_Q^{(t)}}\boldsymbol{x}_j^n, \boldsymbol{q}_1(t)\right\rangle \gtrsim \Theta(1) \cdot K(t) \tag{164}$$

Meanwhile, the value of $p_n'(t)$ will increase to 1 during training, making the component of $\boldsymbol{q}_1(t)$ the major part in $\eta\frac{1}{B}\sum_{n\in\mathcal{B}_b}\frac{\partial\mathbf{Loss}(\boldsymbol{X}^n, y^n)}{\partial\boldsymbol{W}_Q^{(t)}}\boldsymbol{x}_j^n$.

Hence, if $j \in \mathcal{S}_l^n$ for $l \geq 3$,

$$\boldsymbol{W}_Q^{(t+1)}\boldsymbol{x}_j = \boldsymbol{q}_l(t) + \Theta(1) \cdot \boldsymbol{n}_j(t) + \Theta(1) \cdot \sum_{b=0}^{t-1} K(b)\boldsymbol{q}_1(b) + \sum_{l=2}^M \gamma_l'\boldsymbol{q}_l(t) \tag{165}$$

Similarly, for $j \in \mathcal{S}_2^n$,

$$\boldsymbol{W}_Q^{(t+1)}\boldsymbol{x}_j = (1 + K(t))\boldsymbol{q}_2(t) + \Theta(1) \cdot \boldsymbol{n}_j(t) + \sum_{b=0}^{t-1} |K_e(b)|\boldsymbol{q}_1(b) + \sum_{l=2}^M \gamma_l'\boldsymbol{q}_l(t) \tag{166}$$

(b) For the gradient of $\boldsymbol{W}_K$, we have

$$
\begin{aligned}
\frac{\partial \overline{\textbf{Loss}}_b}{\partial \boldsymbol{W}_K} =& \frac{1}{B} \sum_{n \in \mathcal{B}_b} \frac{\partial \textbf{Loss}(\boldsymbol{X}^n, y^n)}{\partial F(\boldsymbol{X})} \frac{F(\boldsymbol{X})}{\partial \boldsymbol{W}_K} \\
=& \frac{1}{B} \sum_{n \in \mathcal{B}_b} (-y^n) \sum_{l \in \mathcal{S}^n} \sum_{i=1}^{m} a_{(l)_i} \mathbb{1}[\boldsymbol{W}_{O_{(i,\cdot)}} \boldsymbol{W}_V \boldsymbol{X} \operatorname{softmax}(\boldsymbol{X}^{n\top} \boldsymbol{W}_K^\top \boldsymbol{W}_Q \boldsymbol{x}_l^n) \geq 0] \\
& \cdot \Big( \boldsymbol{W}_{O_{(i,\cdot)}} \sum_{s \in \mathcal{S}^n} \boldsymbol{W}_V \boldsymbol{x}_s^n \operatorname{softmax}(\boldsymbol{x}_s^{n\top} \boldsymbol{W}_K^\top \boldsymbol{W}_Q \boldsymbol{x}_l^n) \boldsymbol{W}_Q^\top \boldsymbol{x}_l^n \\
& \cdot (\boldsymbol{x}_s^n - \sum_{r \in \mathcal{S}^n} \operatorname{softmax}(\boldsymbol{x}_r^{n\top} \boldsymbol{W}_K^\top \boldsymbol{W}_Q \boldsymbol{x}_l^n) \boldsymbol{x}_r^n)^\top \Big)
\end{aligned}
\tag{167}
$$

Hence, for $j \in \mathcal{S}_1^n$, we can follow the derivation of (144) to obtain

$$
\boldsymbol{W}_K^{(t+1)} \boldsymbol{x}_j = (1 + Q(t)) \boldsymbol{q}_1(t) + \Theta(1) \cdot \boldsymbol{o}_j^n(t) \pm \sum_{b=0}^{t-1} |Q_e(b)|(1-\lambda) \boldsymbol{r}_2(b) + \sum_{l=3}^{M} \gamma_l' \boldsymbol{r}_l(t), \tag{168}
$$

where

$$
Q(t) \geq K(t)(1-\lambda) > 0 \tag{169}
$$

for $\lambda < 1$ introduced in Assumption 3, and

$$
|\gamma_l| \lesssim \frac{1}{B} \sum_{n \in \mathcal{B}_b} Q(t) \cdot \frac{|\mathcal{S}_l^n|}{|\mathcal{S}^n| - |\mathcal{S}_*^n|} \tag{170}
$$

$$
|Q_e(t)| \lesssim \frac{1}{B} \sum_{n \in \mathcal{B}_b} Q(t) \cdot \frac{|\mathcal{S}_\#^n|}{|\mathcal{S}^n| - |\mathcal{S}_*^n|} \tag{171}
$$

Similarly, for $j \in \mathcal{S}_2^n$, we have

$$
\boldsymbol{W}_K^{(t+1)} \boldsymbol{x}_j \approx (1 + Q(t)) \boldsymbol{q}_2(t) + \Theta(1) \cdot \boldsymbol{o}_j^n(t) \pm \sum_{b=0}^{t-1} |Q_e(b)|(1-\lambda) \boldsymbol{r}_1(b) + \sum_{l=3}^{M} \gamma_l' \boldsymbol{r}_l(t), \tag{172}
$$

For $j \in \mathcal{S}_z^n$, $z = 3, 4, \cdots, M$, we have

$$
\begin{aligned}
& \Big| \Big\langle \frac{1}{B} \sum_{n \in \mathcal{B}_b} \frac{\partial \textbf{Loss}(\boldsymbol{X}^n, y^n)}{\partial F(\boldsymbol{X})} \frac{F(\boldsymbol{X})}{\partial \boldsymbol{W}_K} \boldsymbol{x}_j^n, \boldsymbol{q}_1(t) \Big\rangle \Big| \\
\lesssim& \Big| \frac{1}{B} \sum_{n \in \mathcal{B}_b} (-y^n) \sum_{l \in \mathcal{S}_1^n} \sum_{i=1}^{m} a_{(l)_i} \mathbb{1}[\boldsymbol{W}_{O_{(i,\cdot)}} \boldsymbol{W}_V \boldsymbol{X} \operatorname{softmax}(\boldsymbol{X}^{n\top} \boldsymbol{W}_K^\top \boldsymbol{W}_Q \boldsymbol{x}_l^n) \geq 0] \\
& \cdot \Big( \boldsymbol{W}_{O_{(i,\cdot)}} (\sum_{s \in \mathcal{S}_z^n} + \lambda \sum_{s \in \mathcal{S}_1^n}) \boldsymbol{W}_V \boldsymbol{x}_s^n \operatorname{softmax}(\boldsymbol{x}_s^{n\top} \boldsymbol{W}_K^\top \boldsymbol{W}_Q \boldsymbol{x}_l^n) \Big) \| \boldsymbol{q}_1(t) \|^2 \Big| \\
\leq& \lambda |Q_f(t)| \| \boldsymbol{q}_1(t) \|^2
\end{aligned}
\tag{173}
$$

$$
\begin{aligned}
& \Big| \Big\langle \frac{1}{B} \sum_{n \in \mathcal{B}_b} \frac{\partial \textbf{Loss}(\boldsymbol{X}^n, y^n)}{\partial F(\boldsymbol{X})} \frac{F(\boldsymbol{X})}{\partial \boldsymbol{W}_K} \boldsymbol{x}_j^n, \boldsymbol{q}_z(t) \Big\rangle \Big| \\
\lesssim& \Big| \frac{1}{B} \sum_{n \in \mathcal{B}_b} (-y^n) \sum_{l \in \mathcal{S}_z^n} \sum_{i=1}^{m} a_{(l)_i} \mathbb{1}[\boldsymbol{W}_{O_{(i,\cdot)}} \boldsymbol{W}_V \boldsymbol{X} \operatorname{softmax}(\boldsymbol{X}^{n\top} \boldsymbol{W}_K^\top \boldsymbol{W}_Q \boldsymbol{x}_l^n) \geq 0] \\
& \cdot \Big( \boldsymbol{W}_{O_{(i,\cdot)}} (\sum_{s \in \mathcal{S}_z^n} + \lambda \sum_{s \in \mathcal{S}_1^n}) \boldsymbol{W}_V \boldsymbol{x}_s^n \operatorname{softmax}(\boldsymbol{x}_s^{n\top} \boldsymbol{W}_K^\top \boldsymbol{W}_Q \boldsymbol{x}_l^n) \Big) \| \boldsymbol{q}_z(t) \|^2 \Big| \\
\leq& \lambda |Q_f(t)| \| \boldsymbol{q}_z(t) \|^2
\end{aligned}
\tag{174}
$$

$$
\begin{aligned}
\boldsymbol{W}_K^{(t+1)} \boldsymbol{x}_j \approx& (1 \pm c_{k_1} \lambda |Q_f(t)|) \boldsymbol{q}_l(t) + \Theta(1) \cdot \boldsymbol{o}_j^n(t) \\
& \pm c_{k_2} \lambda \cdot \sum_{b=0}^{t-1} |Q_f(b)| \boldsymbol{r}_1(b) \pm c_{k_3} \lambda \cdot \sum_{b=0}^{t-1} |Q_f(b)| \boldsymbol{r}_2(b) + \sum_{i=3}^{M} \gamma_i' \boldsymbol{r}_i(t),
\end{aligned}
\tag{175}
$$

where $0 < c_{k_1}, c_{k_2}, c_{k_3} < 1$, and

$$|Q_f(t)| \lesssim Q(t) \tag{176}$$

Therefore, for $l \in \mathcal{S}_1^n$, if $j \in \mathcal{S}_1^n$,

$$\boldsymbol{x}_j^{n\top} \boldsymbol{W}_K^{(t+1)\top} \boldsymbol{W}_Q^{(t+1)} \boldsymbol{x}_l^n$$

$$\gtrsim (1+K(t))(1+Q(t))\|\boldsymbol{q}_1(t)\|^2 - (\delta+\tau)\|\boldsymbol{q}_1(t)\| + \sum_{b=0}^{t-1} K_e(b) \sum_{b=0}^{t-1} Q_e(b)\|\boldsymbol{q}_2(b)\|\|\boldsymbol{r}_2(b)\|$$

$$+ \sum_{l=3}^{M} \gamma_l \gamma_l' \|\boldsymbol{q}_l(t)\| \|\boldsymbol{r}_l(t)\|$$

$$\gtrsim (1+K(t))(1+Q(t))\|\boldsymbol{q}_1(t)\|^2 - (\delta+\tau)\|\boldsymbol{q}_1(t)\|$$

$$- \sqrt{\sum_{l=3}^{M}(\frac{1}{B}\sum_{n \in \mathcal{B}_b} Q(t)\frac{|\mathcal{S}_l^n|}{|\mathcal{S}^n| - |\mathcal{S}_*^n|})^2 \|\boldsymbol{r}_l(t)\|^2} \cdot \sqrt{\sum_{l=3}^{M}(\frac{1}{B}\sum_{n \in \mathcal{B}_b} K(t)\frac{|\mathcal{S}_l^n|}{|\mathcal{S}^n| - |\mathcal{S}_*^n|})^2 \|\boldsymbol{q}_l(t)\|^2}$$

$$\gtrsim (1+K(t)+Q(t))\|\boldsymbol{q}_1(t)\|^2 - (\delta+\tau)\|\boldsymbol{q}_1(t)\| \tag{177}$$

where the second step is by Cauchy-Schwarz inequality.
If $j \notin \mathcal{S}_1^n$,

$$\boldsymbol{x}_j^{n\top} \boldsymbol{W}_K^{(t+1)\top} \boldsymbol{W}_Q^{(t+1)} \boldsymbol{x}_l^n$$
$$\lesssim Q_f(t)\|\boldsymbol{q}_1(t)\|^2 + (\delta+\tau)\|\boldsymbol{q}_1(t)\| \tag{178}$$

Hence, for $j, l \in \mathcal{S}_1^n$,

$$\text{softmax}(\boldsymbol{x}_j^{n\top} \boldsymbol{W}_K^{(t+1)} \boldsymbol{W}_Q^{(t+1)} \boldsymbol{x}_l^n) \gtrsim \frac{e^{(1+K(t))\|\boldsymbol{q}_1(t)\|^2 - (\delta+\tau)\|\boldsymbol{q}_1(t)\|}}{|\mathcal{S}_1^n| e^{(1+K(t))\|\boldsymbol{q}_1(t)\|^2 - (\delta+\tau)\|\boldsymbol{q}_1(t)\|} + (|\mathcal{S}^n| - |\mathcal{S}_1^n|)} \tag{179}$$

$$\text{softmax}(\boldsymbol{x}_j^{n\top} \boldsymbol{W}_K^{(t+1)\top} \boldsymbol{W}_Q^{(t+1)} \boldsymbol{x}_l^n) - \text{softmax}(\boldsymbol{x}_j^{n\top} \boldsymbol{W}_K^{(t)\top} \boldsymbol{W}_Q^{(t)} \boldsymbol{x}_l^n)$$

$$\gtrsim \frac{e^{(1+K(t))\|\boldsymbol{q}_1(t)\|^2 - (\delta+\tau)\|\boldsymbol{q}_1(t)\|}}{|\mathcal{S}_1^n| e^{(1+K(t))\|\boldsymbol{q}_1(t)\|^2 - (\delta+\tau)\|\boldsymbol{q}_1(t)\|} + (|\mathcal{S}^n| - |\mathcal{S}_1^n|)} - \frac{e^{\|\boldsymbol{q}_1(t)\|^2 - (\delta+\tau)\|\boldsymbol{q}_1(t)\|}}{|\mathcal{S}_1^n| e^{\|\boldsymbol{q}_1(t)\|^2 - (\delta+\tau)\|\boldsymbol{q}_1(t)\|} + (|\mathcal{S}^n| - |\mathcal{S}_1^n|)}$$

$$= \frac{|\mathcal{S}^n| - |\mathcal{S}_1^n|}{(|\mathcal{S}_1^n| e^x + (|\mathcal{S}^n| - |\mathcal{S}_1^n|))^2} e^{\|\boldsymbol{q}_1(t)\|^2 - (\delta+\tau)\|\boldsymbol{q}_1(t)\|}(e^{K(t)} - 1)$$

$$\geq \frac{|\mathcal{S}^n| - |\mathcal{S}_1^n|}{(|\mathcal{S}_1^n| e^{(1+K(t))\|\boldsymbol{q}_1(t)\|^2 - (\delta+\tau)\|\boldsymbol{q}_1(t)\|} + (|\mathcal{S}^n| - |\mathcal{S}_1|))^2} e^{\|\boldsymbol{q}_1(t)\|^2 - (\delta+\tau)\|\boldsymbol{q}_1(t)\|} \cdot K(t) \tag{180}$$

where the second to last step is by the Mean Value Theorem with

$$x \in [\|\boldsymbol{q}_1(t)\|^2 - (\delta+\tau)\|\boldsymbol{q}_1(t)\|, (1+K(t))\|\boldsymbol{q}_1(t)\|^2 - (\delta+\tau)\|\boldsymbol{q}_1(t)\|] \tag{181}$$

We then need to study if $l \notin (\mathcal{S}_1^n \cup \mathcal{S}_2^n)$ and $j \in \mathcal{S}_1^n$, i.e.,

$$\boldsymbol{x}_j^{n\top} \boldsymbol{W}_K^{(t+1)\top} \boldsymbol{W}_Q^{(t+1)} \boldsymbol{x}_l^n \gtrsim (1+Q(t)) \sum_{b=0}^{t-1} |K(b)| \|\boldsymbol{q}_1(t)\| \|\boldsymbol{q}_1(b)\| - (\delta+\tau)\|\boldsymbol{q}_1(t)\| \tag{182}$$

For $j, l \notin (\mathcal{S}_1^n \cup \mathcal{S}_2^n)$,

$$\boldsymbol{x}_j^{n\top} \boldsymbol{W}_K^{(t+1)\top} \boldsymbol{W}_Q^{(t+1)} \boldsymbol{x}_l^n \lesssim \pm c_{k_2}\lambda \cdot \sum_{b=0}^{t-1} |Q_f(b)| \boldsymbol{r}_1(b) \pm c_{k_3}\lambda \cdot \sum_{b=0}^{t-1} |Q_f(b)| \boldsymbol{r}_2(b)$$

$$+ (1 \pm c_{k_1}\lambda|Q_f(t)|)\|\boldsymbol{q}_l(t)\|^2 \tag{183}$$

We know that the magnitude of $\|\boldsymbol{q}_1(t)\|$ increases along the training and finally reaches no larger than $\Theta(\sqrt{\log T})$. At the final step, we have

$$\sum_{b=0}^{t-1} K(b)\|\boldsymbol{q}_1(b)\| \geq \frac{T}{e^{\|\boldsymbol{q}_1(T)\|^2 - (\delta+\tau)\|\boldsymbol{q}_1(T)\|}} \geq \Theta(\sqrt{\log T}) \tag{184}$$

Therefore, when $t$ is large enough during the training but before the final step of convergence, we have if $j', l \notin (\mathcal{S}_1^n \cup \mathcal{S}_2^n)$ and $j \in \mathcal{S}_1^n$, we can obtain

$$(\boldsymbol{x}_j^n - \boldsymbol{x}_{j'}^n)^\top \boldsymbol{W}_K^{(t+1)\top} \boldsymbol{W}_Q^{(t+1)} \boldsymbol{x}_l^n \gtrsim \Theta(1) \cdot ((1 + K(t))\|\boldsymbol{q}_1(t)\|^2 - (\delta + \tau)\|\boldsymbol{q}_1(t)\|) \quad (185)$$

We can derive the same conclusion for $j \in \mathcal{S}_2^n$ in (185). Therefore, by $|\mathcal{S}_2| \le \epsilon_{\mathcal{S}} e^{-(\delta+\tau)}(1 - (\sigma - \tau))|\mathcal{S}_1^n|$ in (157), we can obtain

$$\text{softmax}(\boldsymbol{x}_j^{n\top} \boldsymbol{W}_K^{(t+1)} \boldsymbol{W}_Q^{(t+1)} \boldsymbol{x}_l^n)$$
$$\gtrsim \frac{e^{(1+K(t))\|\boldsymbol{q}_1(t)\|^2 - (\delta+\tau)\|\boldsymbol{q}_1(t)\|}}{(|\mathcal{S}_1^n| + |\mathcal{S}_2^n|)e^{(1+K(t))\|\boldsymbol{q}_1(t)\|^2 - (\delta+\tau)\|\boldsymbol{q}_1(t)\|} + (|\mathcal{S}^n| - |\mathcal{S}_1^n| - |\mathcal{S}_2^n|)} \quad (186)$$
$$\gtrsim \frac{e^{(1+K(t))\|\boldsymbol{q}_1(t)\|^2 - (\delta+\tau)\|\boldsymbol{q}_1(t)\|}}{|\mathcal{S}_1^n|e^{(1+K(t))\|\boldsymbol{q}_1(t)\|^2 - (\delta+\tau)\|\boldsymbol{q}_1(t)\|} + (|\mathcal{S}^n| - |\mathcal{S}_1^n|)}$$

$$\text{softmax}(\boldsymbol{x}_j^{n\top} \boldsymbol{W}_K^{(t+1)\top} \boldsymbol{W}_Q^{(t+1)} \boldsymbol{x}_l^n) - \text{softmax}(\boldsymbol{x}_j^{n\top} \boldsymbol{W}_K^{(t)\top} \boldsymbol{W}_Q^{(t)} \boldsymbol{x}_l^n)$$
$$\gtrsim \frac{|\mathcal{S}^n| - |\mathcal{S}_1^n|}{(|\mathcal{S}_1^n|e^{\Theta(1) \cdot ((1+K(t))\|\boldsymbol{q}_1(t)\|^2 - (\delta+\tau)\|\boldsymbol{q}_1(t)\|)} + (|\mathcal{S}^n| - |\mathcal{S}_1|))^2} e^{\Theta(1)(\|\boldsymbol{q}_1(t)\|^2 - (\delta+\tau)\|\boldsymbol{q}_1(t)\|)} \cdot K(t)$$
$$\gtrsim \frac{|\mathcal{S}^n| - |\mathcal{S}_1^n|}{(|\mathcal{S}_1^n|e^{(1+K(t))\|\boldsymbol{q}_1(t)\|^2 - (\delta+\tau)\|\boldsymbol{q}_1(t)\|} + (|\mathcal{S}^n| - |\mathcal{S}_1|))^2} e^{\|\boldsymbol{q}_1(t)\|^2 - (\delta+\tau)\|\boldsymbol{q}_1(t)\|} \cdot K(t)$$
$$(187)$$

Meanwhile, for $l \in \mathcal{S}_1^n$ and $j \notin \mathcal{S}_1^n$,

$$\text{softmax}(\boldsymbol{x}_j^{n\top} \boldsymbol{W}_K^{(t+1)\top} \boldsymbol{W}_Q^{(t+1)} \boldsymbol{x}_l^n) \lesssim \frac{1}{|\mathcal{S}_1^n|e^{(1+K(t))\|\boldsymbol{q}_1(t)\|^2 - (\delta+\tau)\|\boldsymbol{q}_1(t)\|} + (|\mathcal{S}^n| - |\mathcal{S}_1^n|)} \quad (188)$$

$$\text{softmax}(\boldsymbol{x}_j^{n\top} \boldsymbol{W}_K^{(t+1)} \boldsymbol{W}_Q^{(t+1)} \boldsymbol{x}_l^n) - \text{softmax}(\boldsymbol{x}_j^{n\top} \boldsymbol{W}_K^{(t)\top} \boldsymbol{W}_Q^{(t)} \boldsymbol{x}_l^n)$$
$$\lesssim \frac{1}{|\mathcal{S}_1^n|e^{(1+K(t))\|\boldsymbol{q}_1(t)\|^2 - (\delta+\tau)\|\boldsymbol{q}_1(t)\|} + (|\mathcal{S}^n| - |\mathcal{S}_1^n|)} - \frac{1}{|\mathcal{S}_1^n|e^{\|\boldsymbol{q}_1(t)\|^2 - (\delta+\tau)\|\boldsymbol{q}_1(t)\|} + (|\mathcal{S}^n| - |\mathcal{S}_1^n|)}$$
$$= -\frac{|\mathcal{S}_1^n|}{(|\mathcal{S}_1^n|e^x + (|\mathcal{S}^n| - |\mathcal{S}_1^n|))^2} e^{\|\boldsymbol{q}_1(t)\|^2 - (\delta+\tau)\|\boldsymbol{q}_1(t)\|}(e^{K(t)} - 1)$$
$$\le -\frac{|\mathcal{S}_1^n|}{(|\mathcal{S}_1|e^{(1+K(t))\|\boldsymbol{q}_1(t)\|^2 - (\delta+\tau)\|\boldsymbol{q}_1(t)\|} + (|\mathcal{S}^n| - |\mathcal{S}_1^n|))^2} e^{\|\boldsymbol{q}_1(t)\|^2 - (\delta+\tau)\|\boldsymbol{q}_1(t)\|} \cdot K(t)$$
$$(189)$$

where the second to last step is by the Mean Value Theorem with

$$x \in [\|\boldsymbol{q}_1(t)\|^2 - (\delta + \tau)\|\boldsymbol{q}_1(t)\|, (1 + K(t))\|\boldsymbol{q}_1(t)\|^2 - (\delta + \tau)\|\boldsymbol{q}_1(t)\|] \quad (190)$$

The same conclusion holds if $l \notin (\mathcal{S}_1^n \cup \mathcal{S}_2^n)$ and $j \notin \mathcal{S}_1^n$.
Note that

$$\boldsymbol{q}_1(t+1) = \sqrt{(1 + K(t))}\boldsymbol{q}_1(t) \quad (191)$$

$$\boldsymbol{q}_2(t+1) = \sqrt{(1 + K(t))}\boldsymbol{q}_2(t) \quad (192)$$

$$\boldsymbol{r}_1(t+1) = \sqrt{(1 + Q(t))}\boldsymbol{r}_1(t) \quad (193)$$

$$\boldsymbol{r}_2(t+1) = \sqrt{(1 + Q(t))}\boldsymbol{r}_2(t) \quad (194)$$

It can also be verified that this claim holds when $t = 1$.

### E.3 PROOF OF CLAIM 3 OF LEMMA 2

The computation of the gradient of $\boldsymbol{W}_V$ is straightforward. The gradient would be related to $\boldsymbol{W}_O$ by their connections. One still need to study the influence of the gradient on different patterns, where we introduce the discussion for the term $V_i(t)$'s.

For the gradient of $\boldsymbol{W}_V$, by (18) we have

$$
\begin{aligned}
\frac{\partial \overline{\mathbf{Loss}}_b}{\partial \boldsymbol{W}_V} =& \frac{1}{B} \sum_{n \in \mathcal{B}_b} \frac{\partial \mathbf{Loss}(\boldsymbol{X}^n, y^n)}{\partial F(\boldsymbol{X}^n)} \frac{\partial F(\boldsymbol{X}^n)}{\partial \boldsymbol{W}_V} \\
=& -y \frac{1}{B} \sum_{n \in \mathcal{B}_b} \frac{1}{|\mathcal{S}^n|} \sum_{l \in \mathcal{S}^n} \sum_{i=1}^{m} a^*_{(l)_i} \mathbb{1}[\boldsymbol{W}_{O_{(i,\cdot)}} \boldsymbol{W}_V \boldsymbol{X}^n \text{softmax}(\boldsymbol{X}^{n\top} \boldsymbol{W}_K^\top \boldsymbol{W}_Q \boldsymbol{x}_l^n) \geq 0] \\
& \cdot \boldsymbol{W}_{O_{(i,\cdot)}}{}^\top \text{softmax}(\boldsymbol{X}^{n\top} \boldsymbol{W}_K^\top \boldsymbol{W}_Q \boldsymbol{x}_l^n)^\top \boldsymbol{X}^{n\top}
\end{aligned}
\tag{195}
$$

Consider a data $\{\boldsymbol{X}^n, y^n\}$ where $y^n = 1$. Let $l \in \mathcal{S}_1^n$

$$
\sum_{s \in \mathcal{S}_1^n} \text{softmax}(\boldsymbol{x}_s^{n\top} \boldsymbol{W}_K^{(t)\top} \boldsymbol{W}_Q^{(t)} \boldsymbol{x}_l^n) \geq p_n(t)
\tag{196}
$$

Then for $j \in \mathcal{S}_1^g$, $g \in [N]$,

$$
\begin{aligned}
& \frac{1}{B} \sum_{n \in \mathcal{B}_b} \frac{\partial \mathbf{Loss}(\boldsymbol{X}^n, y^n)}{\partial \boldsymbol{W}_V^{(t)}} \bigg| \boldsymbol{W}_V^{(t)} \boldsymbol{x}_j \\
=& \frac{1}{B} \sum_{n \in \mathcal{B}_b} (-y^n) \frac{1}{|\mathcal{S}^n|} \sum_{l \in \mathcal{S}^n} \sum_{i=1}^{m} a_{(l)_i} \mathbb{1}[\boldsymbol{W}_{O_{(i,\cdot)}}^{(t)} \sum_{s \in \mathcal{S}^n} \text{softmax}(\boldsymbol{x}_s^{n\top} \boldsymbol{W}_K^{(t)\top} \boldsymbol{W}_Q^{(t)} \boldsymbol{x}_l^n) \boldsymbol{W}_V^{(t)} \boldsymbol{x}_s^n \geq 0] \\
& \cdot \boldsymbol{W}_{O_{(i,\cdot)}}^{(t)}{}^\top \sum_{s \in \mathcal{S}^n} \text{softmax}(\boldsymbol{x}_s^{n\top} \boldsymbol{W}_K^{(t)\top} \boldsymbol{W}_Q^{(t)} \boldsymbol{x}_l) \boldsymbol{x}_s^{n\top} \boldsymbol{x}_j^g \\
=& \sum_{i \in \mathcal{W}(t)} V_i(t) \boldsymbol{W}_{O_{(i,\cdot)}}{}^\top + \sum_{i \notin \mathcal{W}(t)} \lambda V_i(t) \boldsymbol{W}_{O_{(i,\cdot)}}{}^\top,
\end{aligned}
\tag{197}
$$

If $i \in \mathcal{W}(t)$, by the fact that $\mathcal{S}_\#^n$ contributes more to $V_i(t)$ compared to $\mathcal{S}_l^n$ for $l \geq 3$ and Assumption 3, we have

$$
\begin{aligned}
V_i(t) \lesssim& \frac{1}{2B} \sum_{n \in \mathcal{B}_{b_+}} -\frac{|\mathcal{S}_1^n|}{a|\mathcal{S}^n|} p_n(t) + \frac{|\mathcal{S}_2^n|}{a|\mathcal{S}^n|} |\lambda| \nu_n(t)(|\mathcal{S}^n| - |\mathcal{S}_1^n|) \\
\lesssim& \frac{1}{2B} \sum_{n \in \mathcal{B}_{b_+}} -\frac{|\mathcal{S}_1^n|}{a|\mathcal{S}^n|} p_n(t)
\end{aligned}
\tag{198}
$$

Similarly, if $i \in \mathcal{U}(t)$,

$$
V_i(t) \gtrsim \frac{1}{2B} \sum_{n \in \mathcal{B}_{b_-}} \frac{|\mathcal{S}_2^n|}{a|\mathcal{S}^n|} p_n(t)
\tag{199}
$$

if $i$ is an unlucky neuron, by Hoeffding's inequality in (26), we have

$$
\begin{aligned}
V_i(t) \geq& \frac{1}{\sqrt{B}} \cdot \frac{1}{a} \cdot \sqrt{M} \xi \|\boldsymbol{p}\| \\
\gtrsim& -\frac{1}{\sqrt{B}a}
\end{aligned}
\tag{200}
$$

For $i \in \mathcal{W}(0)$, we have

$$
-\eta \sum_{b=1}^{t} \boldsymbol{W}_{O_{(i,\cdot)}}^{(b)} \sum_{j \in \mathcal{W}(b)} V_j(b) \boldsymbol{W}_{O_{(j,\cdot)}}^{(b)}{}^{\top}
$$

$$
\gtrsim \frac{\eta m}{M}(1 - \epsilon_m - \frac{(\sigma+\tau)M}{\pi}) \frac{1}{2Bt} \sum_{b=1}^{t} \sum_{n \in \mathcal{B}_{b+}} \frac{|\mathcal{S}_1^n|}{a|\mathcal{S}^n|} p_n(b)(1-(\sigma+\tau))
$$

$$
\cdot \left( \frac{1}{Bt} \sum_{n \in \mathcal{B}_b} \frac{\eta^2 b^2 m |\mathcal{S}_1^n|}{a^2 |\mathcal{S}^n|} \|\boldsymbol{p}\|^2 p_n(b) \right)^2
$$

$$
\gtrsim (1 - \epsilon_m - \frac{(\sigma+\tau)M}{\pi}) \frac{1}{2Bt} \sum_{b=1}^{t} \sum_{n \in \mathcal{B}_{b+}} \frac{|\mathcal{S}_1^n| p_n(b) m}{aM|\mathcal{S}^n|} p_n(b) \cdot \left( \frac{1}{Bt} \sum_{b=1}^{t} \sum_{n \in \mathcal{B}_b} \frac{\eta^2 t b m |\mathcal{S}_1^n|}{a^2 |\mathcal{S}^n|} \|\boldsymbol{p}\|^2 p_n(b) \right)^2
$$

$$
\tag{201}
$$

$$
-\eta \sum_{b=1}^{t} \boldsymbol{W}_{O_{(i,\cdot)}}^{(b)} \sum_{j \in \mathcal{U}(b)} V_j(b) \boldsymbol{W}_{O_{(j,\cdot)}}^{(b)}{}^{\top}
$$

$$
\lesssim -\frac{1}{Bt} \sum_{b=1}^{t} \sum_{n \in \mathcal{B}_b} \frac{|\mathcal{S}_2^n| p_n(b) m}{|\mathcal{S}^n| aM} (1 - \epsilon_m - \frac{(\sigma+\tau)M}{\pi})(\frac{\eta^2 t^2 m}{a^2})^2 (\sigma+\tau) \|\boldsymbol{p}_1\|^2
$$

$$
\tag{202}
$$

$$
-\eta t \boldsymbol{W}_{O_{(i,\cdot)}} \sum_{j \notin (\mathcal{W}(t) \cup \mathcal{U}(t))} V_j(t) \boldsymbol{W}_{O_{(j,\cdot)}}{}^{\top} \lesssim \frac{\eta^2 t^2 m \lambda \xi \sqrt{M} \|\boldsymbol{p}\|^2}{\sqrt{B} a^2}
$$

$$
\tag{203}
$$

Hence,

(1) If $j \in \mathcal{S}_1^n$ for one $n \in [N]$,

$$
\boldsymbol{W}_V^{(t+1)} \boldsymbol{x}_j^n = \boldsymbol{W}_V^{(t)} \boldsymbol{x}_j^n - \eta \left( \frac{\partial L}{\partial \boldsymbol{W}_V} \Big| \boldsymbol{W}_V^{(t)} \right) \boldsymbol{x}_j^n
$$

$$
= \boldsymbol{p}_1 - \eta \sum_{b=1}^{t+1} \sum_{i \in \mathcal{W}(b)} V_i(b) \boldsymbol{W}_{O_{(i,\cdot)}}^{(b)}{}^{\top} - \eta \sum_{b=1}^{t+1} \sum_{i \notin \mathcal{W}(b)} \lambda V_i(b) \boldsymbol{W}_{O_{(i,\cdot)}}^{(b)}{}^{\top} + \boldsymbol{z}_j(t)
$$

$$
\tag{204}
$$

(2) If $j \in \mathcal{S}_2^n$, we have

$$
\boldsymbol{W}_V^{(t+1)} \boldsymbol{x}_j = \boldsymbol{W}_V^{(0)} \boldsymbol{x}_j^n - \eta \left( \frac{\partial L}{\partial \boldsymbol{W}_V} \Big| \boldsymbol{W}_V^{(0)} \right) \boldsymbol{x}_j^n
$$

$$
= \boldsymbol{p}_2 - \eta \sum_{b=1}^{t+1} \sum_{i \in \mathcal{U}(b)} V_i(b) \boldsymbol{W}_{O_{(i,\cdot)}}^{(b)}{}^{\top} - \eta \sum_{b=1}^{t+1} \sum_{i \notin \mathcal{U}(b)} \lambda V_i(b) \boldsymbol{W}_{O_{(i,\cdot)}}^{(b)}{}^{\top} + \boldsymbol{z}_j(t)
$$

$$
\tag{205}
$$

(3) If $j \in \mathcal{S}^n/(\mathcal{S}_1^n \cup \mathcal{S}_2^n)$, we have

$$
\boldsymbol{W}_V^{(t+1)} \boldsymbol{x}_j^n = \boldsymbol{W}_V^{(0)} \boldsymbol{x}_j^n - \eta \left( \frac{\partial L}{\partial \boldsymbol{W}_V} \Big| \boldsymbol{W}_V^{(0)} \right) \boldsymbol{x}_j^n
$$

$$
= \boldsymbol{p}_k - \eta \sum_{b=1}^{t+1} \sum_{i=1}^{m} \lambda V_i(b) \boldsymbol{W}_{O_{(i,\cdot)}}^{(b)}{}^{\top} + \boldsymbol{z}_j(t)
$$

$$
\tag{206}
$$

Here

$$
\|\boldsymbol{z}_j(t)\| \le (\sigma+\tau) \tag{207}
$$

for $t \ge 1$. Note that this claim also holds when $t = 1$.

## F  OTHER USEFUL LEMMAS

**Lemma 3.** *If the number of neurons $m$ is larger enough such that*

$$
m \ge \epsilon_m^{-2} M^2 \log N, \tag{208}
$$

*the number of lucky neurons at the initialization $|\mathcal{W}(0)|$, $|\mathcal{U}(0)|$ satisfies*

$$
|\mathcal{W}(0)|, |\mathcal{U}(0)| \ge \frac{m}{M}(1 - \epsilon_m - \frac{(\sigma+\tau)M}{\pi}) \tag{209}
$$

**Proof:**
Let $\theta_l$ be the angle between $\boldsymbol{p}_l$ and the initial weight for one $i \in [m]$ and all $l \in [M]$. For the lucky neuron $i \in \mathcal{W}(0)$, $\theta_1$ should be the smallest among $\{\theta_l\}_{l=1}^M$ with noise $\Delta\theta$. Hence, the probability of the lucky neuron can be bounded as

$$\Pr\left(\theta_1 + \Delta\theta \le \theta_l - \Delta\theta \le 2\pi, \ 2 \le l \le M\right)$$
$$= \prod_{l=2}^L \Pr\left(\theta_1 + \Delta\theta \le \theta_l - \Delta\theta \le 2\pi\right) \tag{210}$$
$$= \left(\frac{2\pi - \theta_1 - 2\Delta\theta}{2\pi}\right)^{M-1},$$

where the first step is because the Gaussian $\boldsymbol{W}_{O_{(i,\cdot)}}^{(0)}$ and orthogonal $\boldsymbol{p}_l$, $l \in [M]$ generate independent $\boldsymbol{W}_{O_{(i,\cdot)}}^{(0)} \boldsymbol{p}_l$. From the definition of $\mathcal{W}(0)$, we have

$$2 \sin \frac{1}{2}\Delta\theta \le (\sigma + \tau), \tag{211}$$

which implies

$$\Delta\theta \lesssim (\sigma + \tau) \tag{212}$$

for small $\sigma > 0$. Therefore,

$$\Pr\left(i \in \mathcal{W}(0)\right) = \int_0^{2\pi} \frac{1}{2\pi} \cdot \left(\frac{2\pi - \theta_1 - 2\Delta\theta}{2\pi}\right)^{M-1} d\theta_1$$
$$= -\frac{1}{M}\left(\frac{2\pi - 2\Delta\theta - x}{2\pi}\right)^M \Big|_0^{2\pi}$$
$$\gtrsim \frac{1}{M}\left(1 - \frac{\Delta\theta}{\pi}\right)^M \tag{213}$$
$$\gtrsim \frac{1}{M}\left(1 - \frac{(\sigma + \tau)M}{\pi}\right),$$

where the first step comes from that $\theta_1$ follows the uniform distribution on $[0, 2\pi]$ due to the Gaussian initialization of $\boldsymbol{W}_O$. We can define the random variable $v_i$ such that

$$v_i = \begin{cases} 1, & \text{if } i \in \mathcal{W}(0), \\ 0, & \text{else} \end{cases} \tag{214}$$

We know that $v_i$ belongs to Bernoulli distribution with probability $\frac{1}{M}(1 - \frac{(\sigma+\tau)M}{\pi})$. By Hoeffding's inequality in (26), we know that with probability at least $1 - N^{-10}$,

$$\frac{1}{M}\left(1 - \frac{(\sigma+\tau)M}{\pi}\right) - \sqrt{\frac{\log N}{m}} \le \frac{1}{m}\sum_{i=1}^m v_i \le \frac{1}{M}\left(1 - \frac{(\sigma+\tau)M}{\pi}\right) + \sqrt{\frac{\log N}{m}} \tag{215}$$

Let $m \ge \Theta(\epsilon_m^{-2} M^2 \log B)$, we have

$$|\mathcal{W}(0)| = \sum_{i=1}^m v_i \ge \frac{m}{M}\left(1 - \epsilon_m - \frac{(\sigma+\tau)M}{\pi}\right) \tag{216}$$

where we require

$$(\sigma + \tau) \le \frac{\pi}{M} \tag{217}$$

to ensure a positive probability in (216). Likewise, the conclusion holds for $\mathcal{U}(0)$.

**Lemma 4.** *Let $\mathcal{W}(t)$ and $\mathcal{U}(t)$ be defined in Definition 2. We then have*

$$\mathcal{W}(0) \subseteq \mathcal{W}(t) \tag{218}$$

$$\mathcal{U}(0) \subseteq \mathcal{U}(t) \tag{219}$$

as long as

$$B \gtrsim \Theta(1) \tag{220}$$

**Proof:**

We show this lemma by induction.

**(1)** $t = 0$. For $i \in \mathcal{W}(0)$, by Definition 2, we know that the angle between $\boldsymbol{W}_{O_{(i,\cdot)}}^{(0)}$ and $\boldsymbol{p}_1$ is smaller than $(\sigma + \tau)$. Hence, we have

$$\boldsymbol{W}_{O_{(i,\cdot)}}^{(0)} \boldsymbol{p}_1 (1 - (\sigma + \tau)) \geq \boldsymbol{W}_{O_{(i,\cdot)}}^{(0)} \boldsymbol{p} (1 + (\sigma + \tau)) \tag{221}$$

for all $\boldsymbol{p} \in \mathcal{P}/\boldsymbol{p}_1$.

**(2) Suppose that the conclusion holds when $t = s$.** When $t = s + 1$, from Lemma 2 Claim 1, we can obtain

$$\left\langle \boldsymbol{W}_{O_{(i,\cdot)}}^{(s+1)\top}, \boldsymbol{p}_1 \right\rangle - \left\langle \boldsymbol{W}_{O_{(i,\cdot)}}^{(s)\top}, \boldsymbol{p}_1 \right\rangle$$
$$\gtrsim \frac{\eta}{m} \cdot \frac{1}{B} \sum_{n \in \mathcal{B}_b} \left( \frac{|\mathcal{S}_1^n|}{|\mathcal{S}^n|} p_n(s) - \frac{((\sigma + \tau))\eta s |\mathcal{S}_1^n|}{\sqrt{N} m T |\mathcal{S}^n|} p_n(s) - \frac{\xi}{N} \right) \|\boldsymbol{p}_1\|^2 \tag{222}$$

and

$$\left| \left\langle \boldsymbol{W}_{O_{(i,\cdot)}}^{(s+1)\top}, \boldsymbol{p} \right\rangle - \left\langle \boldsymbol{W}_{O_{(i,\cdot)}}^{(s)\top}, \boldsymbol{p} \right\rangle \right|$$
$$\lesssim \frac{\eta}{m} \cdot \frac{1}{B} \sum_{n \in \mathcal{B}_b} \left( \frac{|\mathcal{S}_l^n|}{|\mathcal{S}^n|} |\mathcal{S}_l^n| \nu_n(s) \|\boldsymbol{p}\| + \frac{((\sigma + \tau))\eta s |\mathcal{S}_1^n|}{T m |\mathcal{S}^n|} p_n(s) \right. \tag{223}$$
$$\left. + \frac{|\mathcal{S}_1^n| p_n(s)(\sigma + \tau)\|\boldsymbol{p}_1\|}{|\mathcal{S}^n| M} \right) \sqrt{\frac{\log m \log B}{B}} \|\boldsymbol{p}\| + \frac{\eta}{Bm} \xi \|\boldsymbol{p}\|$$

Combining (222) and (223), we can approximately compute that if

$$B \gtrsim \left( \frac{1 + (\sigma + \tau)}{1 - (\sigma + \tau)} \right)^2 \gtrsim \Theta(1), \tag{224}$$

we can derive

$$\boldsymbol{W}_{O_{(i,\cdot)}}^{(s+1)} \boldsymbol{p}_1 (1 - (\sigma + \tau)) \geq \boldsymbol{W}_{O_{(i,\cdot)}}^{(s+1)} \boldsymbol{p} (1 + (\sigma + \tau)) \tag{225}$$

Therefore, we have

$$\mathcal{W}(0) \subseteq \mathcal{W}(s + 1) \tag{226}$$

In conclusion, we can obtain

$$\mathcal{W}(0) \subseteq \mathcal{W}(t) \tag{227}$$

for all $t \geq 0$.

One can develop the proof for $\mathcal{U}(t)$ following the above steps.

## G  EXTENSION TO MORE GENERAL CASES

### G.1  EXTENSION TO MULTI-CLASSIFICATION

Consider the classification problem with four classes, we use the label $y \in \{+1, -1\}^2$ to denote the corresponding class. Similarly to the previous setup, there are four orthogonal discriminative patterns. In the output layer, $a_{l_{(i)}}$ for the data $(\boldsymbol{X}^n, y^n)$ is changed into an $\mathbb{R}^2$ vector $\boldsymbol{a}_{l_{(i)}}$ for $l \in [|\mathcal{S}^n|]$ and $i \in [m]$. Hence, we define

$$\boldsymbol{F}(\boldsymbol{X}^n) = \frac{1}{|\mathcal{S}^n|} \sum_{l \in \mathcal{S}^n} \boldsymbol{a}_{l_{(i)}} \mathrm{Relu}(\boldsymbol{W}_O \boldsymbol{W}_V \boldsymbol{X}^n \mathrm{softmax}(\boldsymbol{X}^{n\top} \boldsymbol{W}_Q^\top \boldsymbol{W}_K \boldsymbol{x}_l^n)) \tag{228}$$

$$F_1(\boldsymbol{X}^n) = \frac{1}{|\mathcal{S}^n|} \sum_{l \in \mathcal{S}^n} a_{l_{1(i)}} \mathrm{Relu}(\boldsymbol{W}_O \boldsymbol{W}_V \boldsymbol{X}^n \mathrm{softmax}(\boldsymbol{X}^{n\top} \boldsymbol{W}_Q^\top \boldsymbol{W}_K \boldsymbol{x}_l^n)) \tag{229}$$

$$F_2(\boldsymbol{X}^n) = \frac{1}{|\mathcal{S}^n|} \sum_{l \in \mathcal{S}^n} a_{l_{2(i)}} \mathrm{Relu}(\boldsymbol{W}_O \boldsymbol{W}_V \boldsymbol{X}^n \mathrm{softmax}(\boldsymbol{X}^{n\top} \boldsymbol{W}_Q^\top \boldsymbol{W}_K \boldsymbol{x}_l^n)) \tag{230}$$

The dataset $\mathcal{D}$ can be divided into four groups as

$$
\begin{aligned}
\mathcal{D}_1 &= \{(\boldsymbol{X}^n, \boldsymbol{y}^n) | \boldsymbol{y}^n = (1, 1)\} \\
\mathcal{D}_2 &= \{(\boldsymbol{X}^n, \boldsymbol{y}^n) | \boldsymbol{y}^n = (1, -1)\} \\
\mathcal{D}_3 &= \{(\boldsymbol{X}^n, \boldsymbol{y}^n) | \boldsymbol{y}^n = (-1, 1)\} \\
\mathcal{D}_4 &= \{(\boldsymbol{X}^n, \boldsymbol{y}^n) | \boldsymbol{y}^n = (-1, -1)\}
\end{aligned}
\tag{231}
$$

The hinge loss function for data $(\boldsymbol{X}^n, \boldsymbol{y}^n)$ will be

$$
\text{Loss}(\boldsymbol{X}^n, \boldsymbol{y}^n) = \max\{1 - \boldsymbol{y}^{n\top} \boldsymbol{F}(\boldsymbol{X}^n), 0\}
\tag{232}
$$

We can divide the weights $\boldsymbol{W}_{O_{(i,\cdot)}}$ $(i \in [m])$ into two groups, respectively.

$$
\begin{aligned}
\mathcal{W}_1 &= \{i | \boldsymbol{a}_{l_{(i)}} = \frac{1}{\sqrt{m}} \cdot (1, 1)\} \\
\mathcal{W}_2 &= \{i | \boldsymbol{a}_{l_{(i)}} = \frac{1}{\sqrt{m}} \cdot (1, -1)\} \\
\mathcal{W}_3 &= \{i | \boldsymbol{a}_{l_{(i)}} = \frac{1}{\sqrt{m}} \cdot (-1, 1)\} \\
\mathcal{W}_4 &= \{i | \boldsymbol{a}_{l_{(i)}} = \frac{1}{\sqrt{m}} \cdot (-1, -1)\}
\end{aligned}
\tag{233}
$$

Therefore, for $\boldsymbol{W}_{O_u}$ in the network (228), we have

$$
\frac{\partial \text{Loss}(\boldsymbol{X}^n, \boldsymbol{y}^n)}{\partial \boldsymbol{W}_{O_{(i,\cdot)}}^\top} = -y_1^n \frac{\partial F_1(\boldsymbol{X}^n)}{\partial \boldsymbol{W}_{O_{1(i,\cdot)}}} - y_2^n \frac{\partial F_2(\boldsymbol{X}^n)}{\boldsymbol{W}_{O_{2(i,\cdot)}}}
\tag{234}
$$

where the derivation of $\frac{\partial F_1(\boldsymbol{X}^n)}{\partial \boldsymbol{W}_{O_{1(i,\cdot)}}}$ and $\frac{\partial F_2(\boldsymbol{X}^n)}{\partial \boldsymbol{W}_{O_{2(i,\cdot)}}}$ can be found in the analysis of binary classification above. For any $i \in \mathcal{W}_2$, following the proof of Claim 1 of Lemma 2, if the data $(\boldsymbol{X}^n, \boldsymbol{y}^n) \in \mathcal{D}_2$, we have

$$
-\frac{\partial \text{Loss}(\boldsymbol{X}^n, \boldsymbol{y}^n)}{\partial \boldsymbol{W}_{O_{(i,\cdot)}}^\top} = y_1^n \frac{\partial F_1(\boldsymbol{X}^n)}{\partial \boldsymbol{W}_{O_{1(i,\cdot)}}} + y_2^n \frac{\partial F_2(\boldsymbol{X}^n)}{\boldsymbol{W}_{O_{2(i,\cdot)}}} \approx\propto 1 \cdot \frac{1}{\sqrt{m}} \boldsymbol{p}_2 - 1 \cdot (-\frac{1}{\sqrt{m}}) \boldsymbol{p}_2 = \frac{2}{\sqrt{m}} \boldsymbol{p}_2 \tag{235}
$$

$$
(\boldsymbol{W}_{O_{(i,\cdot)}}^{(t+1)} - \boldsymbol{W}_{O_{(i,\cdot)}}^{(t)}) \boldsymbol{p}_2 \propto \|\boldsymbol{p}_2\|^2 > 0
\tag{236}
$$

if $(\boldsymbol{X}^n, y^n) \in \mathcal{D}_1$, we have

$$
-\frac{\partial \text{Loss}(\boldsymbol{X}^n, \boldsymbol{y}^n)}{\partial \boldsymbol{W}_{O_{(i,\cdot)}}^\top} \approx\propto 1 \cdot \frac{1}{\sqrt{m}} \boldsymbol{p}_1 + 1 \cdot (-\frac{1}{\sqrt{m}}) \boldsymbol{p}_1 = 0
\tag{237}
$$

$$
(\boldsymbol{W}_{O_{(i,\cdot)}}^{(t+1)} - \boldsymbol{W}_{O_{(i,\cdot)}}^{(t)}) \boldsymbol{p}_1 \approx= 0
\tag{238}
$$

if $(\boldsymbol{X}^n, y^n) \in \mathcal{D}_3$, we have

$$
-\frac{\partial \text{Loss}(\boldsymbol{X}^n, \boldsymbol{y}^n)}{\partial \boldsymbol{W}_{O_{(i,\cdot)}}^\top} \approx\propto -1 \cdot \frac{1}{\sqrt{m}} \boldsymbol{p}_3 + 1 \cdot (-\frac{1}{\sqrt{m}}) \boldsymbol{p}_3 = -\frac{2}{\sqrt{m}} \boldsymbol{p}_3
\tag{239}
$$

$$
(\boldsymbol{W}_{O_{(i,\cdot)}}^{(t+1)} - \boldsymbol{W}_{O_{(i,\cdot)}}^{(t)}) \boldsymbol{p}_3 \leq 0
\tag{240}
$$

if $(\boldsymbol{X}^n, y^n) \in \mathcal{D}_4$, we have

$$
-\frac{\partial \text{Loss}(\boldsymbol{X}^n, \boldsymbol{y}^n)}{\partial \boldsymbol{W}_{O_{(i,\cdot)}}^\top} \approx\propto -1 \cdot \frac{1}{\sqrt{m}} \boldsymbol{p}_4 - 1 \cdot (-\frac{1}{\sqrt{m}}) \boldsymbol{p}_4 = 0
\tag{241}
$$

$$
(\boldsymbol{W}_{O_{(i,\cdot)}}^{(t+1)} - \boldsymbol{W}_{O_{(i,\cdot)}}^{(t)}) \boldsymbol{p}_4 \approx 0
\tag{242}
$$

By the algorithm, $\boldsymbol{W}_{O_{(i,\cdot)}}$ will update along the direction of $\boldsymbol{p}_2$ for $i \in \mathcal{W}_2$. We can analyze $\boldsymbol{W}_V$, $\boldsymbol{W}_K$ and $\boldsymbol{W}_Q$ similarly.

## G.2 Extension to a More General Data Model

We generalize the patterns from vectors to sets of vectors. Consider that there are $M$ ($2 < M < m_a, m_b$) distinct sets $\{\mathcal{M}_1, \mathcal{M}_2, \cdots, \mathcal{M}_M\}$ where $\mathcal{M}_l = \{\boldsymbol{\mu}_{l,1}, \boldsymbol{\mu}_{l,2}, \cdots, \boldsymbol{\mu}_{l,l_m}\}$, $l_m \geq 1$. $\mathcal{M}_1$, $\mathcal{M}_2$ denote sets of discriminative patterns for the binary labels, and $\mathcal{M}_3, \cdots, \mathcal{M}_M$ are sets of non-discriminative patterns.

$$\kappa = \min_{1 \leq i \neq j \leq M, 1 \leq a \leq i_m, 1 \leq b \leq j_m} \|\boldsymbol{\mu}_{i,a} - \boldsymbol{\mu}_{j,b}\| > 0 \tag{243}$$

is the minimum distance between patterns of different sets. Each token $\boldsymbol{x}_l^n$ of $\boldsymbol{X}^n$ is a noisy version of one pattern, i.e.,

$$\min_{j \in [M], b \in [j_m]} \|\boldsymbol{x}_l^n - \boldsymbol{\mu}_{j,b}\| \leq \tau \tag{244}$$

Define that for $l, s$ corresponding to $b_1, b_2$, respectively,

$$\min_{j \in [M], b_1, b_2 \in [j_m]} \|\boldsymbol{x}_l^n - \boldsymbol{x}_s^n\| \leq \Delta, \tag{245}$$

we have $2\tau + \Delta < \kappa$.

To simplify our theoretical analysis, one can similarly rescale all tokens a little bit like in Assumption 3 such that tokens corresponding to patterns in the same pattern set has an inner product larger than 1, while tokens corresponding to patterns from different pattern sets has an inner product smaller than $\lambda < 1$.

Assumption 1 can be modified such that

$$\|\boldsymbol{W}_V^{(0)} \boldsymbol{\mu}_{j,b} - \boldsymbol{p}_{j,b}\| \leq \sigma \tag{246}$$

$$\|\boldsymbol{W}_K^{(0)} \boldsymbol{\mu}_{j,b} - \boldsymbol{q}_{j,b}\| \leq \delta \tag{247}$$

$$\|\boldsymbol{W}_Q^{(0)} \boldsymbol{\mu}_{j,b} - \boldsymbol{r}_{j,b}\| \leq \delta \tag{248}$$

where $\boldsymbol{p}_{j,b} \perp \boldsymbol{p}_{i,a}$ for any $i, j \in [M]$, $b \in [j_m]$, and $a \in [i_m]$. Likewise, $\boldsymbol{q}_{j,b} \perp \boldsymbol{q}_{i,a}$ for any $i, j \in [M]$, $b \in [j_m]$, and $a \in [i_m]$. $\boldsymbol{r}_{j,b} \perp \boldsymbol{r}_{i,a}$ for any $i, j \in [M]$, $b \in [j_m]$, and $a \in [i_m]$.

Therefore, we make sure that the initial query, key and value features from different sets of patterns are still close to be orthogonal to each other. Then, we can follow our main proof idea. To be more specific, for label-relevant tokens $\boldsymbol{x}_l^n$, by computing (125) and (167), $\boldsymbol{W}_K^{(t)} \boldsymbol{x}_l^n$, $\boldsymbol{W}_Q^{(t)} \boldsymbol{x}_l^n$ will grow in the direction of a fixed linear combination of $\boldsymbol{q}_{l,1}, \cdots, \boldsymbol{q}_{l,l_m}$, and $\boldsymbol{r}_{l,1}, \cdots, \boldsymbol{r}_{l,l_m}$. The coefficient of the linear combination is a function of fractions of different pattern vectors $\boldsymbol{\mu}_{l,b}$ in $\mathcal{M}_l$. One can still derive a sparse attention map with weights of non-discriminative patterns decreasing to be close to zero during the training.

## G.3 Extension to Multi-head Networks

Suppose there are $H$ heads in total. The network is modified to

$$F(\boldsymbol{X}^n) = \frac{1}{|\mathcal{S}^n|} \sum_{l \in \mathcal{S}^n} \boldsymbol{a}_{(l)}^\top \text{Relu}(\boldsymbol{W}_O \overset{H}{\underset{h=1}{\big\|}} \sum_{s \in \mathcal{S}^n} \boldsymbol{W}_{V_h} \boldsymbol{x}_s^n \text{softmax}(\boldsymbol{x}_s^{n\top} \boldsymbol{W}_{K_h}^\top \boldsymbol{W}_{Q_h} \boldsymbol{x}_l^n))$$

$$= \frac{1}{|\mathcal{S}^n|} \sum_{l \in \mathcal{S}^n} \boldsymbol{a}_{(l)}^\top \text{Relu}(\sum_{h=1}^H \boldsymbol{W}_{O_h} \sum_{s \in \mathcal{S}^n} \boldsymbol{W}_{V_h} \boldsymbol{x}_s^n \text{softmax}(\boldsymbol{x}_s^{n\top} \boldsymbol{W}_{K_h}^\top \boldsymbol{W}_{Q_h} \boldsymbol{x}_l^n)) \tag{249}$$

where $\boldsymbol{W}_{V_h} \in \mathbb{R}^{m_a \times d}$, $\boldsymbol{W}_O = (\boldsymbol{W}_{O_1}, \boldsymbol{W}_{O_2}, \cdots \boldsymbol{W}_{O_H}) \in \mathbb{R}^{m \times H m_a}$, and $\boldsymbol{W}_{O_h} \in \mathbb{R}^{m \times m_a}$ for $h \in [H]$.

One can make similar assumptions for $\boldsymbol{W}_{Q_h}^{(0)}$, $\boldsymbol{W}_{K_h}^{(0)}$, and $\boldsymbol{W}_{V_h}^{(0)}$ as in Assumption 1. Note that $\{\boldsymbol{p}_1, \boldsymbol{p}_2, \cdots, \boldsymbol{p}_M\}$ needs to be changed into $\{\boldsymbol{p}_{h_1}, \boldsymbol{p}_{h_2}, \cdots, \boldsymbol{p}_{h_M}\}$, and the set $\{\boldsymbol{p}_{h_1}, \boldsymbol{p}_{h_2}, \cdots, \boldsymbol{p}_{h_M}\}$ can vary for different $h \in [H]$. It is also the same for $\{\boldsymbol{q}_{h_1}, \boldsymbol{q}_{h_2}, \cdots, \boldsymbol{q}_{h_M}\}$ and $\{\boldsymbol{r}_{h_1}, \boldsymbol{r}_{h_2}, \cdots, \boldsymbol{r}_{h_M}\}$ for different $h \in [H]$.

Based on the modified assumption with $H$ heads, the backbone of the proof remains the same. Lucky neurons in $\boldsymbol{W}_O$ tend to learn $(\boldsymbol{p}_{1_1}^\top, \boldsymbol{p}_{2_1}^\top, \cdots, \boldsymbol{p}_{H_1}^\top)^\top$ and $(\boldsymbol{p}_{1_2}^\top, \boldsymbol{p}_{2_2}^\top, \cdots, \boldsymbol{p}_{H_2}^\top)^\top$. Hence, the properties of the Relu activation are almost the same as the single-head case because luck neurons are still activated by either of two label-relevant patterns with a high probability. In fact, one can expect a more stable training process by multiple heads due to a more stable Relu gate for lucky neurons.

## G.4 EXTENSION TO SKIP CONNECTIONS AND NORMALIZATION

Consider a basic case where a skip connection is added after the self-attention layer. Let $m_a = d$. The network is changed into

$$F(\boldsymbol{X}^n) = \frac{1}{|\mathcal{S}^n|} \sum_{l \in \mathcal{S}^n} \boldsymbol{a}_{(l)}^\top \text{Relu}(\boldsymbol{W}_O(\sum_{s \in \mathcal{S}^n} \boldsymbol{W}_V \boldsymbol{x}_s^n \text{softmax}(\boldsymbol{x}_s^{n\top} \boldsymbol{W}_K^\top \boldsymbol{W}_Q \boldsymbol{x}_l^n) + \boldsymbol{x}_l^n)) \tag{250}$$

The assumption of $\boldsymbol{W}_V^{(0)}$ in Assumption 1 should be changed into

$$\|(\boldsymbol{W}_V^{(0)} + \boldsymbol{I})\boldsymbol{\mu}_j - \boldsymbol{p}_j\| \le \sigma, \tag{251}$$

while the assumption of $\boldsymbol{W}_Q^{(0)}$ and $\boldsymbol{W}_K^{(0)}$ remain the same.

One can easily verify that the gradients of $\boldsymbol{W}_K$, $\boldsymbol{W}_Q$, and $\boldsymbol{W}_V$ for (250) are almost the same as those for (1) except for the Relu gate. The major differences come from the gradient of $\boldsymbol{W}_O$, which also helps to determine the Relu gate. One needs to redefine

$$\begin{aligned}
\boldsymbol{V}_l^n(t) &= \boldsymbol{W}_V^{(t)} \boldsymbol{X}^n \text{softmax}(\boldsymbol{X}^{n\top} \boldsymbol{W}_K^{(t)\top} \boldsymbol{W}_Q^{(t)} \boldsymbol{x}_l^n) + \boldsymbol{x}_l^n \\
&= \sum_{s \in \mathcal{S}_1} \text{softmax}(\boldsymbol{x}_s^{n\top} \boldsymbol{W}_K^{(t)\top} \boldsymbol{W}_Q^{(t)} \boldsymbol{x}_l^n) \boldsymbol{p}_1 + \boldsymbol{z}(t) + \sum_{j \ne 1} W_j^n(t)(\boldsymbol{x}_j^n + \boldsymbol{x}_l^n) \\
&\quad - \eta(\sum_{i \in \mathcal{W}(t)} V_i(t) \boldsymbol{W}_{O_{(i,\cdot)}}^{(t)\top} + \sum_{i \notin \mathcal{W}(t)} V_i(t) \lambda \boldsymbol{W}_{O_{(i,\cdot)}}^{(t)\top})
\end{aligned} \tag{252}$$

for $l \in \mathcal{S}_1^n$. The inner product between the lucky neuron and the term $\sum_{j \ne 1} W_j^n(t)(\boldsymbol{x}_j^n + \boldsymbol{x}_l^n)$ can still be upper bounded by the inner product between the lucky neuron and the term $\sum_{s \in \mathcal{S}_1} \text{softmax}(\boldsymbol{x}_s^{n\top} \boldsymbol{W}_K^{(t)\top} \boldsymbol{W}_Q^{(t)} \boldsymbol{x}_l^n) \boldsymbol{p}_1$ given good initialization of $\boldsymbol{W}_K$ and $\boldsymbol{W}_Q$. Therefore, (250) can be analyzed following our proof techniques.

For layer normalization, one usually use that approach to normalize each data. It is consistent with our normalization of $\boldsymbol{x}_l^n$, which plays an important role in our proof. By normalization, the training process becomes more stable because of the unified norm of all tokens.

