# OpenReview forum: "A Theoretical Understanding of Shallow Vision Transformers: Learning, Generalization, and Sample Complexity"
_ICLR.cc/2023/Conference — ICLR 2023 poster_

### Official Review · Reviewer_3WM1 · 2022-10-24

**Confidence:** 3
**Correctness:** 3
**Technical Novelty And Significance:** 2
**Empirical Novelty And Significance:** 2
**Recommendation:** 6

**Clarity, Quality, Novelty And Reproducibility:**

While the main paper is structured clearly, the appendix needs major revision. The proofs are hard to track in the current form, making it hard to check the soundness.
Here are some suggestions:
  - Laying out an outline in English before diving into equations, and around each lemma or claim, explain what it means in English as well as how it connects to the rest of the proof.
  - Since Lemma 2 is used in the proof in Sec B, it perhaps makes sense to have Lemma 2 stated in Sec B, and leave its proof in Sec C.
  - Breaking Sec C into subsections (e.g. one for each claim of Lem 2, and one for Lem 3) could also help with readability.

Other minor points:
- Please standardize the axes for Fig 1 and 2. For example, all plots could use $\alpha_*$ or $\sigma$ as the x-axis, and use $\frac{1}{\sqrt{N}}$ or $T$ for the y-axis.
- Some typos (among others; please proofread):
  - The line below eq (28): $\mathcal{K}+_+$.
  - Proof of Prop 1, the first line: "we need to modify (40)" rather than (50).


**Strength And Weaknesses:**

To my knowledge, this paper is the among the first to analyze the training dynamics of Transformers on binary classification. The data setup (e.g. patterns for multiple classes can co-exist in the same input sample) and proof ideas seem quite similar to those of Allen-Zhu and Li (2020), which is however not cited in the paper.

The main proof idea is to track the growth of a set of lucky neurons, and show that the attention weights concentrates on the label-relevant patterns. I didn't check the proof details too carefully since I find the current appendix hard to track; please see my comments on writing.

 The paper provides empirical results to corroborate the theory which is appreciated, though I think the empirical section needs some improvement; please see my comments below.

Questions:
- "token sparsification method": what sparsification is used in Theorem 1 (i.e. how is $S^n$ defined)? Regarding designing better token sparsification method, how do we know what tokens are label-irrelevant in practice?
- Prop 2: is there a finite step result, i..e at what rate (in terms of $T$) does the sum of the weights goes to 1?
- Sec 4.1
    - How are $\delta, \sigma$ enforced in practice?
    - Sample complexity: why $10^{-3}$ as the threshold? Fig 3 seems to track till reaching a much lower loss.
    - Fig 3-5: are these over 20 duplicates as well?
    - Fig 5: is the shadow area standard deviation or standard error?
    - For Fig 4 and 5, using a larger $M$ (currently $M = 5$) will make the trends clearer and the results more interesting.
- Sec 4.2
    - Fig 6: are all models trained till convergence? Will architectures other than Transformers follow the same trend as $\alpha_*$ varies?
    - Fig 6(a): currently the number of samples are not sufficient to get (near) perfect accuracy; what if we further increase the number of samples? Will the lines for different $\alpha_*$ cross?
        - A more direct comparison may be to plot the number of samples required to reach a certain accuracy (e.g. 99%), as we vary $\alpha_*$, e.g. for $\alpha_* \in \{0.1, 0.2, \cdots, 0.9, 1\}$; the choices of $\alpha_*$ currently seems a bit arbitrary.

Comments:
- The counting of layers is a bit non-standard.
  - The model uses 1 attention layer w/ a 2-layer MLP, which is usually called a 1-layer Transformer (since a Transformer layer includes attention + MLP). Moreover, $W_O$ is usually considered as part of the attention layer, rather than an extra layer.
  - The paper considers the model as 3-layer possibly by counting the weight matrices, however, there is no nonlinearity between $W_O$  and $W_V$ so they can be considered as 1 layer?
- There are some sloppiness in the proof; for instance, $\ll$ in eq (34) and $\approx$ in eq (35) are not precisely quantified.

Reference: Allen-Zhu and Li 2020: https://arxiv.org/abs/2012.09816

**Summary Of The Paper:**

This paper provides a sample complexity bound for Transformers, as well as the required number of SGD steps. These bounds are for the model to achieve 0 generalization loss (using the hinge loss).

The task is binary classification, where the label depends  only on the "label-relevant patterns" in the input. The bound suggests that the sample complexity is improved as we 1) increase the fraction of label-irrelevant patterns, or 2) reduce the initial error of the model parameters (how close are initial encoding to an orthogonal basis; see Def 1), or 3) reduce the noise in the tokens.

The paper also provides empirical results to corroborate the theory.

**Summary Of The Review:**

This paper studies an interesting and challenging problem: the optimization of Transformers.
On the plus side, the paper is among the first to provide such results (there is a concurrent work: https://arxiv.org/abs/2210.09221) and such type of results is of great interests to the community.

However, I'm currently concerned about the novelty of the techniques and the presentation.
I'm willing to update my review after the rebuttal period & revisions.

===========

Update post-rebuttal: my concerns have been addressed by the author responses & the revision. I have raised my score accordingly.

---

### Official Review · Reviewer_8Nb6 · 2022-10-24

**Confidence:** 3
**Clarity, Quality, Novelty And Reproducibility:** This paper is clearly organized and e…
**Correctness:** 3
**Technical Novelty And Significance:** 3
**Empirical Novelty And Significance:** 2
**Recommendation:** 6

**Strength And Weaknesses:**

**Strength**

1. This paper is among the first to formally study the generalization of ViT like neural network models.

2. This paper characterizes sample complexity and evolution of the attention map for such models.

**Weakness**

1. The architecture studied in this paper contains a self attention module, but is still quite different from ViT models, even when restricting to one encoder/decoder block. In particular, there is no normalization layer, and the skip connections are not present. A closer simplified model would be four layers (1 attention layer, 2 fully connected layers for the MLP, 1 final frozen random prediction layer) with skip connections (and better, also with normalization layers).

2. This paper, while making one step closer towards understanding ViTs comparing to previous theoretical papers studying general DNNs and CNNs, is still a big overclaim. The theoretical model is just one encoder block (which is also drastically simplified, see the previous point), and it relied on the inputs already having clearly separating class relevant and irrelevant token embeddings (the experiments also uses ViTs that are already pre-trained on other datasets). I think the title should at least say this is an understanding of *shallow* ViTs.

**Summary Of The Paper:**

This paper theoretically studies the generalization of a multi-layer neural network containing a self-attention module (resembling a simplified vision transformer), trained with SGD on a dataset consists of a mixture of label-relevant and irrelevant tokens. This paper is the first to formally study the generalization of ViT-like neural networks. The analysis characterize how attention map evolves during training and the effects on sample complexity and reduced spurious correlation.

**Summary Of The Review:**

This paper is a theoretical study of the learning and generalization of a 3-layer networks inspired by vision transformer (ViT) models. This is one step towards, but far from a full theoretical understanding of ViTs. That being said, this seems to be one of the first papers making such attempts. I'm willing to raise my rating if the authors clearly address these limitations n the title and abstract, and if other reviewers who are more familiar with the theories acknowledge that there are novelties in the proof techniques and analysis.

------
Thanks for the clarifications, and modification to the paper to make the topic of study more precise. I increased my rating.

---

### Official Review · Reviewer_sr6i · 2022-10-25

**Confidence:** 3
**Correctness:** 4
**Technical Novelty And Significance:** 3
**Empirical Novelty And Significance:** 3
**Recommendation:** 6

**Clarity, Quality, Novelty And Reproducibility:**

The work is very clear, and high quality. No code is provided which limits the reproducibility.

For notation, I have a small point on clarity.

- On eq (1), should there be a transpose for $a_{(l)}$? It seems like there should be, since we are denoting the inner product of two vectors.

**Strength And Weaknesses:**

Strengths
- A valuable problem to study and valuable theoretical results.
- Experiments around token sparsification are compelling (Fig 6).

Weaknesses
- Only single head attention is studied, even though multi-head attention is typically used.

**Summary Of The Paper:**

The paper provides theoretical results on training three-layer ViTs for classification tasks. The authors quantify the importance of self-attention on sample complexity for zero generalization error, as well as the sparsity of attention maps when being trained by SGD. The authors then also show that token sparsification can improve generalization performance by removing class-irrelevant tokens and noisy tokens.

**Summary Of The Review:**

Ultimately, this work provides valuable contributions towards understanding transformer architectures, for which there is not enough theory. While there are some concerns around reproducibility, the contributions of this work outweighs this minor issue.

---

### Official Review · Reviewer_7JzE · 2022-10-31

**Confidence:** 3
**Correctness:** 3
**Technical Novelty And Significance:** 4
**Empirical Novelty And Significance:** 4
**Recommendation:** 6

**Clarity, Quality, Novelty And Reproducibility:**

The writing is generally clear, the topic is introduced well and the main results are easy to understand and to follow. As mentioned above, the theoretical analysis of optimization together with generalization of vision transformer is novel.

**Strength And Weaknesses:**

To my knowledge, this is the first work on theoretical analysis of learning with vision transformers, and one of the few works studying the theory of transformers in general. The fact that the authors analyze both optimization and generalization of transformers is unique, and allows the authors to draw novel conclusions on the theoretical properties of transformers, e.g. showing that the attention maps converge to sparse maps.

There are a few problems I still find with the paper:
1. The theoretical results comparing training with and without self-attention maps are not convincing in my view. To establish a real "separation" between training with and without the self-attention map, showing that the former is indeed better, the authors need to show that training without self-attention requires more samples or training steps (i.e., giving a lower-bound on the sample/computational complexity). Instead, the authors show that using the same technical analysis leads to an inferior upper-bound when not using self-attention. While the authors also admit this in a comment, I think the presentation of the result still suggests that such a separation is established.
2. I do not understand the results on token sparsification: How is the sparsification done? Do you assume prior knowledge about the distribution (namely, which patches are relevant/irrelevant)? Is the sparsification done before or after training? How do you make sure that the relevant patches are not removed?
3. The theoretical results are limited to a very specific data distribution. While I do not necessarily see this as a major problem, since it allows the authors to show stronger results in this context, it would be good to understand whether the results can be generalized beyond this specific case. For example, can this result be extended to the case where there is more than one relevant basis patch for each of the two classes? Is the result on achieving sparse maps in the presence of irrelevant patches true in more general cases? Can we show, for a general distribution, that patches which do not affect the label get zero weight in the self-attention layer?
4. It seems to me that Assumption 1 should hold with high-probability over the initialization, and also that \sigma (the error of the initial model) can be bounded with high-probability. Why is this given as a separate assumption?

**Summary Of The Paper:**

The paper gives a theoretical analysis of a simple (three layer) vision transformer network. The analysis is focused on training a ViT on a simple distribution of images constructed from label-relevant and label-irrelevant patches. Namely, there is a dictionary of M patches, and each patch in the input image is a noisy version of one of the patches. Only 2 out of the M patches are important for determining the label, which is the majority over the number of patches from these two classes of patches. The authors show that when training a ViT using SGD on this distributions, SGD converges to a solution with low error on the distribution. Furthermore, they analyze the sample and runtime complexity of SGD and show their dependence on different parameters of the distribution. The authors also show that the self-attention maps converge to sparse maps, which depend on the relevant/irrelevant patches. Finally, the authors complement their finding with different synthetic experiments.

**Summary Of The Review:**

The paper is overall good, with novel contributions, but I still find some problems with the paper that could be improved (mentioned above).

---

### Decision · Program_Chairs · 2023-01-20

**Decision:**

Accept: poster

**Justification For Why Not Higher Score:**

The reviewers highlight some limiting simplifications of the analysis, thus somewhat limiting the impact of scope of the contribution, although it is still an important step forward.

**Justification For Why Not Lower Score:**

As stated above, the paper contains a solid theoretical contribution towards understanding vision transformers and would thus be of interest to the ICLR community.

**Metareview: Summary, Strengths And Weaknesses:**

The paper provides a theoretical analysis of a transformer architecture, providing optimization and generalization guarantees. The results nicely show the evolution of the learned model towards a sparse attention map.

**Note From Pc:**

if the above contains the word "oral" or "spotlight" please see: "oral" presentation means -> notable-top-5% and "spotlight" means -> notable-top-25%. As stated in our emails, we are disassociating presentation type from AC recommendations

**Summary Of Ac-Reviewer Meeting:**

Upon online discussion, the reviewers raised their score to 6 and above, indicating support for acceptance and thus I don't view this paper as borderline.